# QUANTILE-OPTIMAL POLICY LEARNING UNDER UNMEASURED CONFOUNDING

## ABSTRACT

We study quantile-optimal policy learning where the goal is to find a policy whose reward distribution has the largest $\alpha$-th quantile for some $\alpha \in (0, 1)$. We focus on the offline setting whose generating process involves unobserved confounders. Such a problem suffers from three main challenges: (i) nonlinearity of the quantile objective as a functional of the reward distribution, (ii) unobserved confounding issue, and (iii) insufficient coverage of the offline dataset. To address these challenges, we propose a suite of causal-assisted policy learning methods that provably enjoy strong theoretical guarantees under mild conditions. In particular, to address (i) and (ii), using causal inference tools such as instrumental variables and negative controls, we propose to estimate the quantile objectives by solving nonlinear functional integral equations. Then we adopt a minimax estimation approach with nonparametric models to solve these integral equations, and propose to construct conservative policy estimates that address (iii). The final policy is the one that maximizes these pessimistic estimates. In addition, we propose a novel regularized policy learning method that is more amenable to computation. Finally, we prove that the policies learned by these methods are $\widetilde{\mathcal{O}}(n^{-1/2})$ quantile-optimal under a mild coverage assumption on the offline dataset. To our best knowledge, we propose the first sample-efficient policy learning algorithms for estimating the quantile-optimal policy when there exists unmeasured confounding.

## 1 INTRODUCTION

Offline reinforcement learning (RL) (Levine et al., 2020; Prudencio et al., 2023) aims to learn the optimal decision-making policies from pre-collected datasets, where the learner has no control over the data collection process. This lack of control is further complicated by the fact that many real-world datasets suffer from incomplete information due to unmeasured confounders (Rubin, 1974)—factors influencing both the actions and rewards, yet not observed by the learner. These unobserved confounders naturally arise in the *partially observable* setting, where the behavior policy used to generate the actions is contingent on the state, while only noisy observations of the state are recorded in the data. A prime example can be found in healthcare, where electronic health records (EHR) may lack key details about a patient's condition or treatment environment (Hernán & Robins, 2016), but decisions about treatment still need to be optimized based on the available data.

In many decision-making scenarios, rather than expected rewards, we are interested in optimizing more nuanced objectives involving quantiles of the reward distribution. For instance, to measure the efficacy of job training programs, it may be more meaningful to optimize for the median income increase, as the mean could be skewed by a few extreme cases. Moreover, quantile-based objectives naturally arise when some notion of fairness is of interest (Yang et al., 2019; Liu et al., 2022).

In this paper, we tackle the problem of quantile-optimal policy learning in offline settings where unmeasured confounders play a critical role. Our goal is to find a policy that maximizes the $\alpha$-th quantile of the reward distribution, conditioning on the context. Such a policy is learned on a pre-collected offline dataset that involves unobserved confounders, which are some hidden variables that simultaneously affect the rewards and actions stored in the data.

This problem is particularly challenging due to three core issues: (i) the quantile objective is a nonlinear functional of the reward distribution, making statistical estimation more difficult; (ii) unmeasured confounders introduce bias into the estimation problem, which can lead to misleading results if not properly accounted for; and (iii) the offline dataset often lacks full coverage, meaning

that the distribution of the collected data might have insufficient overlap with that induced by some candidate policy, making it challenging to evaluate the performance of that policy.

To address these challenges, we propose a suite of novel, causal-assisted policy learning approaches. Our methods leverage powerful causal inference techniques, such as instrumental variables (IV) (Angrist et al., 1996; Baiocchi et al., 2014) and negative controls (NC) (Lipsitch et al., 2010; Tchetgen et al., 2020), to account for unmeasured confounding. Leveraging these causal inference tools, we frame the quantile-based estimation problem as solving **nonlinear functional integral equations**, which can be solved using a minimax estimation approach with nonparametric models. This enables us to address Challenges (i) and (ii). Furthermore, to handle Challenge (iii), we propose to adopt the pessimism principle (Jin et al., 2021; Xie et al., 2021; Rashidinejad et al., 2021; Buckman et al., 2020; Lu et al., 2022), which enables us to relax the requirement on data coverage. We introduce two versions of the algorithms based on the pessimism idea – a constrained version and a regularized version. In particular, our method works as long as the offline data has sufficient coverage over the optimal policy, and it does not matter whether the data contains sufficient information about the suboptimal policies — pessimism ignores policies with large uncertainty and enables us to only focus on the policies that are covered by the dataset.

We prove that our algorithms achieve sample efficient $\widetilde{O}(n^{-1/2})$ suboptimality bounds, where $n$ is the sample size. Compared to existing works that study offline policy learning with confounded datasets aiming to maximize expected rewards, our analysis is more complicated due to the nonlinear nature of the quantile-based objective. In particular, to get the suboptimality bounds, we leverage the local curvature of the nonlinear operator in the functional estimating equation. We examine the statistical error of the estimated quantile-based objective in terms of a local norm, defined using the first-order Taylor expansion of the nonlinear operator. To our best knowledge, we establish the first provably sample-efficient algorithms for quantile-optimal policy learning under unmeasured confounding and insufficient support.

## 1.1 RELATED WORKS

**Offline Decision Making**  Our work is built upon the general framework of offline reinforcement learning (Yin et al., 2022; Uehara et al., 2021; Jin et al., 2021; Xie et al., 2021; Rashidinejad et al., 2021) and offline contextual bandits (Li et al., 2012; Lee et al., 2021; Metevier et al., 2019). In particular, Cassel et al. (2023); Zhu & Tan (2020); Prashanth et al. (2020) study nonlinear objectives in contextual bandits without confounding bias. To address the challenge of insufficient coverage in the offline dataset, we incorporate the principle of pessimism (Jin et al., 2021; Xie et al., 2021; Rashidinejad et al., 2021; Buckman et al., 2020; Lu et al., 2022; Chen et al., 2023). Our algorithm leverages the idea of pessimism to handle the challenge of insufficient data coverage. Our problem has two additional challenges due to the nonlinearity of the quantile objective and the unobserved confounders, which are not addressed in most of these works.

**Causal Inference**  A substantial body of work in causal inference has focused on addressing confounding bias through covariate adjustment (Rubin, 1974; Rosenbaum & Rubin, 1983; Lee et al., 2010; Liu et al., 2024), instrumental variables (Angrist et al., 1996; Ai & Chen, 2003; Newey & Powell, 2003; Chen et al., 2003; Chernozhukov & Hansen, 2008; Chen & Pouzo, 2012; Baiocchi et al., 2014; Hartford et al., 2017), and negative controls (Miao et al., 2018; Kallus et al., 2021). Our estimation procedure leverages tools from nonlinear nonparametric instrumental variables, e.g., (Chen & Pouzo, 2012) and negative controls (Miao et al., 2018; Kallus et al., 2021). We need to additionally incorporate pessimism for policy learning, which leads to a more complicated algorithm and analysis.

**Pessimism + Causal Inference**  There are existing works that combine pessimism with causal inference tools to establish algorithms for confounded offline decision making (Dong et al., 2023; Chen et al., 2023). Our work is particularly relevant to Chen et al. (2023), which studies a similar setting, with the goal of maximizing the expected return. In contrast, we study the quantile objective, which involves the nonlinear estimating equation, and thus requires a more sophisticated analysis.

**Quantile Policy Learning**  Quantile treatment effect has been extensively studied in Econometrics (Abadie et al., 2002; Chernozhukov & Hansen, 2005; Horowitz & Lee, 2007; Chernozhukov & Hansen, 2008; Chen & Pouzo, 2012; Gagliardini & Scaillet, 2012). However, limited research has explored quantile policy learning. For instance, Wang et al. (2018) investigates quantile-optimal

treatment regimes, while Linn et al. (2017) proposes quantile regression methods to indirectly and approximately optimize the quantile of outcomes within specific decision rule classes. Additionally, Fang et al. (2023) incorporates the quantile of the reward as a regularization term in their objective to maximize the average reward.

## 2 OFFLINE DECISION MAKING PROBLEM SETUP

In this section, we formalize the problem of offline decision making under the confounded contextual bandit model. Let $A \in \mathcal{A}$ be the action and $\Delta(\mathcal{A})$ denote the set of all distributions over $\mathcal{A}$. Let $X \in \mathcal{X}$ be the context, and $Y \in \mathbb{R}$ be the reward. Denote $O$ as the side observations that can assist in decision making. In this paper, $O$ can be either the Instrumental Variables (IVs) $Z$ (Angrist et al., 1996) or the negative control exposure-outcome pair $(E, V)$(Lipsitch et al., 2010). Informally, IVs are random variables that affect the reward only through the actions. The Negative Control Exposure (NCE), denoted by $E$, are random variables known not to causally affect the reward $Y$. The Negative Control Outcomes (NCO), denoted as $V$ are random variables known to be causally unaffected by either the action $A$ or $E$. We will elaborate on these side observations in detail later. Let $U \in \mathcal{U}$ denote the unmeasured confounders that causally affects $A$ and $Y$ simultaneously. We use the lower-case letter to indicate a realization of the corresponding random variable. The decision-making process involves two steps: first, collecting an offline dataset, and then learning a policy from the offline data to apply in an interventional process.

**The offline data collection process** The offline data collection process (ODCP) describes the process by which the offline dataset is collected. ODCP generates $n$ samples, $\{u_i, x_i, o_i, a_i, y_i\}_{i=1}^n$. We assume tuples of random variables are independent and identically distributed. As $U$ is unmeasured, the offline dataset only includes $\{x_i, a_i, o_i, y_i\}_{i=1}^n$. A motivating example is the dataset of job training programs aimed at improving workers' income distributions (Abadie et al., 2002; Chernozhukov & Hansen, 2008). Here, $Y_i$ represents the utility for the $i$-th worker, defined as the difference between the increase in wage and the cost of participating in the program. $X_i$ captures the worker's pre-intervention covariates. $A_i$ denotes whether the $i$-th worker participates in the program. $U_i$ could represent unmeasured factors such as the worker's motivation, which influence both their participation decision $A_i$ and their utility outcomes $Y_i$. $O_i$ serves as an instrumental variable Angrist et al. (1996), which could be whether the worker was invited to participate in the program.

**The Interventional Procecss** A policy $\pi : \mathcal{X} \to \Delta(\mathcal{A})$ is defined as a mapping from the context to a distribution over the action space, specifying the decision-making rule. From the offline dataset, we learn a policy $\hat{\pi}$, which is subsequently applied in an interventional process. In this process, we no longer observe the side observations. Moreover, the context $X$ follows a different but known distribution from that in the ODCP. We denote the new distribution as $\widetilde{p}(x)$. The environment first generates the context $X \sim \widetilde{p}(x)$. The action $a$ is then selected by the agent using the learned policy $\hat{\pi}$, based solely on the newly generated $X$. In the job training program example, this corresponds to the government using the learned policy to decide whether to allow a new worker to join the program based on their pre-intervention covariates. When $O$ represents the IV, $Z$, Figure 1 illustrates possible causal directed acyclic graphs (DAGs) for both the ODCP and the interventional process. We provide the details of the ODCP, interventional process, and the motivated examples for the case of observing NCs as the side observations in §J.1.

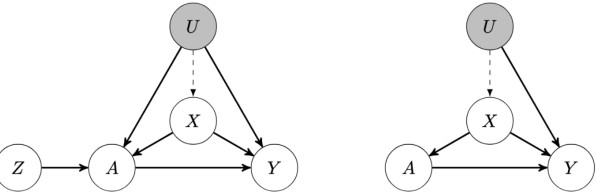

Figure 1: The DAGs when the IVs are observed as the side observations. The dashed edge implies that the causal relationship may be absent. The grey node indicates that $U$ is unmeasured. **Left**: DAG for the ODCP. The IV $Z$ causally affects $Y$ only through $A$. **Right**: DAG for the interventional process. All arrows coming into the node $A$ have been removed other than the one from the node $X$.

**Quantile Objective and Regret.** We denote the expectation over the distribution of random variables in ODCP as $\mathbb{E}$. For a fix $\pi$, we denote the expectation over the distribution of random variables in the interventional process as $\mathbb{E}_{p_{\text{in}}^\pi}$. Existing works focus on learning the policy that maximizes the average reward, $\mathbb{E}_{p_{\text{in}}^\pi}[Y]$ (Lu et al., 2022; Guo et al., 2022). The main goal of this paper, however, is to maximize the average structural quantile of the reward. Formally, given $X = x$, for any action $a$, we let $h_\alpha^*(a, x)$ denote the $\alpha$-th quantile of the potential outcome $Y(a)$. In the language of do-calculus (Pearl, 1995), it holds that $\mathbb{P}[Y \leq h_\alpha^*(A, X) \,|\, X = x, \text{do}(A = a)] = \alpha$. We assume that $Y \,|\, A, X$ is continuously distributed without atoms so that $h_\alpha^*$ is unique. We do not assume that $h_\alpha^*$ takes any parametric form. Our goal is to learn a policy $\pi^* : \mathcal{X} \to \Delta(\mathcal{A})$ that maximizes the average structural quantile function:

$$v_\alpha^\pi := \mathbb{E}_{p_{\text{in}}^\pi}[h_\alpha^*(A, X)]. \tag{2.1}$$

In real-world applications, the objective shifts from maximizing the average outcome to optimizing a quantile of the outcome distribution when the focus is on distributional fairness. For the job training examples, governments aim to ensure that the majority of workers experience income gains rather than optimizing the average utility, which can be skewed by extreme values. Thus, the focus is on maximizing the median or lower quantiles to address income inequality. To measure the performance of our estimated $\hat\pi$, we define the notion of regret for a fixed $\pi$ as

$$\text{Regret}(\pi) := v_\alpha^{\pi^*} - v_\alpha^\pi. \tag{2.2}$$

We remark that our objective is a nonlinear functional of the distribution of $Y$. Unlike the average reward scenario, learning the optimal policy and theoretically analyzing the rate of the regret pose significant challenges.

## 3 DESIGN OF ALGORITHMS

In this section, we identify key challenges that arise in quantile policy learning and propose algorithms specifically designed to address these challenges.

### 3.1 CAUSAL IDENTIFICATION

Optimizing the average quantile function $\mathbb{E}_{p_{\text{in}}^\pi}[h_\alpha^*(A, X)]$ over $\pi$ hinges on first estimating $h_\alpha^*(A, X)$. While a naive approach involves a quantile regression of $Y$ on $A$ and $X$, the unmeasured confounders $U$ bring the endogeneity into the ODCP, making this approach inappropriate for the structural quantile function (Chernozhukov & Hansen, 2008; Chen et al., 2014; Horowitz & Lee, 2007). To address the confounding bias, we propose leveraging the side observations observed in ODCP to transform the estimation of $h_\alpha^*(A, X)$ into solving an integral equation $\mathcal{T}(h) = 0$ for $h$, where $\mathcal{T}$ is a nonlinear operator, and $h_\alpha^*(A, X)$ is a solution of the equation.

For instance, denote $D := (Y, X, A, O)$. When the side observations are instrumental variables $Z$, we write $\mathbf{1}\{Y \leq h(A, X)\} - \alpha$ as $W(D; h)$. Let $h \in \mathcal{H} \subseteq (X, Z) \to \mathbb{R}$ where $\mathcal{H}$ is a functional space that we will determine later. Chen et al. (2014) shows that, under certain regularity conditions, for the nonlinear operator $\mathcal{T}^{\text{IV}} : \mathcal{H} \to L_2(p_{X,Z})$ defined as $\mathcal{T}^{\text{IV}}h := \mathbb{E}[W(D; h) \,|\, X, Z]$, we have $\mathcal{T}^{\text{IV}}h_\alpha^*(A, X) = 0$ almost surely (See details of the derivation of this in §A). When the side observations is the negative control exposure-outcome pair $(E, V)$, we derive a novel set of integral equations that $h_\alpha^*$ satisfies. Denote

$$\mathcal{T}_1(h_1, h_2) := \mathbb{E}[W(D; h_1) - h_2(V, A, X) \,|\, E, A, X],$$
$$\mathcal{T}_2(h_1, h_2) := \mathbb{E}[h_2(V, A', X) \,|\, A', X],$$

where $A'$ is drawn uniformly in the action space $\mathcal{A}$, independent of other variables. The nonlinear operators $\mathcal{T}_1$ and $\mathcal{T}_2$ are defined on their corresponding domains, with values in the $L_2$ spaces of square-integrable functions.

**Theorem 3.1 (Informal Version of Conditional Moment Restrictions for Negative Controls)**
*Let $\mathcal{T}^{NC} := (\mathcal{T}_1, \mathcal{T}_2)$. Under certain regularity conditions imposed on the NCs, there exists bridge functions $h_2^* : \mathcal{V} \times \mathcal{A} \times \mathcal{X} \to \mathbb{R}$ so that*

$$\mathcal{T}^{NC}(h_\alpha^*, h_2^*) = 0. \tag{3.1}$$

**Proof** *See §J.2 for the complete statement of the Theorem and a detailed proof.* ∎

The equation of the form $\mathcal{T}(h) = 0$ is referred to as conditional moment restrictions in econometrics (Chamberlain, 1992; Ai & Chen, 2003; Newey & Powell, 2003; Chen & Pouzo, 2009b). To our best knowledge, Theorem 3.1 is the first conditional moment restrictions result for the structural quantile when negative controls are observed. Consequently, estimating $h_\alpha^*$ reduces to solving the conditional moment restrictions $\mathcal{T}^{\text{IV}}h = 0$ or $\mathcal{T}^{\text{NC}}h = 0$ for $h$. Throughout the remainder of the paper, we primarily focus on the case where IVs are observed in the ODCP, due to space constraints. Similar algorithms and theoretical guarantees apply to the negative control setting, which is discussed in detail in §J. The only distinction between IVs and NCs is that $\mathcal{T}^{\text{NC}}$ requires solving two conditional moment restrictions simultaneously, while $\mathcal{T}^{\text{IV}}$ involves solving one. This difference is minimal, allowing us to use a similar approach for both cases. In fact, our methodology applies to any side observation for which the relationship $\mathcal{T}h = 0$ can be established. For simplicity, we use $\mathcal{T}$ to refer to $\mathcal{T}^{\text{IV}}$ in the main text.

### 3.2 MINIMAX ESTIMATION AND PESSIMISM PRINCIPLE

Solving $\mathcal{T}(h) = 0$ for $h \in \mathcal{H}$ is a nonparametric nonlinear inverse problem. It is generally impossible to find a closed-form solution. Here, inspired by the minimax approach of Dikkala et al. (2020) for estimating nonparametric mean IV regression $\mathbb{E}\left[Y - h(A, X) \mid X, Z\right] = 0$, we propose to use a minimax approach to estimate nonparametric quantile IV function $h_\alpha^*$. Specifically, $\frac{1}{2}\|\mathcal{T}(h)\|_2^2$ can be estimated by leveraging Fenchel duality:

$$\mathcal{L}_n(h) := \sup_{\theta \in \Theta} \left\{ \frac{1}{n} \sum_{i=1}^n \left[W(D_i; h)\theta(X_i, Z_i)\right] - \frac{1}{2n} \sum_{i=1}^n \theta^2(X_i, Z_i) \right\} \tag{3.2}$$

for a chosen real-valued test function class $\Theta$ on $\mathcal{X} \times \mathcal{Z}$. Thus, we can construct an estimator of $h_\alpha^*$ by minimizing $\mathcal{L}_n(h)$ over $h \in \mathcal{H}$ for a suitable hypothesis function space $\mathcal{H}$. We refer the reader to §B for the details of this derivation. However, it would be problematic if we directly plug in the greedy solution $\hat{h} := \arg\inf_{h \in \mathcal{H}} \mathcal{L}_n(h)$ and maximize $v(\hat{h}, \pi) = \mathbb{E}_{p_{\text{in}}^\pi}[\hat{h}(A, X)]$ over $\pi$. The sub-optimal context-action pairs may mislead the learned policy by making the variance of $\hat{h}$ high, due to insufficient sample size (Jin et al., 2021). See §B for a detailed discussion. To tackle this, we learn a pessimistic policy by doing uncertainty quantification on $\mathcal{L}_n(h)$. Specifically, we first construct a solution set (SS), $\mathcal{S}(e_n)$, for $h$ based on $\mathcal{L}_n(h)$ as:

$$\mathcal{S}(e_n) := \left\{ h \in \mathcal{H} : \mathcal{L}_n(h) \le \inf_{h \in \mathcal{H}} \mathcal{L}_n(h) + e_n \right\}, \tag{3.3}$$

where $e_n$ is a small positive threshold such that $h_\alpha^* \in S(e_n)$. We will determine it later based on our novel theoretical analysis. Then we select the policy that optimizes the pessimistic average structural quantile function:

$$\hat{\pi} := \arg\sup_\pi \inf_{h \in \mathcal{S}(e_n)} v(h, \pi). \tag{3.4}$$

We name the policy learning procedure in Equation (3.4) as the **Solution Set Algorithm**.

### 3.3 THE REGULARIZED ALGORITHM

As an intermediate step to learn $\hat{\pi}$, we need to minimize $\mathcal{L}_n(h)$ for $h$ restricted to $h \in \mathcal{S}(e_n)$ in the Solution Set Algorithm. This is an optimization problem with a data-dependent constraint, which is often computationally intractable. Therefore, we propose a computationally benign version of the policy learning algorithm. Let $\mathcal{E}_n(h) = \mathcal{L}_n(h) - \inf_{h \in \mathcal{H}} \mathcal{L}_n(h)$. We modify the objective function in equation 3.4 by adding a regularization term $\lambda_n \mathcal{E}_n(h)$. The regularized version of the pessimistic policy is then defined as:

$$\hat{\pi}_R := \arg\sup_\pi \inf_{h \in \mathcal{H}} \{v(h, \pi) + \lambda_n \mathcal{E}_n(h)\}. \tag{3.5}$$

The proposed approach reformulates the original constrained optimization problem in Equation (3.4), into its augmented lagrangian counterpart (Rashidinejad et al., 2022). The additional regularization term $\lambda_n \mathcal{E}_n(h)$ penalizes $\mathcal{L}_n(h)$, which is equivalent to penalizing the threshold of the

uncertainty constraint equation 3.3. Furthermore, when the estimator $h$ is close to $h_\alpha^*$, the regularization term approaches $\lambda_n \mathcal{E}_n(h_\alpha^*)$, thereby becoming a small quantity. Consequently, we expect less bias when optimizing the augmented lagrangian over $\pi$. We called the policy learning procedure in Equation (3.5) the **Regularized Algorithm**. In practice, the optimization of $h \in \mathcal{H}$ over the regularized objective can be implemented by gradient-based methods. For example, Xie et al. (2021) uses the mirror descent method to learn the pessimistic policy.

**Minimax Estimation for Negative Controls**   The minimax estimation for negative controls closely mirrors that of instrumental variables, with the key difference being the need to solve two conditional moment restrictions simultaneously. In §J.3, we provide the detailed design of the solution set and the regularized version of the algorithms for negative controls.

## 4   THEORETICAL RESULTS

We provide a theoretical analysis of the rate of the regret of the policy $\hat{\pi}$ and $\hat{\pi}_R$. Deriving the regret of a quantile-optimal policy is surprisingly challenging, mainly due to two factors: (1) the standard concentration inequalities used in the minimax literature does not work for $\|\mathcal{T}h\|_2$ because of the nonlinearity of $\mathcal{T}$, and (2) it is difficult to bound the regret in terms of $\|\mathcal{T}h\|_2$. To address the first challenge, we use bracketing number techniques from van de Geer (2009); Chen et al. (2003). For the second challenge and the solution set algorithm, we first demonstrate that $h_\alpha^* \in \mathcal{S}(e_n)$ with high probability and that $\|\mathcal{T}h\|_2$ converges at a fast rate uniformly over $h \in \mathcal{S}(e_n)$. We then define

$$\hat{h}^\pi := \arg \inf_{h \in \mathcal{S}(e_n)} v(h, \pi). \tag{4.1}$$

We show that we can link the regret of $\hat{\pi}$ to $\|\mathcal{T}\hat{h}^{\pi^*}\|_2$ by performing a local expansion of $\mathcal{T}\hat{h}^{\pi^*}$ around $h_\alpha^*$, and hence address the challenge (2). Finally, we derive the convergence rate of the regret of $\hat{\pi}$. For regularized algorithm, we define a quantity analogous to $\hat{h}^\pi$ as:

$$\hat{h}_R^\pi := \arg \inf_{h \in \mathcal{H}} \{v(h, \pi) + \lambda_n \mathcal{E}_n(h)\}.$$

We then perform a local expansion of $\mathcal{T}\hat{h}_R^{\pi^*}$ around $h_\alpha^*$ and bound the regret in a similar manner as in the solution set algorithm. Although we are only demonstrating the analysis when instrumental variables are observed in this section, every theorem presented has an equivalent counterpart for negative controls, as detailed in §K. We also include a discussion on the stated assumptions and conditions in this section in §N. To begin with, we impose a few conditions on the hypothesis and test function spaces.

**Assumption 4.1 (Identifiability and Realizability)** *We assume the structural quantile function $h_\alpha^* \in \mathcal{H}$. For any $h \in \mathcal{H}$ with $\mathbb{E}\left[W(D; h) \mid X, Z\right] = 0$, we have $\|h - h_\alpha^*\|_\infty = 0$.*

**Assumption 4.2 (Compatibility of Test Function Class)** *For any $h \in \mathcal{H}$, $\inf_{\theta \in \Theta} \|\theta - \mathcal{T}h\|_2 = \epsilon_\Theta$, and $\epsilon_\Theta = \widetilde{\mathcal{O}}(n^{-1/2})$.*

**Assumption 4.3 (Regularity of Function Classes)** *We assume $\mathcal{H}$ is compact with respect to the norm $\|\cdot\|_\infty$. We say a function class $\mathbb{F}$ is star-shaped if for every $f \in \mathcal{F}$ and for any $r \in [0, 1]$, we have $rf \in \mathcal{F}$. We assume $\Theta$ is star-shaped. We also assume the support of $h_\alpha^*$ is bounded by $L_Y > 0$. Moreover, it holds that $\sup_{h \in \mathcal{H}} \|h\|_\infty \leq B_h$ and $\sup_{\theta \in \Theta} \|\theta\|_\infty \leq B_\theta$.*

Assumption 4.1 states that the hypothesis space $\mathcal{H}$ captures $h_\alpha^*$ and $h_\alpha^*$ is the unique solution to the conditional moment restriction. Assumption 4.2 ensures that the test function class is rich enough to approximate $\mathcal{T}h$ for all $h \in \mathcal{H}$. Assumption 4.3 can be easily satisfied by choosing $\mathcal{H}$ and $\Theta$ to be standard uniformly bounded and closed function classes. Due to the form of $\mathcal{L}_n(h)$, it is natural to consider the function class

$$\mathcal{Q} := \{W(\cdot; h) \times \theta(\cdot) : h \in \mathcal{H}, \theta \in \Theta\}.$$

We then define an event that quantifies the approximation error between each of the two terms inside the supremum in $\mathcal{L}_n(h)$ and its population counterpart. This event will later be proved to

have a high probability by using empirical process theory. Fix a sequence of small positive constant $\eta_n$ that decreases with $n$. Let $\mathcal{E}$ denote the event

$$\mathcal{E} := \left\{ \left| \mathbb{E}_n \left[ q(D) \right] - \mathbb{E} \left[ q(D) \right] \right| \leq \eta_n \left( \|\theta\|_2 + \eta_n \right), \left| \|\theta\|_{n,2}^2 - \|\theta\|_2^2 \right| \leq \frac{1}{2} \left( \|\theta\|_2^2 + \eta_n^2 \right), \forall q \in \mathcal{Q} \right\}. \quad (4.2)$$

**Condition 4.4** *Suppose that Assumption 4.3 holds. For any $\xi > 0$, there exists $\eta_n > 0$ such that the event $\mathcal{E}$ holds with probability at least $1 - 2\xi$.*

In §E, we establish that Condition 4.4 is satisfied for some $\eta_n = \widetilde{\mathcal{O}}(n^{-1/2})$ given that $\mathcal{H}$ and $\Theta$ are suitably selected. Here, $\widetilde{\mathcal{O}}(k)$ denotes $\mathcal{O}(k \cdot \text{poly}(\log k))$. Consequently, Assumption 4.2 can be stated as $\inf_{\theta \in \Theta} \|\theta - \mathcal{T}h\|_2 = \eta_n^2$.

All the theoretical results in this section are built on the event $\mathcal{E}$. For the rest of the paper, we assume the condition 4.4 holds. We first present the analysis of the solution set version algorithm. We then circle back to the regularized algorithm in Section 4.3. We begin by decomposing the regret (2.2) of $\hat{\pi}$ as:

$$\text{Regret}(\hat{\pi}) = \underbrace{v_\alpha^{\pi^*} - v(\hat{h}^{\pi^*}, \pi^*)}_{(i)} + \underbrace{v(\hat{h}^{\hat{\pi}}, \pi^*) - v(\hat{h}^{\hat{\pi}}, \hat{\pi})}_{(ii)} + \underbrace{v(\hat{h}^{\hat{\pi}}, \hat{\pi}) - v_\alpha^{\hat{\pi}}}_{(iii)}. \quad (4.3)$$

Since $\pi^*$ in term (i) is the oracle policy, it does not depend on our chosen estimator $\hat{h}$. We will show that term (i) can be controlled by imposing a mild condition on the distribution shift of the context between ODCP and the interventional process. Term (ii) is bounded by zero by the optimality of $\hat{\pi}$. To control term (iii), we first demonstrate that on the event $\mathcal{E}$, the solution set $\mathcal{S}(e_n)$ exhibits some favorable properties. In the following, we use the notation $\overset{\mathcal{E}}{\lesssim}$ and $\overset{\mathcal{E}}{\gtrsim}$ to represent the inequality that holds on the event $\mathcal{E}$.

**Theorem 4.5 (Uncertainty Quantification)** *Suppose that Assumptions 4.1, 4.2, and 4.3 hold.*

*(i). $\mathcal{L}_n(h_\alpha^*) \overset{\mathcal{E}}{\lesssim} \frac{13}{4}\eta_n^2$. Moreover, if we set $e_n > \frac{13}{4}\eta_n^2$, then $h_\alpha^* \in \mathcal{S}(e_n)$.*

*(ii). $\sup_{h \in \mathcal{S}(e_n)} \|\mathcal{T}h\|_2 \overset{\mathcal{E}}{\lesssim} \mathcal{O}\left(\sqrt{e_n}\right) + \mathcal{O}\left(\eta_n\right)$.*

**Proof** *See §G.1 for a detailed proof.* ∎

Theorem 4.5 (i) states that when $e_n$ is chosen properly, the solution set $\mathcal{S}(e_n)$ captures $h_\alpha^*$ with high probability. This fact is notably important for managing term (iii), and results in the following Corollary 4.6. Theorem 4.5 (ii) characterizes the rate of conditional RMSE for all $h$ in $\mathcal{S}(e_n)$. In subsequent discussions, we will show that $\|\mathcal{T}\hat{h}^{\pi^*}\|_2$ can be linked to the upper bound of the regret of $\hat{\pi}$ stated in Corollary 4.6.

**Corollary 4.6 (Regret Decomposition)** *Suppose that Assumptions 4.1, 4.2 and 4.3 hold. If $e_n > \frac{13}{4}\eta_n^2$, then the regret corresponding to $\hat{\pi}$ is bounded by*

$$\text{Regret}(\hat{\pi}) \overset{\mathcal{E}}{\lesssim} v_\alpha^{\pi^*} - v(\hat{h}^{\pi^*}, \pi^*) = \mathbb{E}_{p_{in}^{\pi^*}} \left[ h_\alpha^*(A, X) - \hat{h}^{\pi^*}(A, X) \right]. \quad (4.4)$$

**Proof** *See §G.4 for a detailed proof.* ∎

### 4.1 LOCAL EXPANSION WITHIN THE SOLUTION SET

The upper bound in Corollary 4.6 is linear in $\hat{h}^{\pi^*}$ while $\mathcal{T}$ is a nonlinear operator on $\mathcal{H}$. Bridging $\|\mathcal{T}\hat{h}^{\pi^*}\|_2$ to the regret of $\hat{\pi}$ is a nontrivial task. We demonstrate that employing a local expansion of $\mathcal{T}\hat{h}^{\pi^*}$ around $h_\alpha^*$ emerges as a suitable method. A necessary step to perform the local expansion is to ensure that our estimator is consistent to $h_\alpha^*$.

**Assumption 4.7 (Regularity of Conditional Density)** *We assume that the conditional density of $Y$ given $(A, X, Z)$, $p_{Y \mid A,X,Z=(a,x,z)}(y)$ is continuous in $(y, a, x, z)$ and $\sup_y p_{Y \mid A,X,Z}(y) < K$ for some $K > 0$ for almost all $(A, X, Z)$.*

Assumption 4.7 imposes some mild assumptions on the conditional density of $Y$ given $(A, X, Z)$, which is used to ensure that $\mathcal{T}$ is a continuous operator with respect to $\| \cdot \|_\infty$. The assumption is standard in NPQIV (Chen & Pouzo, 2012). We now give a consistent result for any $h \in \mathcal{S}(e_n)$.

**Theorem 4.8 (Consistency of the Estimator of the Solution Set Algorithm)** *Suppose that Assumptions 4.1, 4.2, 4.3 and 4.7 hold. If $e_n = \mathcal{O}(\eta_n^2,)$ then on the event $\mathcal{E}$, for any $h \in \mathcal{S}(e_n)$, $\|h - h_\alpha^*\|_\infty = o_p(1)$ and hence $\|h - h_\alpha^*\|_2 = o_p(1)$.*

**Proof** *See §G.2 for a detailed proof.* ∎

Theorem 4.8 shows any $h$ in $\mathcal{S}(e_n)$ is a consistent estimator of $h_\alpha^*$. With consistency, one remaining step is required before advancing with local expansion: establishing the correct definition of the derivative of $\mathcal{T}h$ with respect to $h$. Following Ai & Chen (2012), we introduce the notion of pathwise derivative of $\mathcal{T}h$, $\frac{d\mathcal{T}h_\alpha^*}{dh}$, and its associated weak norm:

$$||h - h_\alpha^*|| := \sqrt{\mathbb{E}\left[\left(\frac{d\mathcal{T}h_\alpha^*}{dh}[h - h_\alpha^*]\right)^2\right]}. \tag{4.5}$$

The formal definition of the pathwise derivative can be found in Lemma H.1. Let $\mathcal{H}_\epsilon := \{h \in \mathcal{H} : \|h - h_\alpha^*\|_2 \le \epsilon\} \cap \mathcal{S}(e_n)$ be the restricted space of $\mathcal{S}(e_n)$ around the neighborhood of $h_\alpha^*$, where $\epsilon$ is a sufficiently small positive number such that $\mathbb{P}(\mathcal{E} \cap \{\hat{h}^{\pi^*} \in \mathcal{H}_\epsilon\}) \ge 1 - 3\xi$. Such $\epsilon$ is guaranteed to exist by Condition 4.4 and Theorem 4.8.

**Assumption 4.9 (Local Curvature for the Estimator of the Solution Set Algorithm)** *If we set $e_n > \frac{13}{4}\eta_n^2$, then there exists a sufficiently small positive number $\epsilon$ and a finite constant $c_0 > 0$ such that for any $h \in \mathcal{H}_\epsilon$, $||h - h_\alpha^*|| \le c_0 \|\mathcal{T}h\|_2$.*

Assumption 4.9 is where we do the local expansion. The constant $c_0$ controls the approximation error of local expansion.

### 4.2 REGRET OF $\hat{\pi}$

Assumption 4.9 allows us to do local expansion and link $||h - h_\alpha^*||$ to $\|\mathcal{T}h\|_2$. Combining this with the regret decomposition yields the rate of the regret for $\hat{\pi}$.

**Assumption 4.10 (Change of Measure)** *For the marginal distribution of context $\widetilde{p}$ in the interventional process and the optimal interventional policy $\pi^*$, suppose there exists a function $b : \mathcal{X} \times \mathcal{Z} \to \mathbb{R}$ such that $\mathbb{E}\left[b^2(X, Z)\right] < \infty$ and*

$$\mathbb{E}\left[b(X, Z)p_{Y|A,X,Z}(h_\alpha^*(A, X)) \mid A = a, X = x\right] = \frac{\widetilde{p}(x)\pi^*(a \mid x)}{p(x, a)}. \tag{4.6}$$

Assumption 4.10 can be interpreted as a condition that requires the offline data in ODCP to cover the distribution induced by the oracle policy $\pi^*$.

**Theorem 4.11 (Rate of Convergence of the Regret of the Solution Set Algorithm)** *Suppose that structural model Assumption A.1 holds. Suppose that Assumptions 4.1, 4.2, and 4.3 for function classes $\mathcal{H}$ and $\Theta$ hold. Suppose also that the Assumptions 4.9 and 4.10 hold. If the threshold $e_n$ for the solution set is set to $e_n > \frac{13}{4}\eta_n^2$, then the regret corresponding to $\hat{\pi}$ is bounded on event $\mathcal{E} \cap \left\{\hat{h}^{\pi^*} \in \mathcal{H}_\epsilon\right\}$ by*

$$Regret(\hat{\pi}) \lesssim c_0 \|b\|_2 \cdot \left(\mathcal{O}\left(\sqrt{e_n}\right) + \mathcal{O}\left(\eta_n\right)\right).$$

**Proof** *See §G.5 for a detailed proof.* ∎

This theorem shows that the regret is bounded by the product of the expansion error $c_0$, the $\ell_2$ norm of the change-of-measure function $b$ and $\mathcal{O}\left(\sqrt{e_n}\right) + \mathcal{O}\left(\eta_n\right)$. If we set $e_n = \mathcal{O}(\eta_n^2)$, Theorem 4.11 gives a rate of $\mathcal{O}(\eta_n)$. In Appendix E, we show that $\eta_n = \widetilde{\mathcal{O}}(n^{-1/2})$ when $\mathcal{H}$ and $\Theta$ are linear spaces, which corresponds to a "fast statistical rate" (Uehara et al., 2021; Dikkala et al., 2020; Li et al., 2022). Notably, this rate is identical to that in settings where the average reward, rather than the average quantile, is maximized (Chen et al., 2023). This indicates that there is no loss of efficiency when extending the analysis to the nonlinear case. In Theorem K.9, we show that when the negative controls are observed, the regret of the solution set algorithm shares a same order of $\widetilde{\mathcal{O}}(n^{-1/2})$

### 4.3 THEORETICAL ANALYSIS FOR THE REGULARIZED ALGORITHM

For the regularized version of the algorithm, we apply a similar approach to $\hat{h}_R^\pi$ and $\hat{\pi}_R$. We prove that $\hat{h}_R^\pi$ is consistent to $h_\alpha^*$. Consequently, we perform a local expansion of $\hat{h}_R^{\pi^*}$ around $h_\alpha^*$, allowing the regret to be linked to $\mathcal{T}\hat{h}_R^{\pi^*}$. The informal versions of the theorems of consistency and regret rate are presented below, with the detailed analysis provided in §C.

**Theorem 4.12 (Informal Version of the Consistency of the Estimator of the Regularized Algorithm)**
*Under certain regularity conditions, for any $\epsilon_{\lambda_n} > 0$, if we set $\lambda_n \geq \eta_n^{-(1+\epsilon_{\lambda_n})}$, then on the event $\mathcal{E}$, for any $\pi$, $\left\|\hat{h}_R^\pi - h_\alpha^*\right\|_\infty = o_p(1)$ and hence $\left\|\hat{h}_R^\pi - h_\alpha^*\right\|_2 = o_p(1)$.*

**Proof** *See §C.3 for the complete statement and §G.3 for a detailed proof.* ∎

Theorem 4.12 states that on the event $\mathcal{E}$, the estimated structural quantile function $\hat{h}_R^\pi$ is consistent to the ground-truth $h_\alpha^*$. Note the regularization parameter $\lambda_n$ in Theorem C.3 only depends on $\eta_n$, which is the rate of convergence of the tail bounds of the chosen function classes.

**Theorem 4.13 (Informal Version of the Convergence of Regret of the Regularized Algorithm)**
*Under certain regularity conditions, for any $0 < \epsilon_{\lambda_n} < 1$, if the regularized parameter $\lambda_n$ is set to $\lambda_n = \eta_n^{-(1+\epsilon_{\lambda_n})}$. Then with high probability, the regret of $\hat{\pi}_R$ is bounded by*

$$Regret(\hat{\pi}_R) \lesssim \mathcal{O}(\eta_n^{1-\epsilon_{\lambda_n}}).$$

**Proof** *See §C.4 for the complete statement and §G.6 for a detailed proof.* ∎

Theorem 4.13 implies that if we set $\lambda_n = \eta_n^{-(1+\epsilon_{\lambda_n})}$, the regret rate of $\hat{\pi}_R$ would be of $\mathcal{O}(\eta_n^{1-\epsilon_{\lambda_n}})$. Recall by Theorem 4.11, Regret($\hat{\pi}$) has a rate of $\mathcal{O}(\eta_n)$. Thus the rate of $\hat{\pi}_R$ is only slower than that of $\hat{\pi}$ by an infinitesimal amount $\epsilon_{\lambda_n}$. The quantity $\epsilon_{\lambda_n}$ appears in the definition of $\lambda_n$ and is crucial for the consistency result in Theorem C.3. Intuitively, when $\epsilon_{\lambda_n}$ is small, the rate of convergence of $\hat{h}_R^{\pi^*}$ to $h_\alpha^*$ is slow. Consequently, the local expansion error $c_0$ in Assumption 4.9 could be large, which in terms leads to a slower regret rate. Essentially, $\epsilon_{\lambda_n}$ can be viewed as the trade-off when converting an optimization problem with data-dependent constraints into a nonconstrained optimization problem. We further observe that when selecting the function spaces $\mathcal{H}$ and $\Theta$ as the linear function classes, our achieved rate of $\widetilde{\mathcal{O}}(n^{-1/2(1-\epsilon_{\lambda_n})})$ surpasses the $\widetilde{\mathcal{O}}(n^{-1/3})$ rate of the regularized version reported in Xie et al. (2021), and closely matches the $\widetilde{\mathcal{O}}(n^{-1/2})$ rate in Rashidinejad et al. (2022). Note that Rashidinejad et al. (2022) requires the knowledge of the propensity score, which is a strong assumption and is not required by our method. In addition, we focus on estimating the nonlinear mapping of the distribution function, in contrast to the linear mapping in their work. In Theorem K.12, we show that when the negative controls are observed, the regret of the regularized policy shares a same order of $\widetilde{\mathcal{O}}(n^{-1/2(1-\epsilon_{\lambda_n})})$.

**Other Applications** Although we present the analysis of a general structural quantile model, our algorithm remains applicable as long as the problem can be reformulated into solving conditional moment restrictions. When it is impossible to establish the conditional moment restrictions, we can still apply our methodology in some cases. We provide several such examples in §D.

## 5 EXPERIMENT

In this section, we evaluate the performance of the regularized version of the algorithm through simulation experiments. We consider the case where instrumental variables are observed in a synthetic ODCP. To investigate the algorithm's robustness, we consider offline datasets that contain $p\%$ of action distributed as random noise, independent of $(X, Z, U)$, and affecting only $Y$. We set $p$ to be 20, 50 and 70. We take this approach to deliberately contaminate the offline dataset, enabling us to evaluate the robustness of our algorithm. One of the key strengths of pessimism Jin et al. (2021) is its ability to handle suboptimal actions that could mislead the learned policy due to undercoverage in the offline dataset. Through the simulation experiment, we aim to demonstrate this advantage. For implementation, we let both $\mathcal{H}$ and $\Theta$ be some linear function classes. Given a fix $h \in \mathcal{H}$ and $\pi$, the optimization procedure over $\Theta$ can be carried out by deriving the closed-form solution of a standard nonparametric regression problem. We then optimize over $\mathcal{H}$ by substituting the closed-form solution over $\Theta$ and performing the gradient descent. Finally, we update the optimal policy $\hat{\pi}_R$ by solving a linear programming problem. The complete details regarding the data generating process for the synthetic dataset and the implementation of the algorithm can be found in §I. Figure 2 characterizes the regret plotted against the sample size $n$, illustrating the empirical convergence rate of the algorithms. We can see that as $n$ increases, the values and variability of the regrets for the pessimistic algorithm decrease rapidly. In addition, the pessimistic algorithm is able to achieve low regrets for large sample sizes even if $70\%$ of the dataset is contaminated.

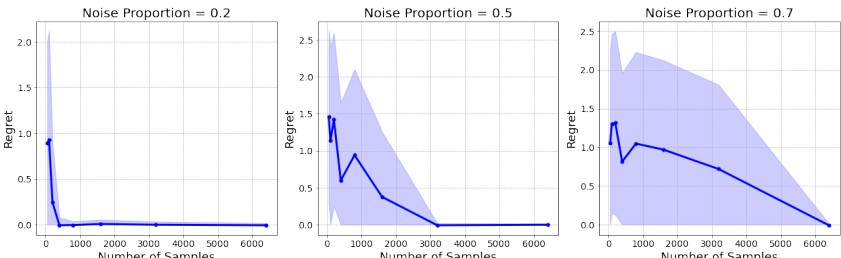

Figure 2: The line plots of the regret over the sample size in $\{50, 100, 200, 400, 800, 1600, 3200, 6400\}$. Each point is the average regret of 20 trials. The boundary of the shaded area corresponds to the values of average regrets plus or minus the standard deviation. The noise proportions refer to the proportion of sample actions that are random noise.

## 6 DISCUSSION

In this work, we propose a policy learning algorithm where the goal is to find a policy whose reward distribution has the largest $\alpha$-th quantile for some $\alpha \in (0, 1)$. To our best knowledge, this is the first sample-efficient policy learning algorithm for estimating the quantile-optimal policy under unmeasured confounding. Our work takes the IVs and NCs as examples of the side observations to tackle the confounding bias. We demonstrate that the proposed algorithm is equally effective when other side observations with different underlying causal structures are observed in the ODCP, as long as the conditional moment restrictions can be established.

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

## A  CAUSAL IDENTIFICATION FOR INSTRUMENTAL VARIABLES

We describe our approach to tackling the problem of confounding bias when we observe IV as the side observation from the ODCP. We introduce the nonparametric quantile IV model (NPQIV) in the ODCP and the conditional moment restrictions it follows.

A standard nonparametric instrumental variables model requires the instrumental variables $Z$ to satisfy three conditions: (i) relevance: the distribution of action is not constant in the instrumental variables, (ii) exclusion: the instrumental variables affect the reward only through action, and (iii) unconfoundedness: the instrumental variables is conditionally independent of the error term (Newey & Powell, 2003; Hartford et al., 2017). The general NPQIV model, however, necessitates a more stringent set of assumptions (Chen et al., 2014). Following the treatment of Section 6 in Chen & Pouzo (2012), we employ a simplified version of the NPQIV model incorporating the additive error assumption and conditional moment restriction:

**Assumption A.1 (Model Assumption for Quantile IV)** *We assume that the following conditions hold for the IV model in the ODCP:*

$$Y = h_\alpha^*(A, X) + \epsilon \text{ and } \mathbb{P}(\epsilon \leq 0 \mid X, Z) = \alpha.$$

By Chen et al. (2014), with some regularity conditions, Assumption A.1 holds by the underlying causal structure of the ODCP in the IV case, as depicted in Figure 1. In contrast to a standard regression model where the mean of the error term is assumed to be 0, we define $\epsilon$ as the structural error. This error term adheres to a distinct conditional distribution assumption and is responsible for the confounding effect induced by the unmeasured confounders $\mathcal{U}$. Consequently, $\epsilon$ is not independent of $(A, X)$ and the condition $\mathbb{P}(\epsilon \leq 0 \mid X, A) = \alpha$ does not hold. This subtlety renders the conventional quantile regression of $Y$ on $A$ and $X$ erroneous. Fortunately, the existence of the IV allows us to write Assumption A.1 as

$$\mathbb{E}\left[\mathbf{1}\{Y \leq h_\alpha^*\} \mid X, Z\right] = \alpha.$$

If we write $\mathbf{1}\{Y \leq h_\alpha^*(A, X)\} - \alpha$ as $W(D; h_\alpha^*)$ where $D = (Y, X, A)$, we would then establish the relationship: $\mathbb{E}\left[W(D; h_\alpha^*) \mid X, Z\right] = 0$. If we further impose the condition of the uniqueness of the $h_\alpha^*$ that satisfies this relationship, we can obtain an estimator of $h_\alpha^*$ by solving the following conditional moment equation:

$$\mathbb{E}\left[W(D; h) \mid X, Z\right] = 0 \tag{A.1}$$

with respect to $h$.

## B  MINIMAX ESTIMATION AND PESSIMISTIC PRINCIPLE FOR THE IVS.

We first define a conditional residual mean squared error (RMSE) on equation A.1 with respect to $h$ :

$$\|\mathcal{T}h\|_2^2 := \mathbb{E}\left[\left(\mathbb{E}\left[W(D; h) \mid X, Z\right]\right)^2\right], \tag{B.1}$$

where $\mathcal{T}$ is the nonlinear operator defined as

$$\mathcal{T}h(\cdot) := \mathbb{E}\left[W(D;h) \mid (X,Z) = \cdot\right].$$

Note that for any function $h$ we have

$$\|\mathcal{T}h\|_2^2 \geq 0, \text{ and } \|\mathcal{T}h\|_2^2 = 0 \text{ if and only if } \mathbb{E}\left[W(D;h) \mid X, Z\right] = 0.$$

Therefore, we can construct an estimator $h \in \mathcal{H}$ of $h_\alpha^*$ by minimizing equation B.1 in a suitable hypothesis function space $\mathcal{H}$. However, Equation $(B.1)$ is a squared of a conditional expectation. Estimating it using the squared empirical expectation introduces variance terms, leading to bias. To address this, we first utilize the Fenchel duality of the function $x^2/2$ and reformulate $\frac{1}{2}\|\mathcal{T}h\|_2^2$ as

$$\frac{1}{2}\|\mathcal{T}h\|_2^2 = \mathbb{E}\left[\sup_\theta \mathcal{T}h(X,Z)\zeta - \frac{1}{2}\theta^2\right].$$

As the supremum of the dual form is achieved by $\zeta = \mathcal{T}h$, we have

$$\frac{1}{2}\|\mathcal{T}h\|_2^2 = \mathbb{E}\left[\sup_{\theta \in \Theta} \mathcal{T}h(X,Z)\theta(X,Z) - \frac{1}{2}\|\theta\|_2^2\right]$$

$$= \sup_{\theta \in \Theta}\left\{\mathbb{E}\left[W(D;h)\theta(X,Z)\right] - \frac{1}{2}\|\theta\|_2^2\right\}. \tag{B.2}$$

The last equality holds by the interchangeability principle (Dai et al., 2017). Therefore, we replace the problem of minimizing B.1 by minimizing B.2. Let $\mathcal{L}_n(h)$ denote the empirical version of equation B.2, i.e.,

$$\mathcal{L}_n(h) := \sup_{\theta \in \Theta}\left\{\mathbb{E}_n\left[W(D;h)\theta(X,Z)\right] - \frac{1}{2}\|\theta\|_{n,2}^2\right\}$$

$$= \sup_{\theta \in \Theta}\left\{\frac{1}{n}\sum_{i=1}^n \left[W(D_i;h)\theta(X_i,Z_i)\right] - \frac{1}{2n}\sum_{i=1}^n \theta^2(X_i,Z_i)\right\}, \tag{B.3}$$

where we introduce a real-valued test function class $\Theta$ on $\mathcal{X} \times \mathcal{Z}$ such that $\mathcal{T}h$ can be well approximated by some function in $\Theta$. As $\mathcal{L}_n(h)$ can now be computed from the offline data, we can then estimate $h_\alpha^*$ by minimizing $\mathcal{L}_n(h)$ with respect to $h$.

**The Pessimistic Principle.** A major challenge of offline decision making is to deal with the distribution shift between the ODCP that generates the observed action and the oracle policy. To see this, given $h$ and $\pi$, we denote $v(h,\pi) = \mathbb{E}_{p_{\text{in}}^\pi}\left[h(A,X)\right]$. Suppose we have already obtained an estimator $\hat{h}$, We then greedily maximize $v(\hat{h},\pi)$, returning $\widetilde{\pi} = \arg\sup_\pi v(\hat{h},\pi)$. We can decompose the regret equation 2.2 of $\widetilde{\pi}$ as:

$$\text{Regret}(\widetilde{\pi}) = v_\alpha^{\pi^*} - v_\alpha^{\widetilde{\pi}}$$

$$= \underbrace{v_\alpha^{\pi^*} - v(\hat{h},\pi^*)}_{\text{(i)}} + \underbrace{v(\hat{h},\pi^*) - v(\hat{h},\widetilde{\pi})}_{\text{(ii)}} + \underbrace{v(\hat{h},\widetilde{\pi}) - v_\alpha^{\widetilde{\pi}}}_{\text{(iii)}}, \tag{B.4}$$

As $\pi^*$ in tern (i) is the oracle policy, it does not depend on our chosen estimator $\hat{h}$. We will show that this term can be controlled by imposing a mild condition on the distribution shift of the context between ODCP and the interventional process in Section 4. Term (ii) is bounded by zero by the optimality of $\widetilde{\pi}$. Term (iii) presents a unique subtlety: $\widetilde{\pi}$ and $\hat{h}$ are both derived from offline data, creating spurious correlation. For example, when the distribution that generates the action is nearly deterministic, the action space is underexplored. The sub-optimal context-action pairs may mislead the learned policy $\widetilde{\pi}$ by making the variance of $\hat{h}$ high, due to insufficient sample size (Jin et al., 2021). To tackle this, we learn a pessimistic policy by doing uncertainty quantification on $\mathcal{L}_n(h)$. Specifically, we first construct a solution set $\mathcal{S}(e_n)$ for $h$ based on $\mathcal{L}_n(h)$ as:

$$\mathcal{S}(e_n) := \left\{h \in \mathcal{H} : \mathcal{L}_n(h) \leq \inf_{h \in \mathcal{H}} \mathcal{L}_n(h) + e_n\right\},$$

where $e_n$ is a small positive threshold we will determine later. We will show in Theorem 4.5 (i) that, with high probability, $h_\alpha^*$ lies in $\mathcal{S}(e_n)$ by choosing $e_n$ properly. Then we select the policy that optimizes the pessimistic average reward function:

$$\hat{\pi} := \arg\sup_\pi \inf_{h \in \mathcal{S}(e_n)} v(h, \pi).$$

In this way, (iii) can be upper bounded by zero due to our definition of $\hat{\pi}$ if we choose $\hat{h}$ to be $\arg\inf_{h \in \mathcal{S}(e_n)} v(h, \hat{\pi})$.

**The Solution Set (SS) Algorithm.** Here's a brief overview of our steps so far: Firstly, We have reframed the causal identification problem as a task of solving the conditional moment restriction, as depicted in Equation equation A.1. Secondly, inspired by the moment equation equation A.1, we devised a loss function equation B.2 that can be computed from the offline data. Thirdly, we construct a solution set equation 3.3 based on equation B.3. Finally, we obtain our learned policy $\hat{\pi}$ from the solution set equation 3.3. We summarize the algorithm in Algorithm 1.

---

**Algorithm 1** Quantile Effect Policy Learning

---

**Input:** Offline dataset $\{a_i, x_i, z_i, y_i\}_{i=1}^n$ from ODCP, hypothesis space $\mathcal{H}$, test function space $\Theta$ and threshold $e_n$.
  (i) Construct solution set $\mathcal{S}(e_n)$ as the level set of $\mathcal{H}$ with respect to metric $\mathcal{L}_n(\cdot)$ and threshold $e_n$.
  (ii) $\hat{\pi} = \arg\sup_\pi \inf_{h \in \mathcal{S}(e_n)} v(h, \pi)$.
**Output:** $\hat{\pi}$.

---

# C    THEORETICAL ANALYSIS OF THE REGULARIZED ALGORITHM

In this section, we provide a rigorous treatment on the regret analysis of the policy learned by the regularized algorithm when IVs are observed. Analogous to Corollary 4.6, we state an upper bound of $\text{Regret}(\hat{\pi}_R)$.

**Corollary C.1 (Regret Decomposition for the Regularized Algorithm)** *The regret corresponding to $\hat{\pi}_R$ is bounded by*

$$Regret(\hat{\pi}_R) \leq \mathbb{E}_{p_{in}^{\pi^*}} \left[ h_\alpha^*(A, X) - \hat{h}_R^{\pi^*}(A, X) \right] + \lambda_n \mathcal{E}_n(h_\alpha^*). \tag{C.1}$$

**Proof** *See the proof of Theorem C.4 for a detailed proof.*  ∎

Corollary C.1 states that the regret of $\hat{\pi}_R$ can be upper bounded by the average difference between $h_\alpha^*(A, X)$ and $\hat{h}_R^{\pi^*}(A, X)$ in the interventional process, plus a regularization term. Recall we define $\mathcal{E}_n(h_\alpha^*)$ as the difference $\mathcal{L}_n(h_\alpha^*) - \inf_{h \in \mathcal{H}} \mathcal{L}_n(h)$. The challenging aspect, bounding $\mathcal{L}_n(h_\alpha^*)$, has already been addressed in Theorem 4.5 (i). To handle $\inf_{h \in \mathcal{H}} \mathcal{L}_n(h)$, we introduce the following assumption:

**Assumption C.2 (Sample Criterion)** *We assume $\inf_{h \in \mathcal{H}} \mathcal{L}_n(h) = \mathcal{O}(\eta_n^2)$.*

Assumption C.2 is a prevalent condition in M-estimation theory (Van der Vaart, 2000). The assumption holds when the optimization method is effective. Consequently, the regularization term mentioned in Corollary C.1 can be effectively bounded. But how about the initial term? The strategy remains to associate it with the RMSE, $||\mathcal{T}\hat{h}_R^{\pi^*}||_2$, through Taylor expansion. We begin by affirming the consistency of $\hat{h}_R^\pi$.

**Theorem C.3 (Consistency of the Estimator of the Regularized Algorithm)** *Suppose that Assumptions 4.1, 4.2, 4.3, 4.7 and C.2 hold. For any $\epsilon_{\lambda_n} > 0$, if we set $\lambda_n \geq \eta_n^{-(1+\epsilon_{\lambda_n})}$, then on the event $\mathcal{E}$, for any $\pi$, $\left\| \hat{h}_R^\pi - h_\alpha^* \right\|_\infty = o_p(1)$ and hence $\left\| \hat{h}_R^\pi - h_\alpha^* \right\|_2 = o_p(1)$.*

**Proof** *See §G.3 for a detailed proof.*  ∎

**Theorem C.4 (Convergence of Regret of the Regularized Algorithm)** *Suppose that the structural model Assumption A.1 holds. Suppose that Assumptions 4.1, 4.2, and 4.3 for the function classes $\mathcal{H}$ and $\Theta$ hold. Suppose also that the regularity of density Assumption 4.7, the Sample Criterion Assumption C.2, the local curvature Assumption 4.9, and the data coverage Assumption 4.10 hold. Foy any $0 < \epsilon_{\lambda_n} < 1$, if the regularized parameter $\lambda_n$ is set to $\lambda_n = \eta_n^{-(1+\epsilon_{\lambda_n})}$. then the regret corresponding to $\hat{\pi}_R$ is bounded on event $\mathcal{E} \cap \left\{ \hat{h}_R^{\pi^*} \in \{h \in \mathcal{H} : ||h - h_\alpha^*||_2 \leq \epsilon\} \right\}$ by*

$$Regret(\hat{\pi}_R) \lesssim \mathcal{O}(\eta_n^{1-\epsilon_{\lambda_n}}).$$

**Proof** *See §G.6 for a detailed proof.* ∎

# D    OTHER APPLICATIONS

In this section, we show that our algorithm remains applicable as long as the problem can be reformulated into solving a conditional moment restriction. For instrumental variables, the conditional moment restriction $\mathbb{E}\left[W(D;h)\,|\,X,Z\right] = 0$, can be replaced by solving $\mathbb{E}\left[\widetilde{W}(D;h)\,|\,X,Z\right] = 0$, where $h$ is the parameter of interest, related to a functional $\widetilde{W}$ that is specified by the problem. In the case of negative controls, we can solve equations equation J.6 and equation J.7 after substituting $\widetilde{W}(D;h)$ for $W(D;h)$. In particular, our algorithm is effective with more restricted forms of quantile models.

**Example D.1 (Single Index Quantile Model (Wu et al., 2010))** *The structural quantile function is restricted to the form $h(A, X) = h_0(\theta_0^T(X, A))$, where $h_0$ and $\theta_0$ are unknown. The functional $\widetilde{W}(D;h)$ is then defined as $\mathbf{1}\{Y \leq h_0(\theta_0^T(X, A))\} - \alpha$. For this example to fall in our framework, we simply choose $\mathcal{H} = \widetilde{\mathcal{H}} \times \mathbb{R}^{dim\mathcal{X}} \times \mathbb{R}^{dim\mathcal{A}}$ to be the function class that contains the function of the form: $h : \mathcal{X} \times \mathcal{A} \to \mathbb{R}$ defined by $h(X, A) = \widetilde{h}(\theta_1^T X + \theta_2^T A)$.*

**Example D.2 (Partially Linear Quantile Model (Chen & Pouzo, 2009a))** *The structural quantile function is restricted to the form $h(A, X) = h_0(A) + \theta_0^T X$, where $h_0$ and $\theta_0$ are unknown. The functional $\widetilde{W}(D;h)$ is then defined as $\mathbf{1}\{Y \leq h_0(A) + \theta_0^T X\} - \alpha$. We then choose $\mathcal{H} = \widetilde{\mathcal{H}} \times \mathbb{R}^{dim\mathcal{X}}$, which contains the function of the form: $h : \mathcal{X} \times \mathcal{A} \to \mathbb{R}$ defined by $h(X, A) = \widetilde{h}(A) + \theta^T X$.*

When it is impossible to find a conditional moment restriction, We can still apply our approach in some cases. In Example D.3, we extend the method to a more general setting than maximizing the quantile function, demonstrating the flexibility of the proposed algorithm.

**Example D.3 (Quantile-Based Risk Measures (Dowd & Blake, 2006))** *We consider the spectral risk measure, $\int_0^1 w(\alpha)h_\alpha^* d\alpha$, which is a member of the family of quantile-based risk measures. Here, $w(\alpha)$ is a known weighting function that makes the spectral risk measure coherent. To estimate the spectral risk measure, we consider a finite index set $\mathcal{I} = \{\alpha_1, \ldots, \alpha_m\}$ such that $\alpha_i \in (0, 1)$ for any $i$. We then Define*

$$M_\phi(A, X; h) := \sum_{i=1}^m \phi(\alpha_i)h_{\alpha_i}(A, X). \tag{D.1}$$

*where $h = (h_{\alpha_1}, \ldots, h_{\alpha_m})$, for some known function $\phi$. The idea is that we simultaneously estimate $\{h_{\alpha_i}(A, X)\}_{i=1}^m$ for each $\alpha_i$ via the proposed algorithm. Then we approximate $-\int_0^1 w(\alpha)h_\alpha d\alpha$ by $M_\phi(A, X; h)$. The choice of $\phi$ depends on the numerical method we used to approximate the integral. For example, Newton–Cotes method is one popular choice that has the form of equation D.1. Given that $\phi(\alpha_i)$ is fixed, we now introduce the policy-learning algorithm that minimizes the spectral risk measure or, equivalently, maximizes the negative spectral risk measure through the use of instrumental variables. The case for negative controls is similar. For each $i \in [m]$, we define*

$$W_i(D; h_{\alpha_i}) := \mathbf{1}\{Y \leq h_{\alpha_i}(A, X)\} - \alpha_i,$$

$$\mathcal{L}_{i,n}(h_{\alpha_i}) := \sup_{\theta \in \Theta} \left\{ \mathbb{E}_n \left[W_i(D; h_{\alpha_i})\theta(X, Z)\right] - \frac{1}{2}||\theta||_{n,2}^2 \right\}.$$

*For the solution set version, we further let*

$$\mathcal{S}_i(e_n) := \left\{ h_{\alpha_i} \in \mathcal{H} : \mathcal{L}_{i,n}(h_{\alpha_i}) \leq \inf_{h_{\alpha_i} \in \mathcal{H}} \mathcal{L}_{i,n}(h_{\alpha_i}) + e_n \right\}, \textit{and}$$

$$\mathcal{S}(e_n) := \mathcal{S}_1(e_n) \times \cdots \times \mathcal{S}_m(e_n).$$

*The estimated policy is then given by $\hat{\pi}^Q = \arg\sup_\pi \inf_{h \in \mathcal{S}(e_n)} \mathbb{E}_{p_{in}^\pi}[M_\phi(A, X; h)]$. For the regularized version, we define $\mathcal{E}_n(h) := \sum_{i=1}^m \mathcal{L}_{i,n}(h_{\alpha_i}) - \sum_{i=1}^m \inf_{h_{\alpha_i} \in \mathcal{H}} \mathcal{L}_{i,n}(h_{\alpha_i})$. The estimated policy is given by $\hat{\pi}_R^Q = \arg\sup_\pi \inf_{h \in \mathcal{H}}\{\mathbb{E}_{p_{in}^\pi}[M_\phi(A, X; h)] + \lambda_n \mathcal{E}_n(h)\}$. We now present a result that characterizes the convergence rates of regret for the algorithms applied in Example D.3.*

**Corollary D.4** *(Convergence of the Regret for Quantile-based Risk Measures). Suppose $\max_{i \in [m]} \phi(\alpha_i) = M$.*

  *(i) Suppose that the assumptions in Theorem 4.11 hold. Specifically, the identifiability Assumption 4.1 and the regularity Assumption 4.3 hold for each $h_{\alpha_i}^*$. The local curvature Assumption 4.9 holds uniformly for all $h_{\alpha_i}^*$ for some constant $c_0$. In addition, there exists a change of measure function for each $\alpha_i$ in Assumption 4.10. Then $Regret(\hat{\pi}^Q) \lesssim \mathcal{O}(Mm \cdot \eta_n)$ with probability $1 - 3m\xi$.*

  *(ii) Suppose that the assumptions in Theorem C.4 hold. Specifically, the identifiability Assumption 4.1 and the regularity Assumption 4.3 hold for each $h_{\alpha_i}^*$. The sample criterion assumption C.2 holds for each $\mathcal{L}_{i,n}(h_{\alpha_i})$. The local curvature Assumption 4.9 holds uniformly for all $h_{\alpha_i}^*$ for some constant $c_0$. In addition, there exists a change of measure function for each $\alpha_i$ in Assumption 4.10. Then $Regret(\hat{\pi}_R^Q) \lesssim \mathcal{O}(Mm \cdot \eta_n^{1-\epsilon_{\lambda_n}})$ with probability $1 - 3m\xi$.*

**Proof** *See §G.7 for a detailed proof.* ∎

*The results presented in Corollary D.4 align with our intuition: selecting a smaller value for $m$ improves the performance on regret. However, given that $m$ dictates the precision of using Equation (D.1) to estimate the risk measure $\int_0^1 w(\alpha) h_\alpha^* d\alpha$, it essentially represents a trade-off between these two aspects.*

# E    CONCENTRATION BOUND

Throughout this section, we let $\phi : \mathcal{X} \times \mathcal{A} \to \mathbb{R}^{H_n}$ and $\psi : \mathcal{X} \times \mathcal{Z} \to \mathbb{R}^{\theta(n)}$ be two feature maps. $H_n$ and $\theta(n)$ represent the dimensions of the embedding spaces, where we assume exponential decay, i.e., $H_n$ and $\theta(n)$ are both $\mathcal{O}(\log n)$. We then define two linear spaces, $\mathcal{H}$ and $\Theta$, as following:

$$\mathcal{H} := \{\mathcal{X} \times \mathcal{A} \to \beta_1^\top \phi(\cdot) : \beta_1 \in \mathbb{R}^{H_n}, \|\beta_1\|_2 \leq C_1, \|\phi\|_{2,\infty} \leq 1\}, \tag{E.1}$$

$$\Theta := \{\mathcal{X} \times \mathcal{Z} \to \beta_2^\top \psi(\cdot) : \beta_2 \in \mathbb{R}^{\theta(n)}, \|\beta_2\|_2 \leq C_2, \|\psi\|_{2,\infty} \leq 1\}. \tag{E.2}$$

The goal of this section is to verify that Condition 4.4 holds with this choice of $\mathcal{H}$ and $\Theta$ with $\eta_n = \widetilde{\mathcal{O}}(n^{-1/2})$. Recall that the event $\mathcal{E}$ in Condition 4.4 is an intersection of two events of the uniform concentration bounds of some function classes. The plan is to first show that the event

$$\left\{ \sup_{w(\cdot)\theta(\cdot) \in \mathcal{Q}} |\mathbb{E}_n[W(D; h)\theta(X, Z)] - \mathbb{E}[W(D; h)\theta(X, Z)]| \leq \eta_n(\|\theta\|_2 + \eta_n) \right\}$$

holds with high probability in Section E.2. Then we will show that the event

$$\left\{ \left| \|\theta\|_{n,2}^2 - \|\theta\|_2^2 \right| \leq \frac{1}{2} \left( \|\theta\|_2^2 + \eta_n^2 \right), \forall \theta \in \Theta \right\}$$

holds with high probability in Section E.3. Finally, we take a union bound of the two events to complete the verification of Condition 4.4.

## E.1    CONCEPTS FROM THE EMPIRICAL PROCESS THEORY

To characterize the concentration bound, we first need to introduce some concepts in the empirical process theory.

**Bracketing Number.** Given two functions $f_1$ and $f_2$ such that $\|f_1 - f_2\| \leq t$, An $t$-bracket $[f_1, f_2]$ is a subset of functions of the real-valued function class $\mathcal{F}$ on $\mathcal{X}$ satisfying for all $f \in [f_1, f_2], \forall x \in \mathcal{X}, f_1(x) \leq f(x) \leq f_2(x)$. The bracketing number $N_{[]}(t; \mathcal{F}, \|\cdot\|_2)$ of a function class $\mathcal{F}$ with respect to the norm $\|\cdot\|_2$ is the smallest number of $t$-brackets needed to cover $\mathcal{F}$.

**Covering Number.** The covering number $N(t; \mathcal{F}, \|\cdot\|)$ of a function class $\mathcal{F}$ with respect to a norm $\|\cdot\|$ is the smallest number of $t$-balls in $\mathcal{F}$ needed to cover $\mathcal{F}$.

**Localized Population Rademacher Complexity.** The localized population Rademacher complexity with respect to $\{X_i\}_{i=1}^n$ and function class $\mathcal{F}$ is defined as

$$\mathcal{R}_n(\eta; \mathcal{F}) := \mathbb{E}_{\{\varepsilon_i\}_{i=1}^n, \{X_i\}_{i=1}^n} \left[ \sup_{f \in \mathcal{F}, \|f\|_2 \leq \eta} \frac{1}{n} \sum_{i=1}^n \varepsilon_i f(X_i) \right].$$

where $\{\varepsilon_i\}_{i=1}^n$ are i.i.d. Rademacher random variables.

**Critical Radius.** Suppose $\mathcal{F}$ is a real-valued function class on $\mathcal{X}$ such that $\forall f \in \mathcal{F}, \|f\|_\infty \leq c$ for some constant $c \geq 0$. The critical radius of $\mathcal{F}$ is the largest possible $\eta$ such that $\mathcal{R}_n(\eta; \mathcal{F}) \leq \eta^2/c$.

### E.2 CONCENTRATION BOUND OF THE FUNCTION CLASS $\mathcal{Q}$ USING BRACKETING NUMBER

Recall that we define the function class $\mathcal{Q}$ as $\mathcal{Q} = \{W(\cdot; h) \times \theta(\cdot) : h \in \mathcal{H}, \theta \in \Theta\}$. In this subsection, we show that the first event of Condition 4.4 holds with high probability.

**Theorem E.1 (Rate of Tail Bound of the Function Class $\mathcal{Q}$.)** *If we choose $\mathcal{H}$ as (E.1) and $\Theta$ as (E.1), then the event $\{|\mathbb{E}_n[W(D; h)\theta(X, Z)] - \mathbb{E}[W(D; h)\theta(X, Z)]| \leq \eta_n(\|\theta\|_2 + \eta_n)\}$ holds with probability $1 - \xi$ for some $\eta_n = \widetilde{\mathcal{O}}(n^{-1/2})$.*

**Proof** We proceed with the help of the following two lemmas.

**Lemma E.2 (Bracketing Number of the Function Class $\mathcal{Q}$)** *For every $t$ small enough, under Assumption 4.7, there exists $C_3 > 0$ such that*

$$\log N_{[]}(t; \mathcal{Q}, \|\cdot\|_2) \leq A_n \log(\frac{C_3}{t}), \tag{E.3}$$

*where $A_n = 4(H_n + \theta(n))$.*

**Proof** *See §F.1 for detailed proof.* ∎

**Lemma E.3 (Lemma 3.11. in Hu et al. (2020))** *Let $\widetilde{\mathcal{Q}}$ be a class of function uniformly bounded by one in $\|\cdot\|_2$ and $q_0$ be a fixed element in $\widetilde{\mathcal{Q}}$. Let $\widetilde{\mathcal{Q}}(t) = \{q \in \widetilde{\mathcal{Q}} : \|q - q_0\|_2) \leq t\}$. Suppose $\log N_{[]}(t; \widetilde{\mathcal{Q}}, \|\cdot\|_2) \leq A_n \log(\frac{C}{t})$ Then there exist universal positive constants $C_4$, $C_5$, $C_6$ and $C_7$ such that for any $\xi > 0$,*

$$\mathbb{P}\left( \sup_{\substack{q \in \widetilde{\mathcal{Q}}; \\ \|q - q_0\|_2 > \sqrt{\frac{A_n}{n}}}} \frac{|(\mathbb{E}_n - \mathbb{E})(q)|}{\sqrt{\frac{A_n}{n}}\|q - q_0\|_2 \log(C\|q - q_0\|_2^{-1})} \leq \frac{C_4}{A_n} \log(C_5/\xi) \right) \geq 1 - \xi/2. \tag{E.4}$$

*Moreover, for $q \in \widetilde{\mathcal{Q}}(\sqrt{\frac{A_n}{n}})$, we have*

$$\mathbb{P}\left( \sup_{q \in \widetilde{\mathcal{Q}}(\sqrt{\frac{A_n}{n}})} |(\mathbb{E}_n - \mathbb{E})(q) \leq \frac{C_6 \log(C_7/\xi)}{A_n \log^2(Cn/A_n)} \frac{A_n}{n} \log(Cn/A_n) \right) \geq 1 - \xi/2. \tag{E.5}$$

*This lemma is a simple variant of Lemma 5.13 in van de Geer (2009).*

By lemma E.2, we can substitute $q_0$ as the zero function and $\widetilde{\mathcal{Q}} = \mathcal{Q}/C_1$ in Lemma E.3. Then the event in (E.4) concerns the supremum over $q \in \mathcal{Q}$ and $\|q\|_2 \geq C_1\sqrt{A_n/n}$. Since when $\|q\|_2 \geq C_1\sqrt{A_n/n}$, $\log(\frac{C_3 C_1}{\|q\|_2}) \leq \log(C_3\sqrt{\frac{n}{A_n}})$, (E.4) becomes:

$$\mathbb{P}\left( \sup_{\substack{q \in \mathcal{Q}; \\ \|q\|_2 > C_1\sqrt{\frac{A_n}{n}}}} \frac{|(\mathbb{E}_n - \mathbb{E})(q)|}{\sqrt{\frac{A_n}{n}}\|q\|_2 \log(C_3\sqrt{\frac{n}{A_n}})} \leq \frac{C_4}{A_n}\log(C_5/\xi) \right) \geq 1 - \xi/2. \tag{E.6}$$

Moreover, the restriction $q \in \widetilde{\mathcal{Q}}(\sqrt{\frac{A_n}{n}})$ of the event in (E.5) is replaced with $q \in \mathcal{Q}(C_1\sqrt{\frac{A_n}{n}})$. We now denote $\eta_n = \max\{\sqrt{\frac{A_n}{n}}\log(C_3\sqrt{\frac{n}{A_n}})\frac{C_4}{A_n}\log(C_5/\xi), [\frac{C_1 A_n}{n}\frac{C_6\log(C_7/\xi)}{A_n\log^2(C_3 n/C_1 A_n)}\log(C_3 n/C_1 A_n)]^{1/2}\}$. Then combine (E.5) and (E.6) and take the union bond, we conclude that

$$\mathbb{P}\left( \sup_{w(\cdot),\theta(\cdot)\in\mathcal{Q}} |\mathbb{E}_n[W(D;h)\theta(X,Z)] - \mathbb{E}[W(D;h)\theta(X,Z)]| \leq \eta_n(\|\theta\|_2 + \eta_n) \right)$$

$$\geq \mathbb{P}\left( \sup_{w(\cdot),\theta(\cdot)\in\mathcal{Q}\left(C_1\sqrt{\frac{A_n}{n}}\right)} |\mathbb{E}_n[W(D;h)\theta(X,Z)] - \mathbb{E}[W(D;h)\theta(X,Z)]| \leq \eta_n^2, \right.$$

$$\left. \sup_{\|w(\cdot)\theta(\cdot)\|_2 \geq C_1\sqrt{\frac{A_n}{n}}} |\mathbb{E}_n[W(D;h)\theta(X,Z)] - \mathbb{E}[W(D;h)\theta(X,Z)]| \leq \eta_n\|\theta\|_2 \right)$$

$$\geq 1 - \xi.$$

We complete the proof by noticing that $\eta_n = \widetilde{\mathcal{O}}(n^{-1/2})$ by choice. ∎

### E.3 CONCENTRATION BOUND USING CRITICAL RADIUS

We now demonstrate that there exists $\eta_n = \widetilde{\mathcal{O}}(n^{-1/2})$ such that the event $\{|\|\theta\|_{n,2}^2 - \|\theta\|_2^2| \leq \frac{1}{2}(\|\theta\|_2^2 + \eta_n^2), \forall\theta \in \Theta\}$ holds with high probability when $\Theta$ is chosen to be the linear space E.2. Subsequently, we can then take the union bound to verify Condition 4.4.

**Lemma E.4** *For any $\xi > 0$, there exists $\eta_n = \widetilde{\mathcal{O}}(n^{-1/2})$ such that the event $\{|\|\theta\|_{n,2}^2 - \|\theta\|_2^2| \leq \frac{1}{2}(\|\theta\|_2^2 + \eta_n^2), \forall\theta \in \Theta\}$ holds with probability $1 - \xi$ when $\Theta$ is chosen to be the linear space (E.2).*

**Proof** *We proceed with the help of two lemmas:*

**Lemma E.5 (Lemma F.4 in Chen et al. (2023))** *The covering numbers $N(t; \mathcal{H}, \|\cdot\|_\infty)$ and $N(t; \Theta, \|\cdot\|_\infty)$ are upper bounded by $\left(1 + \frac{2C_1}{t}\right)^{H_n}$ and $\left(1 + \frac{2C_2}{t}\right)^{\theta(n)}$, respectively.*

**Lemma E.6 (Theorem 14.1 in Wainwright (2019))** *Let $t_n$ be the critical radius of $\Theta$. There exists universal positive constants $k_0$ and $k_1$ such that for any $\eta_n \geq t_n + k_0\sqrt{\log(k_1/\xi)/n}$, we have*

$$|\|\theta\|_n^2 - \|\theta\|_2^2| \leq \frac{1}{2}\|\theta\|_2^2 + \frac{1}{2}\eta_n^2, \quad \forall\theta \in \Theta$$

*with probability at least $1 - \xi$.*

*By Lemma E.5, we can substitute $\Theta$ for $\mathcal{F}$ in F.3. Hence the critical radius of $\Theta$ is bounded by $\mathcal{O}(\sqrt{(\theta(n)\log n)/n}) = \widetilde{\mathcal{O}}(n^{-1/2})$. We then use Lemma E.6 to complete the proof.* ∎

**Theorem E.7** *(Main Theorem of Concentration Bound) Condition 4.4 holds when choosing $\mathcal{H}$ and $\Theta$ as in (E.1) and (E.2) with some $\eta_n = \widetilde{\mathcal{O}}(n^{-1/2})$.*

**Proof** *By Theorem E.1, we have*

$$\mathbb{P}\left(|\mathbb{E}_n\left[W(D;h)\theta(X,Z)\right] - \mathbb{E}\left[W(D;h)\theta(X,Z)\right]| \leq \eta_n\left(\|\theta\|_2 + \eta_n, \forall h \in \mathcal{H}, \forall \theta \in \Theta\right)\right) \geq 1 - \xi.$$

*Combine with Lemma E.4 and take the union bound, and we complete the proof.* ∎

# F SUPPORTING LEMMAS OF §E

In what follows, we introduce and provide proofs for the supporting lemmas concerning the bracketing and covering numbers of certain function classes of interest, as discussed in Section E. These lemmas play a crucial role in verifying Condition 4.4.

## F.1 PROOF OF LEMMA E.2

**Proof** We prove this Lemma with the help of the two Lemmas below.

**Lemma F.1** *The bracketing numbers $N_{[]}(t; \Theta, \|\cdot\|_2)$ is upper bounded by $\left(1 + \frac{4C_1}{t}\right)^{\theta(n)}$.*

**Proof** *Note that $\mathcal{H}$ is indexed by the set $\widetilde{\beta} = \{\beta_2 \in \mathbb{R}^{\theta(n)} : \|\beta_2\|_2 \leq C_2\}$. Let $F(x,z) \equiv 1$. For any $\beta_2^T\psi(\cdot), \widetilde{\beta}_2^T\psi(\cdot) \in \Theta$, i.e., $\beta_2, \widetilde{\beta}_2 \in \widetilde{\beta}$, we have*

$$|\beta_2^T\psi(x,z) - \widetilde{\beta}_2^T\psi(x,z)| \leq \left\|\beta_2 - \widetilde{\beta}_2\right\|_2 \|\psi\|_{2,\infty} F(x,z) \leq \left\|\beta_2 - \widetilde{\beta}_2\right\|_2$$

*almost surely in $(X, Z)$. So by Theorem 2.7.11 in van der vaart & Wellner (2013),*

$$N_{[]}(t; \Theta, \|\cdot\|_2) \leq N(\frac{t}{2}; \widetilde{\beta}, \|\cdot\|_2) \leq N(\frac{t}{2}; \Theta, \|\cdot\|_2) \leq N(\frac{t}{2}; \Theta, \|\cdot\|_\infty) \leq \left(1 + \frac{4C_2}{t}\right)^{\theta(n)}$$

*where the last step leverages Lemma E.5.* ∎

**Lemma F.2 (Theorem 3 & Example 5.1 in Chen et al. (2003))** *Let the indicator function class be*

$$\mathcal{W} := \{W(\cdot; h) : h \in \mathcal{H}\}.$$

*Suppose Assumption 4.7 holds. Then the bracketing number $N_{[]}(t; \mathcal{W}, \|\cdot\|_2)$ is upper bounded by $\left(1 + \frac{8C_1K^2}{t^2}\right)^{H_n}$ for $t$ small enough, where $K > 0$ uniformly upper bounds $\sup_y p_{Y|A,X,Z}(y)$ for almost all $(A, X, Z)$.*

**Proof** *In the proof of Theorem 3 in Chen et al. (2003), for $t$ small enough, with Assumption 4.7:*

$$N_{[]}(t; \mathcal{F}_2, \|\cdot\|_2) \leq N((\frac{t}{2K})^{\frac{1}{s}}; \Theta', \|\cdot\|_\infty)N((\frac{t}{2K})^{\frac{1}{s}}; \mathcal{H}, \|\cdot\|_\infty)$$

*In our case of indicator functions, as also indicated in Example 5.1 in Chen et al. (2003), choose $s = \frac{1}{2}$ and $\mathcal{F}_2 = \mathcal{W}$. Let $\Theta' = \{\alpha\}$ and combine with Lemma E.5, we have*

$$N_{[]}(t, \mathcal{W}, \|\cdot\|_2) \leq 1 \cdot N((\frac{t}{2K})^2; \mathcal{H}, \|\cdot\|_\infty) \leq \left(1 + \frac{8C_1K^2}{t^2}\right)^{H_n}.$$

*Therefore, we complete the proof.* ∎

We prove the upper bound of the function class $\mathcal{Q}$ by combining the upper bounds of the function classes $\Theta$ and $\mathcal{W}$. Consider $t \in (0, 1)$ small enough,

$$
\begin{aligned}
\log N_{[]}(t, \mathcal{Q}, \|\cdot\|_2) &\leq \log N_{[]}(\frac{t}{2}; \mathcal{W}, \|\cdot\|_2) + N_{[]}(\frac{t}{2}; \Theta, \|\cdot\|_2) \\
&\leq H_n \log(1 + \frac{8C_1 K^2}{t^2}) + \theta(n) \log(1 + \frac{4C_2}{t}) \\
&\leq (H_n + \theta(n)) \log\left((1 + \frac{8C_1 K^2}{t^2})(1 + \frac{4C_2}{t^2})\right) \\
&\leq 2(H_n + \theta(n)) \log\left(\frac{1 + \max\{8C_1 K^2, 4C_2\}}{t^2}\right),
\end{aligned}
$$

where the second inequality holds by Lemma F.1 and Lemma F.2. We then let $A_n = 4(H_n + \theta(n))$, $C_3 = \sqrt{1 + \max\{8C_1 K^2, 4C_2\}}$ to complete the proof. ∎

**Lemma F.3 (Lemma F.7 in Chen et al. (2023))** *If the covering number of $C_2$-uniformly bounded function class $\mathcal{F}$ satisfies $\log N(t; \mathcal{F}, \|\cdot\|_\infty) \leq A \log(1 + 2C_2/t)$ for some constant $A$, the critical radius of $\mathcal{F}$ is upper bounded $\mathcal{O}(\sqrt{(A \log n)/n})$.*

# G  PROOF OF THE MAIN RESULTS OF SECTION 4

In this section, we prove the main results in Section 4. The goal is to establish the convergence rates of the regret for the solution set algorithm and the regularized algorithm for the IV case.

## G.1  PROOF OF THEOREM 4.5

Theorem 4.5 closely parallels Theorem 6.4 in Chen et al. (2023), with adaptations made to suit our specific context. Despite the resemblance of the theorem statement and proof techniques, we nevertheless include a detailed proof to accommodate the shift from a linear operator to a nonlinear operator, which results in several changes in how the final results are presented. In particular, with Assumptions 4.1 and 4.2, we prove that $h_\alpha^* \in \mathcal{S}(e_n)$ by deriving an upper bound for $\mathcal{L}_n(h_\alpha^*)$. We then derive the upper bound for $\|\mathcal{T}h\|_2$ for any $h \in \mathcal{S}(e_n)$.

**Theorem 4.5 (i).** In this part, we first compute an upper bound of $\mathcal{L}_n(h_\alpha^*)$. We then deduce that $h_\alpha^* \in \mathcal{S}(e_n)$ if $e_n > \frac{13}{4}\eta_n^2$.

**Proof** It holds on $\mathcal{E}$ that

$$
\begin{aligned}
\mathcal{L}_n(h_\alpha^*) &= \sup_{\theta \in \Theta} \mathbb{E}_n\left[W(D; h_\alpha^*)\theta(Z, X)\right] - \frac{1}{2}\|\theta\|_{n,2}^2 \\
&\leq \sup_{\theta \in \Theta} \Big\{ \left|\mathbb{E}_n\left[W(D; h_\alpha^*)\theta(Z, X)\right] - \mathbb{E}\left[W(D; h_\alpha^*)\theta(Z, X)\right]\right| \\
&\quad + \frac{1}{2}\left|\|\theta\|_{n,2}^2 - \|\theta\|_2^2\right| + \mathbb{E}\left[W(D; h_\alpha^*)\theta(Z, X)\right] - \frac{1}{2}\|\theta\|_2^2 \Big\} \\
&\overset{\mathcal{E}}{\lesssim} \sup_{\theta \in \Theta} \Big\{ \eta_n\left(\|\theta\|_{n,2} + \eta_n\right) + \frac{1}{4}\left(\|\theta\|_2^2 + \eta_n^2\right) \\
&\quad + \mathbb{E}\left[W(D; h_\alpha^*)\theta(Z, X)\right] - \frac{1}{2}\|\theta\|_2^2 \Big\},
\end{aligned}
$$

where the first inequality holds by triangle inequality and the second inequality holds by the definition of $\mathcal{E}$. Let $\mathcal{L}^\lambda(\cdot) = \sup_{\theta \in \Theta} \mathbb{E}\left[W(D; h_\alpha^*)\theta(Z, X)\right] - \lambda\|\theta\|_2^2$. Then $\mathcal{L}_n(h_\alpha^*)$ satisfies

$$
\mathcal{L}_n(h_\alpha^*) \leq \mathcal{L}^{1/8}(h_\alpha^*) - \inf_{\theta \in \Theta}\left(\frac{1}{8}\|\theta\|_2^2 - \eta_n\|\theta\|_2\right) + \frac{5}{4}\eta_n^2, \tag{G.1}
$$

We further let $\theta^\lambda(\cdot; h) = \arg\sup_{\theta \in \Theta} \mathcal{L}^\lambda(h)$. To relate $\mathcal{L}^{1/8}$ to $\mathcal{L}^{1/2}$, we observe that for $\mathcal{L}^{\lambda_1}(h)$ and $\mathcal{L}^{\lambda_2}(h)$ where $0 < \lambda_1 \leq \lambda_2$,

$$
\begin{aligned}
\mathcal{L}^{\lambda_2}(h) &= \sup_{\theta \in \Theta} \mathbb{E}\left[W(D; h)\theta(Z, X)\right] - \lambda_2 ||\theta||_2^2 \\
&= \frac{\lambda_2}{\lambda_1} \cdot \sup_{\theta \in \Theta} \left\{ \mathbb{E}\left[ \frac{\lambda_1}{\lambda_2} W(D; h)\theta(Z, X) \right] - \lambda_1 ||\theta||_2^2 \right\} \\
&\geq \frac{\lambda_2}{\lambda_1} \cdot \left( \mathbb{E}\left[ \frac{\lambda_1}{\lambda_2} W(D; h) \cdot \frac{\lambda_1}{\lambda_2} \theta^{\lambda_1}(Z, X; h) \right] - \lambda_1 \left\| \frac{\lambda_1}{\lambda_2} \theta^{\lambda_1}(Z, X; h) \right\|_2^2 \right) \\
&\geq \frac{\lambda_1}{\lambda_2} \mathcal{L}^{\lambda_1}(h),
\end{aligned}
\tag{G.2}
$$

where the first inequality holds by letting $\theta = \frac{\lambda_1}{\lambda_2}\theta^{\lambda_1}$. Note $\frac{\lambda_1}{\lambda_2}\theta^{\lambda_1} \in \Theta$ as $\lambda_1 \leq \lambda_2$ and $\Theta$ is star-shaped. Now substituting $\lambda_1 = \frac{1}{8}$ and $\lambda_2 = \frac{1}{2}$ and plugging equation G.2 into equation G.1:

$$
\begin{aligned}
\mathcal{L}_n(h_\alpha^*) &\overset{\mathcal{E}}{\lesssim} 4\mathcal{L}^{1/2}(h_\alpha^*) - \inf_{\theta \in \Theta}\left( \frac{1}{8}||\theta||_2^2 - \eta_n ||\theta||_2 \right) + \frac{5}{4}\eta_n^2 \\
&\leq 4\mathcal{L}^{1/2}(h_\alpha^*) + \frac{13}{4}\eta_n^2,
\end{aligned}
\tag{G.3}
$$

where the second inequality holds by computing the minimum value of the quadratic equation of $||\theta||_2^2$. Also note that

$$
\begin{aligned}
\mathcal{L}^{1/2}(h_\alpha^*) &= \sup_{\theta \in \Theta}\left\{ \mathbb{E}\left[W(D; h_\alpha^*)\theta(Z)\right] - \frac{1}{2}||\theta||_2^2 \right\} \\
&\leq \sup_\theta \left\{ \mathbb{E}\left[W(D; h_\alpha^*)\theta(Z)\right] - \frac{1}{2}||\theta||_2^2 \right\} \\
&= \frac{1}{2}\left\| \mathcal{T}h_\alpha^* \right\|_2^2,
\end{aligned}
$$

where the first inequality holds by relaxing the conditions of the event and the second inequality follows by Fenchel duality. Hence G.3 gives

$$
\begin{aligned}
\mathcal{L}_n(h_\alpha^*) &\overset{\mathcal{E}}{\lesssim} 2\left\| \mathcal{T}h_\alpha^* \right\|_2^2 + \frac{13}{4}\eta_n^2 \\
&\leq \frac{13}{4}\eta_n^2,
\end{aligned}
\tag{G.4}
$$

where the third inequality is followed by Assumption 4.1. By the nonnegativity of $\mathcal{L}_n(\cdot)$ [1], it follows that

$$
\mathcal{L}_n(h_\alpha^*) - \inf_{h \in \mathcal{H}} \mathcal{L}_n(h) \overset{\mathcal{E}}{\lesssim} \frac{13}{4}\eta_n^2.
$$

Therefore, by definition of the solution set in equation 3.3, with $e_n > \frac{13}{4}\eta_n^2$, it holds on $\mathcal{E}$ that $h_\alpha^* \in \mathcal{S}(e_n)$. ∎

**Theorem 4.5 (ii).**    In this part, we derive an upper bound for $\left\| \mathcal{T}h \right\|_2^2$ for any $h \in \mathcal{S}(e_n)$.

---

[1]See the remark under Assumption 4.2

**Proof** We first note it holds for all $h \in \mathcal{S}(e_n)$ that

$$
\begin{aligned}
\mathcal{L}_n(h) &= \sup_{\theta \in \Theta} \mathbb{E}_n \left[ W(D;h)\theta(Z,X) \right] - \frac{1}{2} ||\theta||_{n,2}^2 \\
&\geq \sup_{\theta \in \Theta} \Big\{ -\left| \mathbb{E}_n \left[ W(D;h)\theta(Z,X) \right] - \mathbb{E} \left[ W(D;h)\theta(Z,X) \right] \right| \\
&\quad -\frac{1}{2} \left| ||\theta||_{n,2}^2 - ||\theta||_2^2 \right| + \mathbb{E} \left[ W(D;h)\theta(Z,X) \right] - \frac{1}{2} \left\| \theta \right\|_2^2 \Big\} \\
&\overset{\mathcal{E}}{\gtrsim} \sup_{\theta \in \Theta} \Big\{ -\eta_n \left( \left\| \theta \right\|_2 + \eta_n \right) - \frac{1}{4} \left( ||\theta||_2^2 + \eta_n^2 \right) \\
&\quad + \mathbb{E} \left[ W(D;h)\theta(Z,X) \right] - \frac{1}{2} \left\| \theta \right\|_2^2 \Big\}.
\end{aligned}
\tag{G.5}
$$

The first inequality holds by the triangle inequality and the second equality holds by the definition of $\mathcal{E}$. Let $\Theta^+(h) = \{ \theta \in \Theta : \mathbb{E} \left[ W(D;h)\theta(X,Z) > 0 \right\}$. For any $\theta^+ \in \Theta^+(h)$, suppose that $\mathbb{E} \left[ W(D;h)\theta^+(X,Z) \right] = \beta \left\| \theta^+ \right\|_2^2$. By definition of $\theta^+$, we have $\beta > 0$. Let $0 < \kappa \leq 1$. Since $\Theta$ is star-shaped, we have $\kappa\theta^+ \in \Theta$. By plugging in $\kappa\theta^+$ in equation G.5, we have for any $h \in \mathcal{S}(e_n)$ and $\theta^+ \in \Theta^+(h)$ that

$$
\mathcal{L}_n(h) \overset{\mathcal{E}}{\gtrsim} \kappa \left( \beta - \frac{3}{4}\kappa \right) \left\| \theta^+ \right\|_2^2 - \eta_n \kappa \left\| \theta^+ \right\|_2 - \frac{5}{4}\eta_n^2.
\tag{G.6}
$$

Recall the definition of the solution set $\mathcal{S}(e_n)$. For any $h \in \mathcal{S}(e_n)$, it holds that

$$
\begin{aligned}
\mathcal{L}_n(h) &\leq \inf_{h \in \mathcal{H}} \mathcal{L}_n(h) + e_n \\
&\leq \mathcal{L}_n(h_\alpha^*) + e_n \\
&\overset{\mathcal{E}}{\lesssim} \frac{13}{4}\eta_n^2 + e_n,
\end{aligned}
\tag{G.7}
$$

where the second inequality holds by noting that $h_\alpha^* \in \mathcal{H}$ and the last inequality follows from equation G.4. Combine equation G.6 and equation G.7, we get

$$
\kappa \left( \beta - \frac{3}{4}\kappa \right) \left\| \theta^+ \right\|_2^2 - \eta_n \kappa \left\| \theta^+ \right\|_2 - \Delta_n \overset{\mathcal{E}}{\lesssim} 0,
\tag{G.8}
$$

where we define

$$
\Delta_n = \frac{18}{4}\eta_n^2 + e_n.
\tag{G.9}
$$

Note equation G.8 holds for any $0 < \kappa \leq 1$. By setting $\kappa = \min\{1, \beta\}$, we have $\beta - \frac{3}{4}\kappa > 0$. Now we solve the quadratic inequality in equation G.8, and deduce that on event $\mathcal{E}$, for all $h \in \mathcal{S}(e_n)$ and $\theta^+ \in \Theta^+(h)$ that

$$
\left\| \theta^+ \right\|_2 \overset{\mathcal{E}}{\lesssim} \frac{\eta_n \kappa + \sqrt{(\eta_n \kappa)^2 + \kappa \left( 4\beta - 3\kappa \right) \Delta_n}}{\kappa \left( 2\beta - \frac{3}{2}\kappa \right)}.
\tag{G.10}
$$

Considering the following two cases.

**Case (i) when $\beta \geq 1$.** If $\beta \geq 1$, $\kappa = \min\{\beta, 1\} = 1$. On $\mathcal{E}$,

$$
\begin{aligned}
\left\| \theta^+ \right\|_2^2 &\overset{\mathcal{E}}{\lesssim} \left( \frac{\eta_n + \sqrt{\eta_n^2 + (4\beta - 3) \Delta_n}}{(2\beta - \frac{3}{2})} \right)^2 \\
&\leq 4 \left( \eta_n + \sqrt{\eta_n^2 + \Delta_n} \right)^2 \\
&\leq 8 \left( 2\eta_k^2 + \Delta_n \right),
\end{aligned}
\tag{G.11}
$$

where the first inequality holds by equation G.10, the second inequality holds by noting that $\beta = 1$ will maximize the right-hand side. Plugging this result into equation G.8 and rearranging with $\kappa = 1$, we have

$$
\begin{aligned}
\mathbb{E}\left[W(D;h)\theta^+(Z)\right] &\leq \left\|\theta^+\right\|_2^2 \\
&= \frac{3}{4}\left\|\theta^+\right\|_2^2 + \frac{1}{4}\left\|\theta^+\right\|_2^2 \\
&\overset{\mathcal{E}}{\lesssim} \frac{3}{4}\left\|\theta^+\right\|_2^2 + \eta_n\left\|\theta^+\right\|_2 + \Delta_n \\
&\leq 6\left(2\eta_k^2 + \Delta_n\right) + \eta_n\left(4\eta_n + 3\sqrt{\Delta_n}\right) + \Delta_n \\
&\leq 16\eta_n^2 + 7\Delta_n + 3\eta_n\sqrt{\Delta_n} \\
&\leq 20\eta_n^2 + 11\Delta_n,
\end{aligned}
\tag{G.12}
$$

where the first inequality holds by noting that $\kappa(\beta - \frac{3}{4}\kappa) = \frac{1}{4}$ and the third inequality holds by the upper bound of $\left\|\theta^+\right\|_2$ in equation G.11.

**Case (ii) when $\beta < 1$.** If $\beta < 1$, we plug in $\kappa = \beta$. It holds on $\mathcal{E}$ that

$$
\beta\left\|\theta^+\right\|_2 \overset{\mathcal{E}}{\lesssim} 2\left(\eta_n + \sqrt{(\eta_n\kappa)^2 + \Delta_n}\right) \leq 2\left(2\eta_n + \sqrt{\Delta_n}\right),
$$

which suggests that

$$
\mathbb{E}\left[W(D;h)\theta^+(X,Z)\right] = \beta\left\|\theta^+\right\|_2^2 \overset{\mathcal{E}}{\lesssim} 2\left(2\eta_n + \sqrt{\Delta_n}\right)\left\|\theta^+\right\|_2.
\tag{G.13}
$$

**Combination of Case (i) and Case (ii).** Combining equation G.12 and equation G.13, we then have for any $h \in \mathcal{S}(e_n)$ and $\theta_k^+ \in \Theta_k^+(h)$ that

$$
\begin{aligned}
\mathbb{E}\left[W(D;h)\theta^+(X,Z)\right] &\overset{\mathcal{E}}{\lesssim} \max\left\{2\left(\sqrt{\Delta_n} + 2\eta_n\right)\|\theta\|_2, 20\eta_n^2 + 11\Delta_n\right\} \\
&\leq \max\left\{C_n\|\theta\|_2, C_n^2\right\},
\end{aligned}
$$

where $C_n = 5\eta_n + 4\sqrt{\Delta_n}$. We then consider the case when $\mathbb{E}\left[W(D;h)\theta(X,Z)\right] < 0$ for $\theta \in \Theta\backslash\Theta^+(h)$, it follows for any $h \in \mathcal{S}(e_n), \theta \in \Theta$ that

$$
\mathbb{E}\left[W(D;h)\theta(X,Z)\right] \overset{\mathcal{E}}{\lesssim} \max\left\{C_n\|\theta\|_2, C_n^2\right\}.
\tag{G.14}
$$

Now let $\theta_h^*(X,Z) = \arg\min_{\theta\in\Theta}\|\theta - \mathcal{T}h\|_2$. By equation G.14, it then holds for any $h \in \mathcal{S}(e_n)$ that

$$
\mathbb{E}\left[\mathcal{T}h(X,Z)\theta_h^*(X,Z)\right] \overset{\mathcal{E}}{\lesssim} \max\left\{C_n\|\theta_h^*\|_2, C_n^2\right\}.
\tag{G.15}
$$

For the left-hand side of equation G.15, we have

$$
\begin{aligned}
\mathbb{E}\left[\mathcal{T}h(X,Z)\theta_h^*(X,Z)\right] &= \mathbb{E}\left[\mathcal{T}h(X,Z)\left(\theta_h^*(X,Z) - \mathcal{T}h(X,Z) + \mathcal{T}h(X,Z)\right)\right] \\
&\geq \|\mathcal{T}h\|_2^2 - \|\mathcal{T}h\|_2\|\theta_h^* - \mathcal{T}h\|_2 \\
&= \|\mathcal{T}h\|_2^2,
\end{aligned}
\tag{G.16}
$$

where the first inequality holds by the Cauchy-Schwartz inequality and the last equality holds by Assumption 4.2. For the right-hand side of equation G.15, we have,

$$
\begin{aligned}
\max\left\{C_n\|\theta_h^*(X,Z)\|_2, C_n^2\right\} &\leq C_n\left(C_n + \|\theta_h^* - \mathcal{T}h + \mathcal{T}h\|_2\right) \\
&\leq C_n\left(C_n + \|\mathcal{T}h\|_2\right),
\end{aligned}
\tag{G.17}
$$

where the last inequality holds by the triangular inequality and Assumption 4.2. Combining equation G.16 and equation G.17 with equation G.15, for all $h \in \mathcal{S}(e_n)$, we have:

$$
\|\mathcal{T}h\|_2^2 - C_n^2 - C_n\|\mathcal{T}h\|_2 \overset{\mathcal{E}}{\lesssim} 0,
$$

which gives that

$$\|\mathcal{T}h\|_2 \overset{\mathcal{E}}{\lesssim} \frac{1}{2}\left(C_n + \sqrt{C_n^2 + 4C_n^2}\right)$$
$$\leq O(C_n)$$
$$= O(5\eta_n + 4\sqrt{\frac{18}{4}\eta_n^2 + e_n})$$
$$= O\left(\sqrt{e_n}\right) + O\left(\eta_n\right),$$

where the third equality holds by definition of $C_n$ and the definition of $\Delta_n$ in equation G.9. ∎

### G.2 PROOF OF THEOREM 4.8

We prove that, on event $\mathcal{E}$, any $h \in \mathcal{S}(e_n)$ converges in probability to $h_\alpha^*$ with respect to $\|\cdot\|_\infty$ and hence $\|\cdot\|_2$, when $e_n = O(\eta_n^2)$.

**Proof** We first show that it converges in probability in $\|\cdot\|_\infty$ by using two lemmas that investigate the property of the loss function $\mathcal{L}_n$. Lemma G.1 below gives a lower bound of the empirical loss on event $\mathcal{E}$. Lemma G.2 shows that the RMSE $\|\mathcal{T}h\|_2^2$ is continuous on $(\mathcal{H}, \|\cdot\|_\infty)$.

**Lemma G.1 (Lower Bound of the Empirical Loss)** *Suppose that Assumptions 4.3 and 4.2 hold. On the event $\mathcal{E}$, for all $h \in \mathcal{H}$, we have*

$$\mathcal{L}_n(h) = \sup_{\theta \in \Theta} \mathbb{E}_n\left[W(D; h)\theta(X, Z)\right] - \frac{1}{2}\|\theta\|_{n,2}^2 \geq \min\{\frac{\eta_n}{2}\|\mathcal{T}h\|_2\|\mathcal{T}h\|_2^2\} - \mathcal{O}(\eta_n^2).$$

**Proof** See §H.1 for a detailed proof. ∎

**Lemma G.2 (Continuity of the Linear Operator)** *Suppose the Assumption 4.7 holds. $\|\mathcal{T}h\|_2^2$ is continuous on $(\mathcal{H}, \|\cdot\|_\infty)$.*

**Proof** See §H.3 for a detailed proof. ∎

We are ready to prove consistency. For any $\epsilon > 0$, we have

$$\mathbb{P}\left[\|h - h_\alpha^*\|_\infty > \epsilon, h \in \mathcal{S}(e_n)\right]$$
$$\leq \mathbb{P}\left[\inf_{h \in \mathcal{H}, \|h - h_\alpha^*\|_\infty > \epsilon} \mathcal{L}_n(h) \leq \mathcal{O}(\eta_n^2)\right]$$
$$\leq \mathbb{P}\left[\min\left\{\inf_{h \in \mathcal{H}, \|h - h_\alpha^*\|_\infty > \epsilon} \frac{\eta_n}{2}\|\mathcal{T}h\|_2, \inf_{h \in \mathcal{H}, \|h - h_\alpha^*\|_\infty > \epsilon} \|\mathcal{T}h\|_2^2\right\} \leq \mathcal{O}(\eta_n^2)\right],$$

where the second inequality holds by Lemma G.1, and $\mathcal{O}(\cdot)$ means multiplying by a constants. Denote $\varphi_\epsilon := \inf_{h \in \mathcal{H}, \|h - h_\alpha^*\|_\infty > \epsilon} \|\mathcal{T}h\|_2$. Assumption 4.1, 4.3 and Lemma G.2 implies that $\mathcal{H}$ is compact and $\|\mathcal{T}h\|_2$ is continuous on $(\mathcal{H}, \|\cdot\|_\infty)$. Hence $\varphi_\epsilon$ is strictly positive. We then have

$$\mathbb{P}\left[\|h - h_\alpha^*\|_\infty > \epsilon, h \in \mathcal{S}(e_n)\right] \leq \mathbb{P}\left[\min\left\{\frac{\eta_n}{2}\varphi_\epsilon, \varphi_\epsilon^2\right\} \leq \mathcal{O}(\eta_n^2)\right],$$

which converges to zero as $n$ goes to infinity. The result follows from the fact that the supremum norm is stronger than the $L_2$ norm under a finite measure:

$$\mathbb{P}\left[\|h - h_\alpha^*\|_2 > \epsilon, h \in \mathcal{S}(e_n)\right] \leq \mathbb{P}\left[\|h - h_\alpha^*\|_\infty > \epsilon, h \in \mathcal{S}(e_n)\right],$$

which converges to one as $n$ goes to infinity. ∎

### G.3 PROOF OF THEOREM C.3

We prove that if we set $\lambda_n \geq \eta_n^{-(1+\epsilon_{\lambda_n})}$ where $\epsilon_{\lambda_n}$ is an arbitrary positive real number, then on the event $\mathcal{E}$, $\hat{h}_R^\pi$ converges in probability to $h_\alpha^*$ with respect to $\|\cdot\|_\infty$ and hence $\|\cdot\|_2$.

**Proof** We use a similar argument as in the proof of Theorem 4.8. But we first need quantify an upper bound of the empirical loss of the regularized version estimator:

**Lemma G.3 (Upper Bound of the Empirical Loss of Regularized Version Estimator)** *Suppose that Assumption 4.1, Assumption 4.3 and Assumption C.2 hold. If we set $\lambda_n \geq \eta_n^{-(1+\epsilon_{\lambda_n})}$ where $0 < \epsilon_{\lambda_n} < 1$ is arbitrary, then on the event $\mathcal{E}$, $\mathcal{E}_n(\hat{h}_R^\pi) = \mathcal{O}(\eta_n^{1+\epsilon_{\lambda_n}})$ and $\mathcal{L}_n(\hat{h}_R^\pi) = \mathcal{O}(\eta_n^{1+\epsilon_{\lambda_n}})$.*

**Proof** *See §H.2 for a detailed proof.* ∎

Therefore, for any $\epsilon > 0$, we have

$$
\mathbb{P}\left[\left\|\hat{h}_R^\pi - h_\alpha^*\right\|_\infty > \epsilon\right]
$$

$$
\leq \mathbb{P}\left[\inf_{h \in \mathcal{H}, \|h - h_\alpha^*\|_\infty > \epsilon} \mathcal{L}_n(h) \leq \mathcal{O}(\eta_n^{1+\epsilon_{\lambda_n}})\right]
$$

$$
\leq \mathbb{P}\left[\min\left\{\inf_{h \in \mathcal{H}, \|h - h_\alpha^*\|_\infty > \epsilon} \frac{\eta_n}{2}\|\mathcal{T}h\|_2, \inf_{h \in \mathcal{H}, \|h - h_\alpha^*\|_\infty > \epsilon} \|\mathcal{T}h\|_2^2\right\} \leq \mathcal{O}(\eta_n^{1+\epsilon_{\lambda_n}})\right],
$$

where the first inequality holds by Lemma G.3, the second inequality holds by Lemma G.1, and $\mathcal{O}(\cdot)$ omits absolute constants. Denote $\varphi_\epsilon := \inf_{h \in \mathcal{H}, \|h - h_\alpha^*\|_\infty > \epsilon} \|\mathcal{T}h\|_2$. Assumption 4.1, 4.3 and Lemma G.2 implies that $\mathcal{H}$ is compact and $\|\mathcal{T}h\|_2$ is continuous on $(\mathcal{H}, \|\cdot\|_\infty)$. Hence $\varphi_\epsilon$ is strictly positive. We then have

$$
\mathbb{P}\left[\left\|\hat{h}_R^\pi - h_\alpha^*\right\|_\infty > \epsilon\right] \leq \mathbb{P}\left[\min\left\{\frac{\eta_n}{2}\varphi_\epsilon, \varphi_\epsilon^2\right\} \leq \mathcal{O}(\eta_n^{1+\epsilon_{\lambda_n}})\right],
$$

which converges to zero as $n$ goes to infinity. The result follows from the fact that the supremum norm is stronger than the $L_2$ norm under a finite measure:

$$
\mathbb{P}\left[\left\|\hat{h}_R^\pi - h_\alpha^*\right\|_2 > \epsilon\right] \leq \mathbb{P}\left[\left\|\hat{h}_R^\pi - h_\alpha^*\right\|_\infty > \epsilon\right],
$$

which converges to one as $n$ goes to infinity. ∎

### G.4 DECOMPOSITION OF THE REGRET WITH PESSIMISM & PROOF OF COROLLARY 4.6

In this section, we study the regret of the estimated policy $\hat{\pi}$ with pessimism. The result in this section will be utilized in §G.5 for the proof of the convergence rate of the regret. Recall the definition of $\hat{h}^\pi$ and $\hat{\pi}$:

$$
\hat{h}^\pi := \arg\inf_{h \in \mathcal{S}(e_n)} v(h, \pi),
$$

$$
\hat{\pi} := \arg\sup_\pi \inf_{h \in \mathcal{S}(e_n)} v(h, \pi) = \arg\sup_\pi v(\hat{h}^\pi, \pi).
$$

The regret of policy $\hat{\pi}$ is given by

$$
\text{Regret}(\hat{\pi}) = v_\alpha^{\pi^*} - v_\alpha^{\hat{\pi}}
$$

$$
= \underbrace{v_\alpha^{\pi^*} - v(\hat{h}^{\pi^*}, \pi^*)}_{(i)} + \underbrace{v(\hat{h}^{\pi^*}, \pi^*) - v(\hat{h}^{\hat{\pi}}, \hat{\pi})}_{(ii)} + \underbrace{v(\hat{h}^{\hat{\pi}}, \hat{\pi}) - v_\alpha^{\hat{\pi}}}_{(iii)}, \tag{G.18}
$$

Here, (ii) $\leq 0$ holds by optimality of $\hat{\pi}$ and (iii) $\overset{\mathcal{E}}{\lesssim} 0$ holds by definition of $\hat{h}^{\hat{\pi}}$ in equation 4.1 and the fact that $h_\alpha^* \in \mathcal{S}(e_n)$ on event $\mathcal{E}$ by Theorem 4.5. Therefore, we just need to bound (i). We now give the upper bound for (i) §G.5.

## G.5 PROOF OF THEOREM 4.11

We prove the convergence rate of the regret of the solution set algorithm for IV. By Corollary 4.6, we can focus on the estimation error of the average reward function for the optimal interventional policy $\pi^*$, i.e., the term (i) in equation G.18.

**Proof** By definition of the change of measure function (Assumption 4.10), we have:

$$
v_\alpha^{\pi^*} - v(\hat{h}^{\pi^*}, \pi^*) = \mathbb{E}_{p_{\text{in}}^{\pi^*}} \left[ h_\alpha^*(A, X) - \hat{h}^{\pi^*}(A, X) \right]
$$

$$
= \mathbb{E} \left[ (h_\alpha^*(A, X) - \hat{h}^{\pi^*}(A, X)) \mathbb{E} \left[ b(X, Z) p_{Y \mid A, X, Z}(h_\alpha^*(A, X)) \mid A, X \right] \right].
$$

Then by the Tower property,

$$
v_\alpha^{\pi^*} - v(\hat{h}^{\pi^*}, \pi^*) = \mathbb{E} \left[ \mathbb{E} \left[ (h_\alpha^*(A, X) - \hat{h}^{\pi^*}(A, X)) b(X, Z) p_{Y \mid A, X, Z}(h_\alpha^*(A, X)) \mid A, X \right] \right]
$$

$$
= \mathbb{E}[p_{Y \mid A, X, Z}(h_\alpha^*(A, X))(h_\alpha^*(A, X) - \hat{h}^{\pi^*}(A, X)) b(X, Z)]
$$

$$
= \mathbb{E} \left[ \mathbb{E} \left[ p_{Y \mid A, X, Z}(h_\alpha^*(A, X))(h_\alpha^*(A, X) - \hat{h}^{\pi^*}(A, X)) \mid X, Z \right] b(X, Z) \right]
$$

$$
\leq ||\hat{h}^{\pi^*} - h_\alpha^*|| \cdot \|b\|_2,
$$

where the last inequality holds by the Cauchy-Schwartz inequality. Denote $\mathcal{E}_\epsilon := \mathcal{E} \cap \left\{ \hat{h}^{\pi^*} \in \mathcal{H}_\epsilon \right\}$. Thus we can bound the regret as

$$
\mathrm{Regret}(\hat{\pi}) \overset{\mathcal{E}_\epsilon}{\lesssim} ||\hat{h}^{\pi^*} - h_\alpha^*|| \cdot \|b\|_2
$$

$$
\leq c_0 \left\| \mathcal{T} \hat{h}^{\pi^*} \right\|_2 \cdot \|b\|_2
$$

$$
\leq c_0 \|b\|_2 \cdot [\mathcal{O}\left(\sqrt{e_n}\right) + \mathcal{O}\left(\eta_n\right)],
$$

where the first inequality holds by the regret decomposition in §G.4, the second inequality holds by Assumption 4.9, and the last inequality holds by Theorem 4.5. Hence, we complete the proof of Theorem 4.11. ∎

## G.6 PROOF OF THEOREM C.4

In this section, we prove that if the regularized parameter $\lambda_n$ is set to $\lambda_n = \eta_n^{-(1+\epsilon_\lambda)}$, then on the event $\mathcal{E}$ the convergence rate of the regret of the regularized algorithm is of order $\eta_n^{1-\epsilon_{\lambda_n}}$.

**Proof** We first decompose the regret corresponding to $\hat{\pi}_R$ as:

$$
\mathrm{Regret}(\hat{\pi}_R) = v_\alpha^{\pi^*} - v_\alpha^{\hat{\pi}_R}
$$

$$
= \left[ v_\alpha^{\pi^*} + \lambda_n \mathcal{E}_n(h_\alpha^*) \right] - \left[ v(\hat{h}_R^{\hat{\pi}_R}, \hat{\pi}_R) + \lambda_n \mathcal{E}_n(\hat{h}_R^{\hat{\pi}_R}) \right] + \left[ v(\hat{h}_R^{\hat{\pi}_R}, \hat{\pi}_R) + \lambda_n \mathcal{E}_n(\hat{h}_R^{\hat{\pi}_R}) \right]
$$

$$
- \left[ v(h_\alpha^*, \hat{\pi}_R) + \lambda_n \mathcal{E}_n(h_\alpha^*) \right].
$$

By the optimality of $\hat{h}_R^{\hat{\pi}}$, we have

$$
\mathrm{Regret}(\hat{\pi}_R) \leq \left[ v_\alpha^{\pi^*} + \lambda_n \mathcal{E}_n(h_\alpha^*) \right] - \left[ v(\hat{h}_R^{\hat{\pi}_R}, \hat{\pi}_R) + \lambda_n \mathcal{E}_n(\hat{h}_R^{\hat{\pi}_R}) \right]
$$

$$
\leq \left[ v_\alpha^{\pi^*} + \lambda_n \mathcal{E}_n(h_\alpha^*) \right] - \sup_\pi \inf_{h \in \mathcal{H}} \{ v(h, \pi) + \lambda_n \mathcal{E}_n(h) \}
$$

$$
\leq \left[ v_\alpha^{\pi^*} + \lambda_n \mathcal{E}_n(h_\alpha^*) \right] - \left[ v(\hat{h}_R^{\pi^*}, \pi^*) + \lambda_n \mathcal{E}_n(\hat{h}_R^{\pi^*}) \right].
$$

After rearranging the terms, we get

$$
\mathrm{Regret}(\hat{\pi}_R) = \mathbb{E}_{p_{\text{in}}^{\pi^*}} \left[ h_\alpha^*(A, X) - \hat{h}_R^{\pi^*}(A, X) \right] + \lambda_n \mathcal{E}_n(h_\alpha^*) - \lambda_n \mathcal{E}_n(\hat{h}_R^{\pi^*})
$$

$$
\leq \mathbb{E}_{p_{\text{in}}^{\pi^*}} \left[ h_\alpha^*(A, X) - \hat{h}_R^{\pi^*}(A, X) \right] + \lambda_n \mathcal{E}_n(h_\alpha^*) \tag{G.19}
$$

Following the same argument as in Section G.5, we conclude that for the first term,

$$\mathbb{E}_{p_{\text{in}}^{\pi^*}}\left[h_\alpha^*(A, X) - \hat{h}_R^{\pi^*}(A, X)\right] \overset{\mathcal{E}_{R\epsilon}}{\lesssim} c_2 \|b\|_2 \cdot \left\|\mathcal{T}\hat{h}_R^{\pi^*}\right\|_2. \tag{G.20}$$

For the second term, we have by Theorem 4.5 (ii) that

$$\lambda\mathcal{E}_n(h_\alpha^*) \leq \lambda\mathcal{L}_n(h_\alpha^*) \leq \mathcal{O}(\eta_n^{1-\epsilon_\lambda}). \tag{G.21}$$

Moreover, using the fact that $\text{Regret}(\hat{\pi}_R) \geq 0$ and rearranging terms, we have

$$\mathbb{E}_{p_{\text{in}}^{\pi^*}}\left[h_\alpha^*(A, X) - \hat{h}_R^{\pi^*}(A, X)\right] \geq \lambda_n\mathcal{E}_n(\hat{h}_R^{\pi^*}) - \lambda_n\mathcal{E}_n(h_\alpha^*)$$

$$\overset{\mathcal{E}_{R\epsilon}}{\gtrsim} \lambda_n\mathcal{L}_n(\hat{h}_R^{\pi^*}) - \lambda_n\mathcal{O}(\eta_n^2)$$

$$\geq \lambda_n \min\{\frac{\eta_n}{2}\|\mathcal{T}\hat{h}_R^{\pi^*}\|_2, \|\mathcal{T}\hat{h}_R^{\pi^*}\|_2^2\} - \lambda_n\mathcal{O}(\eta_n^2) \tag{G.22}$$

We now show that $\|\mathcal{T}\hat{h}_R^{\pi^*}\|_2 = \mathcal{O}(\eta_n)$. When $\frac{\eta_n}{2}\|\mathcal{T}\hat{h}_R^{\pi^*}\|_2 \geq \|\mathcal{T}\hat{h}_R^{\pi^*}\|_2^2$, the result is immediate. When $\frac{\eta_n}{2}\|\mathcal{T}\hat{h}_R^{\pi^*}\|_2 < \|\mathcal{T}\hat{h}_R^{\pi^*}\|_2^2$, combining equation G.20 and equation G.22, we have

$$c_2 \|b\|_2 \cdot \|\mathcal{T}\hat{h}_R^{\pi^*}\|_2 \overset{\mathcal{E}_{R\epsilon}}{\gtrsim} \lambda_n \cdot \frac{\eta_n}{2}\|\mathcal{T}\hat{h}_R^{\pi^*}\|_2 - \lambda_n\mathcal{O}(\eta_n^2).$$

Solving for $\|\mathcal{T}\hat{h}_R^{\pi^*}\|_2$ and substituting $\lambda_n = \eta_n^{-(1+\epsilon_{\lambda_n})}$ yields

$$\|\mathcal{T}\hat{h}_R^{\pi^*}\|_2 = \mathcal{O}(\eta_n).$$

Substituting this into equation G.20, we know the first term of equation G.19 is upper bounded by $\mathcal{O}(\eta_n)$. By equation G.21, we know the second term of equation G.19 is upper bounded by $\mathcal{O}(\eta_n^{1-\epsilon_\lambda})$. By summing up the two terms, we conclude that

$$\text{Regret}(\hat{\pi}_R) \overset{\mathcal{E}_{R\epsilon}}{\lesssim} \mathcal{O}(\eta_n^{1-\epsilon_{\lambda_n}}).$$

Therefore, we complete the proof. ∎

### G.7 Proof of Corollary D.4

We present the order of the convergence rate of the regret for minimizing the quantile-based risk measures. The idea is to aggregate the individual regret of each $\alpha_i$.

**Proof** Define $\mathcal{E}_i$ as the event

$$\mathcal{E}_i := \Big\{ |\mathbb{E}_n\left[W_i(D; h_{\alpha_i})\theta(X, Z)\right] - \mathbb{E}\left[W_i(D; h_{\alpha_i})\theta(X, Z)\right]| \leq \eta_n\left(\|\theta\|_2 + \eta_n\right),$$

$$\big|\|\theta\|_{n,2}^2 - \|\theta\|_2^2\big| \leq \frac{1}{2}\left(\|\theta\|_2^2 + \eta_n^2\right), \forall h_{\alpha_i} \in \mathcal{H}, \forall \theta \in \Theta \Big\}, \tag{G.23}$$

and then define $\mathcal{E}' = \bigcap_{i=1}^m \mathcal{E}_i$. So $\mathcal{E}'$ holds with probability 1-2m$\xi$. We can upper bound the regret of $\hat{\pi}^Q$ as:

$$\text{Regret}(\hat{\pi}^Q) = \mathbb{E}_{p_{\text{in}}^\pi}\left[\sum_{i=1}^m \phi(\alpha_i)h_{\alpha_i}^*(A, X)\right] - \mathbb{E}_{p_{\text{in}}\hat{\pi}^Q}\left[\sum_{i=1}^m \phi(\alpha_i)h_{\alpha_i}(A, X)\right]$$

$$\overset{\mathcal{E}'}{\lesssim} \mathbb{E}_{p_{\text{in}}^{\pi^*}}\left[\sum_{i=1}^m \phi(\alpha_i)(h_{\alpha_i}^*(A, X) - \hat{h}_{\alpha_i}^{\pi^*}(A, X))\right]$$

$$\leq M \sum_{i=1}^m \mathbb{E}_{p_{\text{in}}^\pi}\left[h_{\alpha_i}^*(A, X) - \hat{h}_{\alpha_i}^{\pi^*}(A, X))\right]$$

For the regularized algorithm, we have a similar upper bound:

$$\text{Regret}(\hat{\pi}_R^Q) \leq \mathbb{E}_{p_{\text{in}}^{\pi^*}} \left[ \sum_{i=1}^{m} \phi(\alpha_i)(h_{\alpha_i}^*(A, X) - \hat{h}_{R\alpha_i}^{\pi^*}(A, X)) \right] + \lambda_n(\mathcal{E}_n(h^*) - \mathcal{E}_n(\hat{h}_R^{\pi^*}))$$

$$\leq M \sum_{i=1}^{m} \mathbb{E}_{p_{\text{in}}^{\pi}} \left[ h_{\alpha_i}^*(A, X) - \hat{h}_{R\alpha_i}^{\pi^*}(A, X) \right] + \lambda_n(\mathcal{E}_n(h^*) - \mathcal{E}_n(\hat{h}_R^{\pi^*}))$$

The rest of the proof now follows from applying the proof in Theorem 4.11 and Theorem C.4 to each $i$. ∎

# H SUPPORTING LEMMAS FOR SECTION 4

In what follows, we present the statement and proofs of the supporting lemmas used in §4.

**Lemma H.1 (Directional Derivative)** *Following Ai & Chen (2012), we introduce the notion of pathwise derivative of $\mathcal{T}h$ in the direction $h - h_\alpha^*$ evaluated at $h_\alpha^*$ as follows:*

$$\frac{d\mathcal{T}h_\alpha^*}{dh}[h - h_\alpha^*] := \frac{d\mathbb{E}[W(D; (1-r)h_\alpha^* + rh) \mid X, Z]}{dr} \Big|_{r=0}.$$

*We then define a pseudo distance between $h$ and $h_\alpha^*$ on $\mathcal{S}(e_n)$ based on the pathwise derivative:*

$$||h - h_\alpha^*|| := \sqrt{\mathbb{E} \left[ \left( \frac{d\mathcal{T}h_\alpha^*}{dh}[h - h_\alpha^*] \right)^2 \right]}. \tag{H.1}$$

*Suppose $\mathcal{H}$ is convex and the regularity density assumption 4.7 hold. For any $h \in \mathcal{H}$,*

$$\frac{d\mathcal{T}h_\alpha^*}{dh}[h - h_\alpha^*] = \mathbb{E}[p_{Y|A,X,Z}(h_\alpha^*(A, X)) \{h(X, A) - h_\alpha^*(A, X)\} \mid X, Z].$$

*Hence $||h - h_\alpha^*|| = \sqrt{\mathbb{E}[(\mathbb{E}[p_{Y|A,X,Z}(h_\alpha^*(A, X)) \{h(X, A) - h_\alpha^*(A, X)\} \mid X, Z])^2]}$.*

**Proof** *Recall that $W(D; h_\alpha^*) = \mathbf{1}\{Y \leq h_\alpha^*(A, X)\} - \alpha$. Assumption 4.7 assures that the conditional density $p_{Y|A,X,Z}$ exists almost surely. Then*

$$\frac{d\mathcal{T}h_\alpha^*}{dh}[h - h_\alpha^*] = \frac{d\mathbb{E}[W(D; (1-r)h_\alpha^*(A, X) + rh(A, X)) \mid X, Z]}{dr} \Big|_{r=0}$$

$$= \frac{d\mathbb{E}[\mathbf{1}\{Y \leq h_\alpha^*(A, X) + r[h(A, X) - h_\alpha^*(A, X)]\} - \alpha \mid X, Z]}{dr} \Big|_{r=0}$$

$$= \frac{d\mathbb{P}(Y \leq h_\alpha^*(A, X)) + r(h(A, X) - h_\alpha^*(A, X) \mid X, Z)}{dr} \Big|_{r=0}$$

*By Assumption 4.9, we can write the conditional probability as the integral of the conditional density $p_{Y|A,X,Z}$:*

$$\frac{d\mathcal{T}h_\alpha^*}{dh}[h - h_\alpha^*] = \frac{d}{dr} \mathbb{E}[\int_{-\infty}^{h_\alpha^*(A,X) + r(h(A,X) - h_\alpha^*(A,X))} p_{Y|A,X,Z}(y)dy \mid X, Z] \Big|_{r=0}$$

$$= \mathbb{E}[p_{Y|A,X,Z}(h_\alpha^*(A, X) + r[h(X, A) - h_\alpha^*(A, X)]) \{h(X, A) - h_\alpha^*(A, X)\} \mid X, Z] \Big|_{r=0}$$

$$= \mathbb{E}[p_{Y|A,X,Z}(h_\alpha^*(A, X)) \{h(A, X) - h_\alpha^*(A, X)\} \mid X, Z]. \tag{H.2}$$

*Hence $||h - h_\alpha^*|| = \sqrt{\mathbb{E}[(\mathbb{E}[p_{Y|A,X,Z}(h_\alpha^*(A, X)) \{h(X, A) - h_\alpha^*(A, X)\} \mid X, Z])^2]}$ by definition of the pseudometric.* ∎

## H.1 PROOF OF LEMMA G.1

**Proof** Recall that on the event $\mathcal{E}$, $|\mathbb{E}_n[W(D;h)\theta(X,Z)] - \mathbb{E}[W(D;h)\theta(X,Z)]| \leq \eta_n(\|\theta\|_2 + \eta_n)$ and $|\|\theta\|_{n,2}^2 - \|\theta\|_2^2| \leq \frac{1}{2}(\|\theta\|_2^2 + \eta_n^2)$. Note we define $\theta_h^*(X,Z)$ as $\arg\min_{\theta \in \Theta} \|\theta - \mathcal{T}h\|_2$. We consider the following two cases:

**Case (i) where** $\|\theta_h^*\|_2 \geq \eta_n$. Let $r = \eta_n/(2\|\theta_h^*\|_2)$. As $r \in [0, 1/2]$ and $\Theta$ is star-shaped, $r\theta_h^* \in \Theta$. Thus

$$\mathcal{L}_n(h) \geq \mathbb{E}_n[W(D;h)r\theta_h^*(X,Z)] - \frac{1}{2}\|r\theta_h^*\|_{n,2}^2$$

On the event $\mathcal{E}$, for the second term on the RHS,

$$\frac{1}{2}\|r\theta_h^*\|_{n,2}^2 \leq \frac{r^2}{2}\left[\frac{3}{2}\|\theta_h^*\|_2^2 + \eta_n^2\right]$$

$$\leq \left[\eta_n^2/(4\|\theta_h^*\|_2^2)\right] \cdot \frac{1}{2}\left[\frac{3}{2}\|\theta_h^*\|_2^2 + \eta_n^2\right]$$

$$\leq \mathcal{O}(\eta_n^2),$$

for the first term regarding the empirical norm of the $\theta_h^*$,

$$\mathbb{E}_n[W(D;h)r\theta_h^*(X,Z)] \geq \mathbb{E}[W(D;h)r\theta_h^*(X,Z)] - \eta_n(\|r\theta_h^*\|_2 + \eta_n)$$

$$\geq r\mathbb{E}[W(D;h)\theta_h^*(X,Z)] - \mathcal{O}(\eta_n^2)$$

$$= r\mathbb{E}[\mathcal{T}h(X,Z)\theta_h^*(X,Z)] - \mathcal{O}(\eta_n^2).$$

By adding and subtracting $\theta_h^*(X,Z)$, we further have

$$\mathbb{E}_n[W(D;h)r\theta_h^*(X,Z)] \geq r\mathbb{E}[(\mathcal{T}h(X,Z) - \theta_h^*(X,Z) + \theta_h^*(X,Z))\theta_h^*(X,Z)] - \mathcal{O}(\eta_n^2)$$

$$\geq \frac{\eta_n}{2}\|\theta_h^*\|_2 - \mathcal{O}(\eta_n^2)$$

$$\geq \frac{\eta_n}{2}\|\mathcal{T}h\|_2 - \mathcal{O}(\eta_n^2),$$

where the second inequality follows by Assumption 4.2. Combining both terms, we have

$$\mathcal{L}_n(h) = \sup_{\theta \in \Theta} \mathbb{E}_n[W(D;h)\theta(X,Z)] - \frac{1}{2}\|\theta\|_{n,2}^2 \geq \frac{\eta_n}{2}\|\mathcal{T}h\|_2 - \mathcal{O}(\eta_n^2).$$

**Case (ii) where** $\|\theta_h^*\|_2 < \eta_n$. We simply choose $r = 1$:

$$\mathcal{L}_n(h) \geq \mathbb{E}_n[W(D;h)\theta_h^*(X,Z)] - \frac{1}{2}\|\theta_h^*\|_{n,2}^2$$

We can upper bound the second term by

$$\frac{1}{2}\|\theta_h^*\|_{n,2}^2 \leq \frac{3}{2}\|\theta_h^*\|_2^2 + \eta_n^2 \leq \mathcal{O}(\eta_n^2),$$

Thus the first term can be lower bounded by

$$\mathbb{E}_n[W(D;h)\theta_h^*(X,Z)] \geq \mathbb{E}[W(D;h)\theta_h^*(X,Z)] - \eta_n(\|\theta_h^*\|_2 + \eta_n)$$

$$\geq \mathbb{E}[W(D;h)\theta_h^*(X,Z)] - \mathcal{O}(\eta_n^2).$$

By add and subtracting $\theta_h^*(X,Z)$, we have

$$= \mathbb{E}[(\mathcal{T}h(X,Z) - \theta_h^*(X,Z) + \theta_h^*(X,Z))\theta_h^*(X,Z)] - \mathcal{O}(\eta_n^2)$$

$$\geq \|\theta_h^*\|_2^2 - \mathcal{O}(\eta_n^2)$$

$$\geq \|\mathcal{T}h\|_2^2 - \mathcal{O}(\eta_n^2).$$

We then complete the proof by combining the above two cases. ∎

## H.2  PROOF OF LEMMA G.3

**Proof** By definition of $\hat{h}_R^\pi$, we have

$$v(\hat{h}_R^\pi, \pi) + \lambda_n \mathcal{E}_n(\hat{h}_R^\pi) = \inf_{h \in \mathcal{H}} \{v(h, \pi) + \lambda_n \mathcal{E}_n(h)\}$$
$$\leq v(h_\alpha^*, \pi) + \lambda_n \mathcal{E}_n(h_\alpha^*).$$

The inequality holds as $h_\alpha^* \in \mathcal{H}$ Assumption 4.1. After rearranging the terms and note that by Theorem 4.5(ii), $\mathcal{E}_n(h_\alpha^*) \leq \mathcal{L}_n(h_\alpha^*) \leq \frac{13}{4}\eta_n^2$, we have

$$\mathcal{E}_n(\hat{h}_R^\pi) \leq \frac{1}{\lambda_n}\left[v(h_\alpha^*, \pi) - v(\hat{h}_R^\pi, \pi)\right] + \mathcal{E}_n(h_\alpha^*)$$
$$\leq \frac{2L_h}{\lambda_n} + \frac{13}{4}\eta_n^2$$
$$\leq 2L_h\eta_n^{1+\epsilon_{\lambda_n}} + \frac{13}{4}\eta_n^2.$$

Then by Assumption C.2, we have $\mathcal{L}_n(\hat{h}_R^\pi) = \mathcal{O}(\eta_n^{1+\epsilon_{\lambda_n}})$. ∎

## H.3  PROOF OF LEMMA G.2

**Proof** Recall the definition of $\mathcal{T}$

$$\mathcal{T}h(X, Z) = \mathbb{E}\left[W(D; h) \mid X, Z\right]$$
$$= \mathbb{E}\left[\mathbf{1}\{Y \leq h(X, A)\} - \alpha \mid X, Z\right]$$
$$= \mathbb{E}\left[\int_{-\infty}^{h(A,X)} p_{Y|A,X,Z}(y)dy \mid X, Z\right] - \alpha.$$

Thus for any $h_1, h_2 \in \mathcal{H}$, by the mean value theorem and the Assumption 4.7 we have

$$|\mathcal{T}h_1 - \mathcal{T}h_2| \leq \mathbb{E}\left[\sup_{t \in [0,1]} p_{Y|A,X,Z}(h_1(X,A) + t\left[h_2(X,A) - h_1(X,A)\right])\left[h_2(X,A) - h_1(X,A)\right] \mid X, Z\right]$$
$$\leq \mathbb{E}\left[\sup_{t \in [0,1]} p_{Y|A,X,Z}(h_1(X,A) + t\left[h_2(X,A) - h_1(X,A)\right]) \mid X, Z\right]\left[\sup_y |h_2(y) - h_1(y)|\right].$$

Then by observing that $\sup_{(X,Z) \in \mathcal{X} \times \mathcal{Z}, h \in \mathcal{H}} |\mathcal{T}h| \leq 1$, we have

$$\|\mathcal{T}h_1\|_2^2 - \|\mathcal{T}h_2\|_2^2 \leq 2\mathbb{E}\left[|\mathcal{T}h_2 - \mathcal{T}h_1|\right]$$
$$\leq 2\mathbb{E}\left[\sup_{t \in [0,1]} p_{Y|A,X,Z}(h_1(X,A) + t\left[h_2(X,A) - h_1(X,A)\right]) \mid X, Z\right]\left[\sup_y |h_2(y) - h_1(y)|\right]$$
$$\leq 2\mathbb{E}\left[\sup_{t \in [0,1]} p_{Y|A,X,Z}(h_1(X,A) + t\left[h_2(X,A) - h_1(X,A)\right]) \mid X, Z\right]\|h_2 - h_1\|_\infty.$$

The Assumption 4.7 now implies the continuity result. ∎

## I  EXPERIMENT DETAILS

In this section, we detail the data-generating process and the implementation of the algorithm used in the simulation experiment discussed in section 5.

**Data Generating Process**

- We set the context $X \sim N(0,1)$ to be a one-dimensional Normal random variable.

- We set the instumental variable $Z \sim N(0,1)$ to be a one-dimensional Normal random variable.

- We set the quantile of interest to be $\alpha = 0.2$.

- We set the error term induced by the unmeasured confounders $U$ to be $\epsilon \sim N(-\Phi^{-1}(\alpha), 1)$, where $\Phi(\cdot)$ is the CDF of a standard Normal distribution.

- To generate the binary action $A$, we first set $t$ to be a variable such that with $p\%$ of the $n$ samples, it is a random noise: $t \sim N(0,1)$. With the other samples, we let $t = X + Z + \epsilon + \gamma \mathbf{1}_{\{X>0\}}$, where we will specify the value of $\gamma$ later.

- We generate $A \sim \text{Bernoulli}(q)$, where $q = \frac{\exp(t)}{1+\exp(t)}$.

- We set the reward to be $Y = X + 3XA + \epsilon$.

It is easy to see that the oracle policy $\pi^*$ is given by $\pi^*(A = 1 \mid X) = \mathbf{1}_{\{X>0\}}$. Hence the value of $\gamma$ determines the level of concentration of the ODCP around the oracle policy. We set $\gamma = 8$, so that around $(100 - p)\%$ of the samples are concentrated around the optimal policy. We choose $p \in \{20, 50, 70\}$ in the experiments.

**Algorithm Implementation** Recall that given the offline dataset, we learn the policy by solving the following optimization problem:

$$\hat{\pi}_R = \arg\sup_{\pi} \inf_{h \in \mathcal{H}} \left\{ v(h,\pi) + \lambda_n \sup_{\theta \in \Theta} \left\{ \mathbb{E}_n\left[W(D;h)\theta(X,Z)\right] - \frac{1}{2}||\theta||^2_{n,2} \right\} \right\}. \qquad (\text{I.1})$$

Since the solution does not admit a closed form, we perform gradient descent when optimizing over $h \in \mathcal{H}$. As $W(D;h)$ is not differentiable, we first smooth it by replacing the indicator function $\mathbf{1}\{Y \le h(A,X)\}$ with $1 + \exp(-5[h(A,X) - Y])$. We denote $W$ the vector in $\mathbb{R}^n$ such that $W_i = 1 + \exp(-5[h(A_i, X_i) - Y_i]) - \alpha$ for a fixed $h$. We choose $\mathcal{H}$ and $\Theta$ to be the linear function classes so that condition 4.4 holds by the results shown in §E. In particular, since $h^*_\alpha(A,X) = X + AX$ by our choice, we can choose the feature map of $\mathcal{H} = \{\mathcal{X} \times \mathcal{A} \to \beta_1^\top \phi(\cdot)\}$ to be $\phi(x,a) = (x, xa)$. Hence the realizability assumption in Assumption 4.1 holds. We also choose the feature map of the test function class $\Theta$ to contain more polynomial features to ensure that the compatibility assumption 4.2 holds. For $\Theta = \{\mathcal{X} \times \mathcal{Z} \to \beta_2^\top \psi(\cdot)\}$, we let $\psi(x,z) = (x, z, xz, x^2, z^2, xz^2, zx^2, x^3, z^3, x^2z^2)$. We denote $\Psi(X,Z)$ the $d$ by $n$ matrix such that its $i$-th column is $\psi(X_i, Z_i)$. As the condition of Theorem C.4 requires that $\lambda_n$ to be roughly $\widetilde{\mathcal{O}}(\eta_n^{-1})$ and that §E shows that $\eta_n = \widetilde{\mathcal{O}}(n^{-1/2})$, we set $\lambda_n = 1.6n^{1/2}$. After a random initialization of $\pi$ and $h \in \mathcal{H}$, we repeatedly carry out the following procedures to update $\pi$:

- By setting the gradient to 0, we derive that $\theta^* := \beta^{*T}\phi(\cdot) \in \Theta$ solves

$$\sup_{\theta \in \Theta} \left\{ \mathbb{E}_n\left[W(D;h)\theta(X,Z)\right] - \frac{1}{2}||\theta||^2_{n,2} \right\},$$

where $\beta^* := \psi(X,Z)(\psi(X,Z)^T\psi(X,Z))^{-1}W$.

- We perform gradient descent on $h \in \mathcal{H}$ to minimize $v(h,\pi) + \lambda_n \left\{ \mathbb{E}_n\left[W(D;h)\theta^*(X,Z)\right] - \frac{1}{2}||\theta^*||^2_{n,2} \right\}$. Let $h^*$ be the $\arg\min$.

- We then update $\pi$ by solving the linear programming problem that maximizes $v(h^*, \pi)$ with respect to $\pi$, subject to the constraint that each of $\pi(X_i, Z_i)$ is a probability distribution over $A$.

## J NEGATIVE CONTROLS

In this section, our focus shifts to the scenario where, instead of the instrumental variables, we observe the negative controls as the side observations in the offline dataset. The structure of this

section is parallel to Section 2 - Section 4. Each subsection within this section is designed to mirror the corresponding sections for IVs in the main text of the paper, providing a parallel exploration and analysis framework. Unless otherwise specified, the notations in this section are consistent with those used previously.

### J.1 ODCP AND INTERVENTIONAL PROCESS FOR NEGATIVE CONTROLS

**Offline Data Collection Process for NCs** Suppose we observe the NCE and the NCO instead of the IV. In the $i$-th sample, the environment generates $(u_i, x_i, e_i, v_i) \sim (\mathcal{U}, \mathcal{X}, \mathcal{E}, \mathcal{V})$ that is jointly distributed as $p(u_i, x_i, e_i, v_i)$. The environment then selects an action following $p(a_i \,|\, u_i, x_i, e_i, v_i)$. Given $(u_i, x_i, e_i, v_i, a_i)$, the reward $y_i$ is generated. The joint distribution of the variables in the observational process is hence given by

$$p(u, x, e, v, a, y) = p(u, x, e, v) \cdot p(a \,|\, u, x, e, v) \cdot p(y \,|\, u, x, e, v, a).$$

The left figure of Figure 3 illustrates a possible causal DAG of the ODCP when the NCE and the NCO are observed. Note that we have $V \perp\!\!\!\perp A \,|\, (U, X)$ and $E \perp\!\!\!\perp (Y, V) \,|\, (A, X, U)$. As the confounders $U$ are unobserved, the offline dataset consists of $\{a_i, x_i, e_i, v_i, y_i\}_{i=1}^n$.

**Interventional Process for NCs** Same as the IV case, the context $X$ follows a different marginal distribution from that in ODCP, denoted by $\widetilde{p}(x)$. We assume $\widetilde{p}(x)$ is known. The action $a$ is chosen by the agent based on the context $x$ and the learned policy $\pi$. Given a policy $\pi$, the joint distribution of random variables in the interventional process is

$$p_{\text{in}}^\pi(u, x, v, e, a, y) = \widetilde{p}(x)p(u, v, e \,|\, x)\pi(a \,|\, x)p(y \,|\, u, x, z, a).$$

The DAG of the interventional process is formed by eliminating any arrow coming into the node of $A$ except the one that comes from the context $X$ in the DAG of ODCP. The right figure of Figure 3 illustrates the possible DAG of the interventional process.

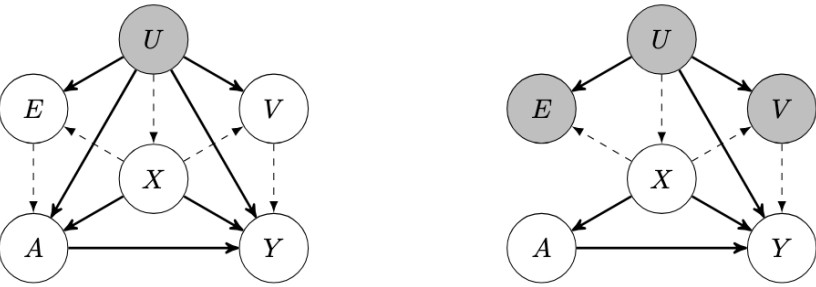

Figure 3: **Left:** A DAG illustrating the causal relationship between random variables of the ODCP when the NCs are observed. The dashed edge implies that the causal relationship may be absent. **Right:** A DAG encoding the causal relationship between random variables in the interventional process. All arrows coming into the node $A$ have been removed other than the one from the node $X$. Note in addition to $U$, $E$ and $V$ are also not observed, indicated by the grey nodes.

**Motivated Example** Consider the scenario of reducing extreme hospital readmission times for patients post-surgery. Here, $Y_i$ represents the post-surgery readmission time for the $i$-th patient. $A_i$ represents the type of post-surgery care provided, such as intensive monitoring or a routine discharge process. $X_i$ is a set of pre-surgery covariates of the patient. $E_i$ could be the patient's location, which is known not to causally affect the readmission time. $V_i$ could be the length of initial hospital stay, which is causally unaffected by the post-surgery care. In this case, the goal of the health system is not just to reduce the average readmission time but to reduce the extreme cases where patients experience the longest delays before being readmitted, as these cases often indicate poor post-surgical outcomes. Thus, the focus is on reducing the higher quantiles of the readmission time distribution.

## J.2 CONDITIONAL MOMENT RESTRICTIONS FOR NEGATIVE CONTROLS

During the ODCP, we now observe the negative control exposure (NCE) denoted as $E$ and the negative control outcome (NCO), represented by $V$. The NCE are variables known to not causally affect the reward $Y$. The NCO are variables known to be causally unaffected by either the action $A$ or the NCE $E$. Formally, the following conditions for $V$ and $E$ hold in ODCP:

**Assumption J.1 (Negative Controls Assumption)** *For the action $A$, unmeasured confounder $U$, and the context $X$,*

 *(i) Latent Unconfoundedness: $V \perp\!\!\!\perp A \,|\, (U, X)$ and $E \perp\!\!\!\perp (Y, V) \,|\, (A, X, U)$;*

 *(ii) Completeness Condition: for any $a \in \mathcal{A}, x \in \mathcal{X}$,*

$$\mathbb{E}\left[\sigma(U, A, X) \,|\, E = e, A = a, X = x\right] = 0$$

 *holds for any $e \in \mathcal{E}$ if and only if $\sigma(U, A, X) \overset{a.s.}{=} 0$. For simplicity, we omit the notation of a.s. in the rest of the paper.*

Here Assumption J.1 (i) formalizes the conditional dependence criteria of the negative controls. Assumption J.1 (ii) is the completeness condition that implies that $E$ captures the variability of the unmeasured confounders $U$. This condition appears extensively in the NC literature (Tchetgen et al., 2020). Note we slightly abuse the notation here: $W(D; h)$ continues to denote the nonlinear functional $\mathbf{1}\{Y \leq h(A, X)\} - \alpha$ while $V$ along refers to NCO. The meaning should be clear from the context. We can now encode the causal structural quantile function into conditional moment restrictions via Assumption J.1.

**Theorem J.2 (Conditional Moment Restrictions for Negative Controls)** *Suppose Assumption J.1 holds. If there exists bridge functions $h_1^* : \mathcal{A} \times \mathcal{X}$ and $h_2^* : \mathcal{V} \times \mathcal{A} \times \mathcal{X} \to \mathbb{R}$ such that,*

$$\mathbb{E}\left[W(D; h_1^*) - h_2^*(V, A, X) \,|\, E = e, A = a, X = x\right] = 0, \tag{J.1}$$
$$\mathbb{E}\left[h_2^*(V, a', X) \,|\, X = x\right] = 0, \tag{J.2}$$

*for any $(e, a, x, a') \in \mathcal{E} \times \mathcal{A} \times \mathcal{X} \times \mathcal{A}$, then it follows that*

$$\mathbb{P}\left[Y \leq h_1^*(X, A) \,|\, X = x, do(A = a)\right] = \alpha.$$

*Therefore, $h_1^*$ is the structural quantile function. We will henceforth refer to $h_1^*$ and $h_\alpha^*$ interchangeably.*

**Proof** *From equation J.1, we have*

$$0 = \mathbb{E}\left[W(D; h_1^*) - h_2^*(V, A, X) \,|\, E = e, A = a, X = x\right]$$
$$= \mathbb{E}\left[\mathbb{E}\left[W(D; h_1^*) - h_2^*(V, A, X) \,|\, U, E, A, X\right] \,|\, E = e, A = a, X = x\right]$$
$$= \mathbb{E}\left[\mathbb{E}\left[W(D; h_1^*) - h_2^*(V, A, X) \,|\, U, A, X\right] \,|\, E = e, A = a, X = x\right],$$

*where the last equality holds by the Assumption J.1 (i). Now by Assumption J.1 (ii), we have*

$$\mathbb{E}\left[W(D; h_1^*) \,|\, U, A, X\right] = \mathbb{E}\left[h_2^*(V, A, X) \,|\, U, A, X\right]. \tag{J.3}$$

*From equation J.2, $\forall a' \in \mathcal{A}, \forall a \in \mathcal{A}$, we have*

$$0 = \mathbb{E}\left[h_2^*(V, a', X) \,|\, X = x\right]$$
$$= \mathbb{E}\left[\mathbb{E}\left[h_2^*(V, a', X) \,|\, U, X\right] \,|\, X = x\right]$$
$$= \mathbb{E}\left[\mathbb{E}\left[h_2^*(V, a', X) \,|\, U, A = a', X\right] \,|\, X = x\right], \tag{J.4}$$

*where the third equality holds by Assumption J.1 (i). Now by combining equation J.3 and equation J.4, we have*

$$0 = \mathbb{E}\left[\mathbb{E}\left[W(D; h_1^*) \,|\, U, A = a, X\right] \,|\, X = x\right]$$
$$= \mathbb{E}\left[W(D; h_1^*) \,|\, do(A = a), X = x\right].$$

*By the definition of $W(D; h_1^*)$, we conclude that*

$$\mathbb{P}\left[Y \leq h_1^* \,|\, X = x, do(A = a)\right] = \alpha.$$

*Therefore, we complete the proof.* ∎

**Data Augmentation** In the conditional moment restriction equation J.2, the equality holds pointwise for any $a' \in \mathcal{A}$. However, solving this equation becomes computationally intractable if either $|\mathcal{A}|$ is large or the action space $\mathcal{A}$ is continuous. To address this issue, we augment the offline dataset by introducing a random variable $A'$ that is distributed uniformly over $\mathcal{A}$ and is independent of $(U, X, E, A, Y)$. Since

$$\mathbb{E}\left[h_2^*(V, a', X) \,|\, X = x\right] = \mathbb{E}\left[h_2^*(V, A', X) \,|\, A' = a', X = x\right],$$

we can replace equation J.2 with

$$\mathbb{E}\left[h_2^*(V, A', X) \,|\, A' = a', X = x\right] = 0. \tag{J.5}$$

In practice, we integrate $A'$ into the offline dataset by appending $a_i' \sim \text{Unif}(\mathcal{A})$ to each sample $(a_i, x_i, e_i, v_i, y_i)$ in ODCP. Consequently, the offline dataset is expanded to $\{a_i, x_i, e_i, v_i, y_i, a_i'\}_{i=1}^n$. We can then estimate $h_2^*$ by solving Equation (J.2) using the augmented offline dataset.

## J.3 Minimax Estimation for Negative Controls

Denote $h = (h_1, h_2)$. Based on equation J.1 and equation J.5, we aim to solve simultaneously for $h$ that satisfies:

$$\mathbb{E}\left[W(D; h_1) - h_2(V, A, X) \,|\, E, A, X\right] = 0 \text{ and} \tag{J.6}$$

$$\mathbb{E}\left[h_2(V, A', X) \,|\, A', X\right] = 0. \tag{J.7}$$

We can define the RMSE on equation J.6 and equation J.7 with respect to $h$ as

$$\|\mathcal{T}h\|_{2,2}^2 := \mathbb{E}\left[\left(\mathcal{T}_1 h(E, A, X)\right)^2\right] + \mathbb{E}\left[\left(\mathcal{T}_2 h(A', X)\right)^2\right], \tag{J.8}$$

where the operators $\mathcal{T}_1 h$ and $\mathcal{T}_2 h$ denote

$$\mathcal{T}_1 h(\cdot) := \mathbb{E}\left[W(D; h_1) - h_2(V, A, X) \,|\, (E, A, X) = \cdot\right],$$

$$\mathcal{T}_2 h(\cdot) := \mathbb{E}\left[h_2(V, A', X) \,|\, (A', X) = \cdot\right].$$

We then choose the hypothesis function space $\mathcal{H} = \mathcal{H}_1 \times \mathcal{H}_2$ and the test function space $\Theta = \Theta_1 \times \Theta_2$. Motivating by the Fenchel duality of the function $x^2/2$, we can then minimize the empirical loss function $\mathcal{L}_n(h) := \mathcal{L}_{1,n}(h) + \mathcal{L}_{2,n}(h)$ where we define $\mathcal{L}_{1,n}$ and $\mathcal{L}_{2,n}$ as

$$\mathcal{L}_{1,n}(h) := \sup_{\theta_1 \in \Theta_1} \left\{\frac{1}{n}\sum_{i=1}^n \{[W(D_i; h_1) - h_2(V_i, A_i, X_i)]\,\theta_1(E_i, A_i, X_i)\} - \frac{1}{2n}\sum_{i=1}^n \theta_1^2(E_i, A_i, X_i)\right\}, \tag{J.9}$$

$$\mathcal{L}_{2,n}(h) := \sup_{\theta_2 \in \Theta_2} \left\{\frac{1}{n}\sum_{i=1}^n [h_2(V_i, A_i', X_i)]\,\theta_2(A_i', X_i) - \frac{1}{2n}\sum_{i=1}^n \theta_2^2(A_i', X_i)\right\}. \tag{J.10}$$

Since both $\mathcal{L}_{1,n}(h)$ and $\mathcal{L}_{2,n}(h)$ can be evaluated from the offline data, we can obtain an estimator of $h_1, h_2$ by minimizing $\mathcal{L}_n(h) = \mathcal{L}_{1,n}(h) + \mathcal{L}_{2,n}(h)$ over $\mathcal{H}$.

## J.4 Algorithm of Policy Learning via Negative Controls

We now present the solution set version algorithm for policy learning via negative controls. We also provide the regularized version of the algorithm in §J.4. The underlying concept of the algorithms closely mirrors that of the instrumental variables. The primary caveat is that we are now simultaneously solving two conditional moment equations, with our focus primarily on $h_1$, as it represents the estimated structural quantile function of interest. Consequently, this necessitates a minor modification to the definition of the solution set. Apart from this adjustment, the foundational ideas remain consistent.

**The Solution Set Algorithm for the Negative Controls** We build the solution set for $h$ based on the empirical loss function $\mathcal{L}_n(h)$ and the threshold $e_n$:

$$\mathcal{S}(e_n) := \left\{h \in \mathcal{H} : \mathcal{L}_n(h) \le \inf_{h \in \mathcal{H}} \mathcal{L}_n(h) + e_n\right\}. \tag{J.11}$$

As we are only interested in $h_1$, we additionally define the projection of $\mathcal{S}(e_n)$ onto its first coordinate:

$$\mathcal{S}_1(e_n) := \{h_1 \in \mathcal{H}_1 : \exists (h_1, h_2) \in \mathcal{S}(e_n)\}.$$

We then select the policy $\pi$ that optimizes the pessimistic average reward function $v(h_1, \pi)$ over the solution set $\mathcal{S}_1(e_n)$. The full algorithm is summarized in Algorithm 2.

---

**Algorithm 2** Quantile action Effect Policy Learning for Negative Controls

---

**Input:** Offline dataset $\{a_i, x_i, e_i, v_i, y_i, a_i'\}_{i=1}^n$ from ODCP, hypothesis space $\mathcal{H}$, test function space $\Theta$, and threshold $e_n$.
    (i) Construct the solution set $\mathcal{S}(e_n)$ as the level set of $\mathcal{H}$ with respect to metric $\mathcal{L}_n(\cdot)$ and threshold $e_n$.
    (ii) $\hat{\pi} = \arg\sup_\pi \inf_{h_1 \in \mathcal{S}_1(e_n)} v(h_1, \pi)$.
**Output:** $\hat{\pi}$.

---

**The Regularized Algorithm for the Negative Controls** Similar to the instrumental variables case, the solution set algorithm faces the computational challenges of solving an optimization problem with data-dependent constraints. We introduce a more practical, regularized version of the algorithm. We denote $\mathcal{E}_n(h) = \mathcal{L}_n(h) - \inf_{h \in \mathcal{H}} \mathcal{L}_n(h)$. The regularized version of the algorithm is summarized in Algorithm 3.

---

**Algorithm 3** The Regularized Policy Learning Algorithm for Negative Controls

---

**Input:** Offline dataset $\{a_i, x_i, e_i, v_i, y_i, a_i'\}_{i=1}^n$, hypothesis space $\mathcal{H}$, test function space $\Theta$, and regularization parameter $\lambda_n$.
    $\hat{\pi}_R = \arg\sup_\pi \inf_{h_1 \in \mathcal{H}_1} \{v(h_1, \pi) + \lambda_n \mathcal{E}_n(h)\}$.
**Output:** $\hat{\pi}_R$.

---

## J.5 Theoretical Results for Policy Learning Algorithm via the Negative Controls

For the negative controls, we are solving two conditional moment equations simultaneously in contrast to solving one equation in the case of instrumental variables. Consequently, the underlying assumptions and theoretical results are analogous to those in Section 4. Thus, in this section, we confine our discussion only to the main assumptions and theorems that directly characterize the rate of convergence of regret of $\hat{\pi}$ and $\hat{\pi}_R$. See §K for a complete derivation of the theoretical results.

A crucial assumption unique to the negative control case is Assumption J.3, which parallels the change of measure Assumption 4.10. However, in this instance, we require the existence of two change-of-measure functions, with each corresponding to one of the two conditional moment equations.

**Assumption J.3 (Change of Measure for Negative Controls)** *For the marginal distribution of context $\widetilde{p}$ in the interventional process and the optimal interventional policy $\pi^*$,*

    *(i). There exists a function $b_1 : \mathcal{E} \times \mathcal{A} \times \mathcal{X} \to \mathbb{R}$ such that $\mathbb{E}\left[b_1^2(E, A, X)\right] < \infty$ and*

$$\mathbb{E}\left[b_1(E, A, X) p_{y \mid E, A, X}(h_\alpha^*(A, X)) \mid A = a, X = x\right] = \frac{\widetilde{p}(x) \pi^*(a \mid x)}{p(x, a)}. \tag{J.12}$$

    *(ii). Define $b_2 : \mathcal{V} \times \mathcal{E} \times \mathcal{A} \times \mathcal{X} \times \mathcal{A} \to \mathbb{R}$ by*

$$b_2(v, e, a, x, a') = \frac{\hat{h}_2^{\pi^*}(v, a, x) p(v \mid e, a, x)}{\hat{h}_2^{\pi^*}(v, a', x) p(v \mid x)}. \tag{J.13}$$

    *We assume $\|b_2\|_\infty < \infty$.*

**Remark on Assumption J.3** Beyond the change of measure condition outlined in Assumption 4.10 for the instrumental variables, which requires the function of $b$ to have a finite $L_2$ norm, Assumption J.3 (ii) incorporates a more stringent requirements. First, since we now have to solve two conditional moment restrictions, we now need two change of measure functions, $b_1$ and $b_2$. Second, the assumption asserts the change of measure function $b_2$ to have a finite $L_\infty$ norm. This is a stricter condition as $L_\infty$ is a stronger norm than $L_2$. Practically, this implies the necessity for the offline dataset to uniformly cover the covariate space $\mathcal{X}$, and the negative control outcome space $W$. In addition, to ensure that $\hat{h}_2^{\pi^*}(v, a', x)$ is bounded away from 0, we can do a location shift on the reward.

**Theorem J.4 (Regret of Solution Set Algorithm for the NC; Informal Version of Theorem K.9)**
*Under appropriate conditions, if the threshold $e_n$ for the solution set is set to $e_n > (2L_h^2 + \frac{5}{4})\eta_n^2$, then with probability $1 - 5\xi$, the regret corresponding to $\hat{\pi}$ is bounded by*

$$Regret(\hat{\pi}) \lesssim c_1(1 + \|b_2\|_\infty) \|b_1\|_2 \cdot \left(\mathcal{O}\left(\sqrt{e_n}\right) + \mathcal{O}\left(\eta_n\right)\right).$$

**Proof** *See Theorem K.9 for the complete statement of the theorem and §L.1 for a detailed proof.* ■

**Theorem J.5 (Regret of Regularized Algorithm for the NC; Informal Version of Theorem K.12)**
*Under appropriate conditions, foy any $0 < \epsilon_{\lambda_n} < 1$, if the regularized parameter $\lambda_n$ is set to $\lambda_n = \eta_n^{-(1+\epsilon_{\lambda_n})}$, then with probability $1 - 5\xi$, the regret corresponding to $\hat{\pi}_R$ is upper bounded by*

$$Regret(\hat{\pi}_R) \lesssim \mathcal{O}(\eta_n^{1-\epsilon_{\lambda_n}}).$$

**Proof** *See Theorem K.12 for the complete statement of the theorem and §L.3 for a detailed proof.* ■

# K  THEORETICAL ANALYSIS FOR ALGORITHM VIA NEGATIVE CONTROLS

The structure of this section is parallel to that of Section 4. The ultimate goal is to link the regret of $\hat{\pi}$ and $\hat{\pi}_R$ to $\|\mathcal{T}h\|_{2,2}^2$. We first construct two events of high probability that bridges the empirical loss $\mathcal{L}_n$ to $\|\mathcal{T}h\|_{2,2}^2$.

$$\widetilde{\mathcal{E}}_1 := \left\{ \left| \mathbb{E}_n \left\{ [W(D; h_1) - h_2(V, A, X)] \theta_1(E, A, X) \right\} - \mathbb{E}\left\{ [W(D; h_1) - h_2(V, A, X)] \theta_1(E, A, X) \right\} \right| \right.$$

$$\left. \leq \eta_n \left( \|\theta_1\|_2 + \eta_n \right), \quad \left| \|\theta_1\|_{n,2}^2 - \|\theta_1\|_2^2 \right| \leq \frac{1}{2} \left( \|\theta_1\|_2^2 + \eta_n^2 \right), \forall h \in \mathcal{H}, \forall \theta_1 \in \Theta_1 \right\}.$$

$$\widetilde{\mathcal{E}}_2 := \left\{ \left| \mathbb{E}_n \left\{ [h_2(V, A', X)] \theta_2(V, A', X) \right\} - \mathbb{E}\left\{ [h_2(V, A', X)] \theta_2(V, A', X) \right\} \right| \right.$$

$$\left. \leq \eta_n \left( \|\theta_2\|_2 + \eta_n \right), \quad \left| \|\theta_2\|_{n,2}^2 - \|\theta_2\|_2^2 \right| \leq \frac{1}{2} \left( \|\theta_2\|_2^2 + \eta_n^2 \right), \forall h \in \mathcal{H}, \forall \theta_2 \in \Theta_2 \right\}.$$

**Assumption K.1 (Regularity of Function Classes for Negative Controls)** *We assume $\mathcal{H}$ is compact with respect to the norm $\|\cdot\|_{\sup}$ and $\Theta$ is star-shaped. We also assume $\sup_{h \in \mathcal{H}} \|h\|_{\sup} \leq L_H$.*

**Condition K.2** *Suppose that Assumption K.1 holds. For any $\xi > 0$, there exists $\eta_n > 0$ that decreases with $n$ such that the event $\widetilde{\mathcal{E}} = \widetilde{\mathcal{E}}_1 \cap \widetilde{\mathcal{E}}_2$ holds with probability at least $1 - 4\xi$.*

Note each $\widetilde{\mathcal{E}}_i$ bridges $\mathcal{T}_i h(\cdot)$ to $\|\mathcal{T}_i h\|_2^2$. Hence we denote $\widetilde{\mathcal{E}} = \widetilde{\mathcal{E}}_1 \cap \widetilde{\mathcal{E}}_2$, so that $\widetilde{\mathcal{E}}$ is an event that controls the difference between the empirical loss function $\mathcal{L}_n(h)$ and $\|\mathcal{T}h\|_{2,2}^2$. If we choose $\mathcal{H}_1, \mathcal{H}_2, \Theta_1$ and $\Theta_2$ to be linear function classes, Condition K.2 holds for some $\eta_n = \widetilde{\mathcal{O}}(n^{-1/2})$. The reason is that by applying a similar strategy as in Appendix E, we can show that each $\widetilde{\mathcal{E}}_i$ holds with probability at least $1 - 2\xi$. Taking the union bound gives us $1 - 4\xi$. We define a norm $\|\cdot\|_{\sup}$ on $\mathcal{H} = \mathcal{H}_1 \times \mathcal{H}_2$ as $\|h\|_{\sup} = \|h_1\|_\infty + \|h_2\|_\infty$.

**Assumption K.3 (Identifiability and Realizability for the Negative Controls)** *Suppose* $(h_1^*, h_2^*) \in \mathcal{H}$. *For any* $h \in \mathcal{H}$ *that satistfies the equations equation J.6 and equation J.7, we have* $\|h - h_1^*\|_{\sup} = 0$.

Assumption K.3 ensures that the structural quantile function is uniquely identified through the conditional moment restrictions. Consequently, we can substitute $h_1^*$ with $h_\alpha^*$ without any ambiguity.

**Assumption K.4 (Compatibility of Test Function Class)** *For any* $h \in \mathcal{H}$, $\inf_{\theta \in \Theta} \|\theta - \mathcal{T}h\|_2 = \epsilon_\Theta$, *and* $\epsilon_\Theta = \widetilde{\mathcal{O}}(n^{-1/2})$.

### K.1 SOLUTION SET ALGORITHM FOR THE NEGATIVE CONTROLS

We introduce an analogue to Theorem 4.5 tailored for negative controls. This demonstrates that within the event $\widetilde{\mathcal{E}}$, the solution set $\mathcal{S}_1(e_n)$ exhibits several favorable properties.

**Theorem K.5 (Uncertainty Quantification for Negative Controls)** *Suppose that Assumptions K.3, K.4 and K.1 hold.*

  (i). *On* $\widetilde{\mathcal{E}}$, $\mathcal{L}_n(h^*) \leq (2L_h^2 + \frac{5}{4})\eta_n^2$ *where* $h^* = (h_\alpha^*, h_2^*)$. *Moreover, if we set* $e_n > (2L_h^2 + \frac{5}{4})\eta_n^2$, *then* $h_\alpha^* \in \mathcal{S}_1(e_n)$.

  (ii). *On* $\widetilde{\mathcal{E}}$, *for all* $h \in \mathcal{S}_1(e_n)$, *we have,*

$$\|\mathcal{T}h\|_2 \overset{\widetilde{\mathcal{E}}}{\lesssim} \mathcal{O}\left(\sqrt{e_n}\right) + \mathcal{O}\left(\eta_n\right).$$

**Proof** *This is identical to the proof of Theorem 4.5, which is in §G.1.* ∎

**Assumption K.6 (Regularity of density for Negative Controls)** *We assume that* $p_{y \mid e,a,x}$ *the conditional density of $Y$ given* $(E, A, X)$ *exists. Moreover,* $p_{y \mid e,a,x}(y)$ *is continuous in* $(y, e, a, x)$ *and* $\sup_y p_{y \mid e,a,x}(y) < \infty$ *for almost all* $(y, e, a, x)$.

**Theorem K.7 (Consistency of the Solution Set Version Algorithm for Negative Controls)** *Suppose that Assumptions K.3, K.4, K.1 and K.6 hold. If we set* $e_n = \mathcal{O}(\eta_n^2,)$ *then on the event* $\widetilde{\mathcal{E}}$, *for any* $h \in \mathcal{S}(e_n)$, $\|h_1 - h_\alpha^*\|_\infty = o_p(1)$ *and hence* $\|h_1 - h_\alpha^*\|_2 = o_p(1)$.

**Proof** *Given Lemma M.3, this is identical to the proof of Theorem 4.8, which is in §G.2.* ∎

Assumption K.6 ensures that the conditional density of the reward $Y$ given $(E, A, X)$ is well-defined and bounded away from infinity. Note we only present the result of consistency of $h_1$ in theorem 4.8. This focus is deliberate, as the upper bound of regret of the negative controls case is the same as that detailed in Corollary 4.6. Thus, the upper bound only concerns with the conditional average difference between $h_1$ and $h_\alpha^*$. Therefore, a local expansion of $h_1$ around $h_\alpha^*$ suffices to serve our purpose.

let $\mathcal{H}_{1\epsilon} := \{h_1 \in \mathcal{H}_1 : \|h_1 - h_\alpha^*\|_2 \leq \epsilon\} \cap \mathcal{S}_1(e_n)$ where $\epsilon$ is a sufficiently small positive number such that $\mathbb{P}(\widetilde{\mathcal{E}} \cap \{\hat{h}_1^{\pi^*} \in \mathcal{H}_{1\epsilon}\}) \geq 1 - 5\xi$. Such $\epsilon$ is guaranteed to exist by Condition K.2 and Theorem K.7.

**Assumption K.8 (Local Curvature of the Solution Set Estimator via Negative Controls)** *If we set* $e_n > (2L_h^2 + \frac{5}{4})\eta_n^2$, *then there exists a finite constant* $c_1 > 0$ *such that*

  (i) $\mathcal{H}_{1\epsilon}$ *is convex and* $\mathbb{E}[W(D; h_1) \mid E, A, X]$ *is continuously pathwise differentiable with respect to* $h_1 \in \mathcal{H}_{1\epsilon}$.

  (ii) *For any* $h_1 \in \mathcal{H}_{1\epsilon}$, $\|h_1 - h_\alpha^*\| \leq c_1 \|\mathbb{E}[W(D; h_1) \mid E, A, X]\|_2$. *Note*

$$\|h_1 - h_\alpha^*\| = \sqrt{\mathbb{E}[(p_{y \mid E,A,X}(h_\alpha^*(A, X)) \{h_1(X, A) - h_\alpha^*(A, X)\})^2]}.$$

**Theorem K.9 (Regret of the Solution Set Version Algorithm for the Negative Controls)**
*Suppose that the negative controls conditions Assumption J.1 holds. Suppose that Assumptions K.3, K.4, and K.1 for function classes $\mathcal{H}$ and $\Theta$ hold. Suppose also that the Assumptions K.8 and J.3 hold. If the threshold $e_n$ for the solution set is set to $e_n > (2L_h^2 + \frac{5}{4})\eta_n^2$, then the regret corresponding to $\hat{\pi}$ is bounded on event $\widetilde{\mathcal{E}} \cap \left\{ \hat{h}_1^{\pi^*} \in \mathcal{H}_{1\epsilon} \right\}$ by*

$$Regret(\hat{\pi}) \lesssim c_1(1 + \|b_2\|_\infty)\|b_1\|_2 \cdot \left( \mathcal{O}\left(\sqrt{e_n}\right) + \mathcal{O}\left(\eta_n\right) \right),$$

**Proof** *See §L.1 for a detailed proof.* ∎

### K.2 REGULARIZED ALGORITHM FOR NEGATIVE CONTROLS

We now detail the theoretical analysis for the regularized algorithm for negative controls. We define $\hat{h}_R^\pi = (\hat{h}_{R1}^\pi, \hat{h}_{R2}^\pi) = \arg\inf_{h \in \mathcal{H}}\{v(h_1, \pi) + \lambda_n \mathcal{E}_n(h)\}$.

**Assumption K.10 (Sample Criterion for Negative Controls)** *We assume* $\inf_{h \in \mathcal{H}} \mathcal{L}_n(h) = \mathcal{O}(\eta_n^2)$.

**Theorem K.11 (Consistency of the Regularized Algorithm for Negative Controls)** *Suppose that Assumptions K.3, K.4, K.1, K.6 and K.10 hold. For any $\epsilon_{\lambda_n} > 0$. For any $\epsilon_{\lambda_n} > 0$, if we set $\lambda_n \geq \eta_n^{-(1+\epsilon_{\lambda_n})}$, then on the event $\widetilde{\mathcal{E}}$, for any $\pi$, $\left\|\hat{h}_{R1}^\pi - h_\alpha^*\right\|_\infty = o_p(1)$ and hence $\left\|\hat{h}_{R1}^\pi - h_\alpha^*\right\|_2 = o_p(1)$.*

**Proof** *See §L.2 for a detailed proof.* ∎

Given the consistency result, we then perform local expansion of $\hat{h}_{R1}^{\pi^*}$ around $h_\alpha^*$. We now define a restricted space of $\mathcal{H}$. For a fixed $\lambda_n \geq \eta_n^{-(1+\epsilon_{\lambda_n})}$, let $\mathcal{H}_{1\epsilon} := \{h_1 \in \mathcal{H}_1 : ||h_1 - h_\alpha^*||_2 \leq \epsilon\}$. We then fix a sufficiently small positive number $\epsilon$ such that the event $\widetilde{\mathcal{E}}_{R\epsilon} = \widetilde{\mathcal{E}} \cap \left\{ \hat{h}_{R1}^{\pi^*} \in \mathcal{H}_{1\epsilon} \right\}$ occurs with probability at least $1 - 5\xi$ by Condition K.2.

**Theorem K.12 (Regret of the Regularized Algorithm for Negative Controls)** *Suppose that the conditions for negative controls Assumption J.1 holds. Suppose that Assumptions K.3, K.4, and K.1 for function classes $\mathcal{H}$ and $\Theta$ hold. Suppose also that the Assumptions K.10, K.8, and J.3 hold. Foy any $0 < \epsilon_{\lambda_n} < 1$, if the regularized parameter $\lambda_n$ is set to $\lambda_n = \eta_n^{-(1+\epsilon_{\lambda_n})}$, then the regret corresponding to $\hat{\pi}_R$ is bounded on event $\widetilde{\mathcal{E}}_{R\epsilon}$ by*

$$Regret(\hat{\pi}_R) \overset{\widetilde{\mathcal{E}}_{R\epsilon}}{\lesssim} \mathcal{O}(\eta_n^{1-\epsilon_{\lambda_n}}).$$

**Proof** *See §L.3 for a detailed proof.* ∎

## L  PROOF OF MAIN RESULTS OF §K

In this section, we give the proofs of the main theorems in section §K. The goal is to establish the convergence rate of the regret for the policy learning algorithms in the case that we observe negative controls in ODCP.

### L.1  PROOF OF THEOREM K.9

In this section, we analyze the convergence rate of the regret of the solution set algorithm for the negative controls.

**Proof** We firs note that the regret decomposition for the case of negative control is exactly the same as that of the instrumental variable. Therefore, we begin by considering the term (i) in equation G.18. By the change of measure Assumption J.3, we have:

$$v_\alpha^{\pi^*} - v(\hat{h}_1^{\pi^*}, \pi^*) = \mathbb{E}_{p_{\mathrm{in}}^{\pi^*}} \left[ h_\alpha^*(A, X) - \hat{h}_1^{\pi^*}(A, X) \right]$$

$$= \mathbb{E}\left[ (h_\alpha^*(A, X) - \hat{h}^{\pi^*}(A, X)_1) \mathbb{E}\left[ b_1(E, A, X) p_{y \mid E, A, X}(h_\alpha^*(A, X)) \mid A, X \right] \right].$$

Then by Tower property,

$$v_\alpha^{\pi^*} - v(\hat{h}_1^{\pi^*}, \pi^*) = \mathbb{E}\left[ \mathbb{E}\left[ (h_\alpha^*(A, X) - \hat{h}_1^{\pi^*}(A, X)) b_1(E, A, X) p_{y \mid E, A, X}(h_\alpha^*(A, X)) \mid A, X \right] \right]$$

$$= \mathbb{E}[p_{y \mid E, A, X}(h_\alpha^*(A, X))(h_\alpha^*(A, X) - \hat{h}_1^{\pi^*}(A, X)) b_1(E, A, X)]$$

$$= \mathbb{E}\left[ \mathbb{E}\left[ p_{y \mid E, A, X}(h_\alpha^*(A, X))(h_\alpha^*(A, X) - \hat{h}_1^{\pi^*}(A, X)) \mid E, A, X \right] b_1(E, A, X) \right].$$

Now let $\widetilde{\mathcal{E}}_\epsilon := \widetilde{\mathcal{E}} \cap \left\{ \hat{h}_1^{\pi^*} \in \mathcal{H}_{1\epsilon} \right\}$. By Cauchy-Schwart and the form of directional derivative in Lemma H.1 (with a slight change of the conditional variables), we have

$$\mathrm{Regret}(\hat{\pi}) \overset{\widetilde{\mathcal{E}}_\epsilon}{\lesssim} \|\hat{h}_1^{\pi^*} - h_\alpha^*\| \cdot \|b_1\|_2$$

$$\leq c_1 \|b_1\|_2 \cdot \left\| \mathbb{E}[W(D; \hat{h}_1^{\pi^*}) \mid E, A, X] \right\|_2, \tag{L.1}$$

where the last inequality holds by the local curvature Assumption K.8. We now obtain an upper bound for $\left\| \mathbb{E}[W(D; \hat{h}_1^{\pi^*}) \mid E, A, X] \right\|_2$:

$$\left\| \mathbb{E}[W(D; \hat{h}^{\pi^*}) \mid E, A, X] \right\|_2 = \left\| \mathcal{T}_1 \hat{h}^{\pi^*} + \mathbb{E}\left[ \hat{h}_2^{\pi^*}(V, A, X) \mid E, A, X \right] \right\|_2$$

$$\leq \|\mathcal{T}_1 \hat{h}^{\pi^*}\|_2 + \left\| \mathbb{E}\left[ \hat{h}_2^{\pi^*}(V, A, X) \mid E, A, X \right] \right\|_2$$

$$= \|\mathcal{T}_1 \hat{h}^{\pi^*}\|_2 + \left\| \int_\mathcal{V} \hat{h}_2^{\pi^*}(v, A, X) p(v \mid E, A, X) \mathrm{d}v \right\|_2.$$

By Assumption J.3 (ii), we have

$$\left\| \mathbb{E}[W(D; \hat{h}^{\pi^*}) \mid E, A, X] \right\|_2 \leq \|\mathcal{T}_1 \hat{h}^{\pi^*}\|_2 + \left\| \int_\mathcal{W} \hat{h}_2^{\pi^*}(v, A', X) p(v \mid X) b_2(v, E, A, X, A') \mathrm{d}v \right\|_2$$

$$\leq \|\mathcal{T}_1 \hat{h}^{\pi^*}\|_2 + \|b_2\|_\infty \left\| \int_\mathcal{V} \hat{h}_2^{\pi^*}(v, A', X) p_{\mathrm{ob}}(v \mid A', X) \mathrm{d}v \right\|_2$$

$$= \|\mathcal{T}_1 \hat{h}^{\pi^*}\|_2 + \|b_2\|_\infty \|\mathcal{T}_2 \hat{h}^{\pi^*}\|_2.$$

Substitute the above result back to equation L.1, we complete the proof:

$$\mathrm{Regret}(\hat{\pi}) \overset{\widetilde{\mathcal{E}}_\epsilon}{\lesssim} c_1 \|b_1\|_2 \cdot \|\mathcal{T}_1 \hat{h}^{\pi^*}\|_2 + c_1 \|b_1\|_2 \cdot \|b_2\|_\infty \|\mathcal{T}_2 \hat{h}^{\pi^*}\|_2$$

$$= c_1(1 + \|b_2\|_\infty) \|b_1\|_2 \cdot \left( \mathcal{O}\left(\sqrt{e_n}\right) + \mathcal{O}\left(\eta_n\right) \right),$$

where the equality holds by Theorem K.5 (ii). ∎

### L.2 Proof of Theorem K.11

We show that our estimated structural quantile function $\hat{h}_{R1}^\pi$ is consistent to $h_\alpha^*$ in $l_2$ norm. The proof is similar to the proof of Theorem C.3, except that the dimension of $\hat{h}_R^\pi$ is now two. We deduce the result by first showing that $\hat{h}_R^\pi = (\hat{h}_{R1}^\pi, \hat{h}_{R2}^\pi)$ is consistent to $h^* = (h_\alpha^*, h_2^*)$ in $\|\cdot\|_{\mathrm{sup}}$.

**Proof** For any $\epsilon > 0$, we have

$$\mathbb{P}\left[ \|\hat{h}_R^\pi - h^*\|_{\mathrm{sup}} > \epsilon \right] \leq \mathbb{P}\left[ \inf_{h \in \mathcal{H}, \|h - h^*\|_{\mathrm{sup}} > \epsilon} \mathcal{L}_{1,n}(h) \leq \mathcal{O}(\eta_n^{1+\epsilon_\lambda}) \right]$$

$$\leq \mathbb{P}\left[ \min\left\{ \inf_{h \in \mathcal{H}, \|h - h^*\|_{\mathrm{sup}} > \epsilon} \frac{\eta_n}{2} \|\mathcal{T}_1 h\|_2 , \inf_{h \in \mathcal{H}, \|h - h^*\|_{\mathrm{sup}} > \epsilon} \|\mathcal{T}_1 h\|_2^2 \right\} \leq \mathcal{O}(\eta_n^{1+\epsilon_\lambda}) \right],$$

Denote $\varphi_\epsilon := \inf_{h \in \mathcal{H}, \|h - h^*\|_\infty > \epsilon} \|\mathcal{T}_1 h\|_2$. Since Lemma M.3 states that $\|\mathcal{T}_1 h\|_2$ is continuous on $(\mathcal{H}, \|\cdot\|_{\sup})$, $\varphi_\epsilon$ is strictly positive. We then have

$$\mathbb{P}\left[\|\hat{h}_R^\pi - h^*\|_{\sup} > \epsilon\right] \leq \mathbb{P}\left[\min\left\{\frac{\eta_n}{2}\varphi_\epsilon, \varphi_\epsilon^2\right\} \leq \mathcal{O}(\eta_n^{1+\epsilon_\lambda})\right],$$

which converges to zero as $n$ goes to infinity. The result follows from the fact that the supremum norm is stronger than the $L_2$ norm under a finite measure:

$$\mathbb{P}\left[\|\hat{h}_R^\pi - h^*\|_2 > \epsilon\right] \leq \mathbb{P}\left[\|\hat{h}_R^\pi - h^*\|_\infty > \epsilon\right],$$

which converges to one as $n$ goes to infinity. Hence, $\|\hat{h}_{R1}^\pi - h_\alpha^*\|_\infty = o_p(1)$ and $\|\hat{h}_{R1}^\pi - h_\alpha^*\|_2 = o_p(1)$. ∎

### L.3 Proof of Theorem K.12

In this section, we analyze the rate of convergence of the regret of the regularized algorithm for negative controls. The proof is similar to the proof of Theorem C.4.

**Proof** We first decompose the regret corresponding to $\hat{\pi}_R$ as:

$$\text{Regret}(\hat{\pi}_R) = v_\alpha^{\pi^*} - v_\alpha^{\hat{\pi}_R}$$

$$= \left[v_\alpha^{\pi^*} + \lambda_n \mathcal{E}_n(h^*)\right] - \left[v(\hat{h}_{R1}^{\hat{\pi}_R}, \hat{\pi}_R) + \lambda_n \mathcal{E}_n(\hat{h}_R^{\hat{\pi}_R})\right] + \left[v(\hat{h}_{R1}^{\hat{\pi}_R}, \hat{\pi}_R) + \lambda_n \mathcal{E}_n(\hat{h}_R^{\hat{\pi}_R})\right]$$

$$- \left[v(h_\alpha^*, \hat{\pi}_R) + \lambda_n \mathcal{E}_n(h^*)\right]$$

By the optimality of $\hat{h}_R^{\hat{\pi}_R}$, we can drop the last two terms. It follows that

$$\text{Regret}(\hat{\pi}_R) \leq \left[v_\alpha^{\pi^*} + \lambda_n \mathcal{E}_n(h^*)\right] - \left[v(\hat{h}_{R1}^{\hat{\pi}_R}, \hat{\pi}_R) + \lambda_n \mathcal{E}_n(\hat{h}_R^{\hat{\pi}_R})\right]$$

$$\leq \left[v_\alpha^{\pi^*} + \lambda_n \mathcal{E}_n(h^*)\right] - \sup_\pi \inf_{h \in \mathcal{H}} \{v(h_1, \pi) + \lambda_n \mathcal{E}_n(h)\}$$

$$\leq \left[v_\alpha^{\pi^*} + \lambda_n \mathcal{E}_n(h^*)\right] - \left[v(\hat{h}_{R1}^{\pi^*}, \pi^*) + \lambda_n \mathcal{E}_n(\hat{h}_R^{\pi^*})\right].$$

. Then by rearranging terms, we have

$$\text{Regret}(\hat{\pi}_R) = \mathbb{E}_{p_{\text{in}}^{\pi^*}}\left[h_\alpha^*(A, X) - \hat{h}_{R1}^{\pi^*}(A, X)\right] + \lambda_n \mathcal{E}_n(h^*) - \lambda_n \mathcal{E}_n(\hat{h}_R^{\pi^*})$$

$$\leq \mathbb{E}_{p_{\text{in}}^{\pi^*}}\left[h_\alpha^*(A, X) - \hat{h}_{R1}^{\pi^*}(A, X)\right] + \lambda_n \mathcal{E}_n(h^*). \tag{L.2}$$

Following the same argument as in the proof of Theorem K.9, we conclude that for the first term,

$$\mathbb{E}_{p_{\text{in}}^{\pi^*}}\left[h_\alpha^*(A, X) - \hat{h}_{R1}^{\pi^*}(A, X)\right] \overset{\widetilde{\mathcal{E}}_{R\epsilon}}{\lesssim} c_1 \|b_1\|_2 \cdot \|\mathcal{T}_1 \hat{h}^{\pi^*}\|_2 + c_1 \|b_1\|_2 \cdot \|b_2\|_\infty \|\mathcal{T}_2 \hat{h}^{\pi^*}\|_2 \tag{L.3}$$

For the second term, we have by Theorem K.5 (ii) that

$$\lambda_n \mathcal{E}_n(h^*) \leq \lambda_n \mathcal{L}_n(h^*) \leq \mathcal{O}(\eta_n^{1-\epsilon_{\lambda_n}}). \tag{L.4}$$

Moreover, using the fact that $\text{Regret}(\hat{\pi}_R) \geq 0$ and rearranging terms, we have

$$\mathbb{E}_{p_{\text{in}}^{\pi^*}}\left[h_\alpha^*(A, X) - \hat{h}_{R1}^{\pi^*}(A, X)\right] \geq \lambda_n \mathcal{E}_n(\hat{h}_R^{\pi^*}) - \lambda_n \mathcal{E}_n(h^*)$$

$$\overset{\widetilde{\mathcal{E}}_\epsilon}{\gtrsim} \lambda_n \mathcal{L}_n(\hat{h}_R^{\pi^*}) - \lambda_n \mathcal{O}(\eta_n^2).$$

We can further lower bound it as

$$\lambda_n \min\{\frac{\eta_n}{2}\|\mathcal{T}_1 \hat{h}_R^{\pi^*}\|_2, \|\mathcal{T}_1 \hat{h}_R^{\pi^*}\}_2^2\| + \lambda_n \min\{\frac{\eta_n}{2}\|\mathcal{T}_2 \hat{h}_R^{\pi^*}\|_2, \|\mathcal{T}_2 \hat{h}_R^{\pi^*}\|_2^2\} - \lambda_n \mathcal{O}(\eta_n^2). \tag{L.5}$$

Then by following a similar step as in the proof of Theorem C.4, it follows that

$$\|\mathcal{T}_1 \hat{h}^{\pi^*}\|_2 + \|\mathcal{T}_2 \hat{h}^{\pi^*}\|_2 = \mathcal{O}(\eta_n).$$

Then combining this result with equation L.2, equation L.3 and equation L.5, we conclude that

$$\text{Regret}(\hat{\pi}_R) \overset{\widetilde{\mathcal{E}}_{R\epsilon}}{\lesssim} \mathcal{O}(\eta_n^{1-\epsilon_{\lambda_n}}).$$

Therefore, we complete the proof. ∎

## M    SUPPORTING LEMMAS FOR SECTION J.5

In what follows, we present the statement and proofs of the supporting lemmas used in §J.5.

**Lemma M.1 (Upper Bound of the Empirical Loss of Negative Controls)** *If we set* $\lambda_n \geq \eta_n^{-(1+\epsilon_{\lambda_n})}$ *where* $0 < \epsilon_{\lambda_n} < 1$ *is arbitrary, then on the event* $\widetilde{\mathcal{E}}$, $\mathcal{E}_n(\hat{h}_R^\pi) = \mathcal{O}(\eta_n^{1+\epsilon_{\lambda_n}})$ *and* $\mathcal{L}_n(\hat{h}_R^\pi) = \mathcal{O}(\eta_n^{1+\epsilon_{\lambda_n}})$.

**Proof** *By definition of* $\hat{h}_R^\pi$, *we have*

$$v(\hat{h}_{R1}^\pi, \pi) + \lambda_n \mathcal{E}_n(\hat{h}_R^\pi) = \inf_{h \in \mathcal{H}}\{v(h_1, \pi) + \lambda_n \mathcal{E}_n(h)\}$$
$$\leq v(h_\alpha^*, \pi) + \lambda_n \mathcal{E}_n(h^*).$$

*After rearranging the terms and note that by Theorem 4.5(ii),* $\mathcal{E}_n(h^*) \leq \mathcal{L}_n(h^*) \leq (2L_h^2 + \frac{5}{4})\eta_n^2$, *we have*

$$\mathcal{E}_n(\hat{h}_R^\pi) \leq \frac{1}{\lambda_n}\left[v(h_\alpha^*, \pi) - v(\hat{h}_{R1}^\pi, \pi)\right] + \mathcal{E}_n(h^*)$$
$$\leq \frac{2L_h}{\lambda_n} + \frac{13}{4}\eta_n^2$$
$$\leq 2L_h \eta_n^{1+\epsilon_{\lambda_n}} + (2L_h^2 + \frac{5}{4})\eta_n^2.$$

*Then by Assumption K.10, we have* $\mathcal{L}_n(\hat{h}_R^\pi) = \mathcal{O}(\eta_n^{1+\epsilon_{\lambda_n}})$. ∎

**Lemma M.2 (Lower Bound of the Empirical Loss for Negative Controls)** *Suppose that Assumptions K.1 and K.4 hold. On the event* $\widetilde{\mathcal{E}}$, *for all* $h \in \mathcal{H}$, $k = 0, 1$ *we have*

$$\mathcal{L}_{k,n}(h) \geq \min\{\frac{\eta_n}{2}\|\mathcal{T}_k h\|_2, \|\mathcal{T}_k h\|_2^2\} - \mathcal{O}(\eta_n^2).$$

**Proof** *This lemma is a direct variation of Lemma G.1. The proofs are almost the same.* ∎

**Lemma M.3 (Continuity of the Operator for Negative Controls)** $\|\mathcal{T}_1 h\|_2^2$ *is continuous on* $(\mathcal{H}, \|\cdot\|_{\sup})$.

**Proof** *Recall that* $\mathcal{T}_1 h := \mathbb{E}\left[W(D; h_1) - h_2(V, A, X) \mid (E, A, X)\right]$, *Thus for any* $h_1 = (h_1^{(1)}, h_1^{(2)}), h_2 = (h_2^{(1)}, h_2^{(2)}) \in \mathcal{H}$, *we have*

$$|\mathcal{T}_1 h_1 - \mathcal{T}_1 h_2| = |\mathbb{E}\left[W(D; h_1^{(1)}) - h_1^{(2)}(V, A, X) \mid (E, A, X)\right] - \mathbb{E}\left[W(D; h_2^{(1)}) - h_2^{(2)}(V, A, X) \mid (E, A, X)\right]|$$
$$\leq |\mathbb{E}\left[W(D; h_1^{(1)}) - W(D; h_2^{(1)}) \mid (E, A, X)\right]|$$
$$+ |\mathbb{E}\left[h_1^{(2)}(V, A, X) - h_2^{(2)}(V, A, X) \mid (E, A, X)\right]|. \tag{M.1}$$

*Also, recall that*

$$\mathbb{E}\left[W(D; h_1) \mid E, A, X\right] = \mathbb{E}\left[\mathbf{1}\{Y \leq h_1(X, A)\} - \alpha \mid E, A, X\right]$$
$$= \int_{-\infty}^{h(A, X)} p(y|E, A, X)dy - \alpha.$$

*Thus by the mean value theorem, we have*

$$\left| \mathbb{E}\left[ W(D; h_1^{(1)}) - W(D; h_2^{(1)}) \,|\, (E, A, X) \right] \right| \leq \sup_{t \in [0,1]} \left\{ \left| p_{Y\,|\,E,A,X}\left( h_1^{(1)}(X, A) + t[h_2^{(1)}(X, A) - h_1^{(1)}(X, A)] \right) \right| \right.$$

$$\left. \cdot |h_2^{(1)}(X, A) - h_1^{(1)}(X, A)| \right\}$$

$$\leq \sup_{t \in [0,1]} \left| p_{Y\,|\,E,A,X}\left( h_1^{(1)}(X, A) + t[h_2^{(1)}(X, A) - h_1^{(1)}(X, A)] \right) \right|$$

$$\cdot \sup_{x,a} |h_1^{(1)}(x, a) - h_2^{(1)}(x, a)|,$$

*which is upper bounded by*

$$\sup_{t \in [0,1]} \left| p_{Y\,|\,E,A,X}\left( h_1^{(1)}(X, A) + t[h_2^{(1)}(X, A) - h_1^{(1)}(X, A)] \right) \right| \cdot ||h_1^{(1)} - h_2^{(1)}||_\infty. \qquad \text{(M.2)}$$

*Then by observing that the* $|\mathcal{T}_1 h_1 + \mathcal{T}_1 h_2|$ *is upper bounded by* $2L_h + 2$*, we have*

$$\|\mathcal{T}h_1\|_2^2 - \|\mathcal{T}h_2\|_2^2 \leq (2L_h + 2)\mathbb{E}\left[ |\mathcal{T}h_1 - \mathcal{T}h_2| \right]$$

$$\leq (2L_h + 2) \sup_{t \in [0,1]} \left| p_{Y\,|\,E,A,X}\left( h_1^{(1)}(X, A) + t[h_2^{(1)}(X, A) - h_2^{(1)}(X, A)] \right) \right| \cdot ||h_1^{(1)} - h_1^{(2)}||_\infty$$

$$+ (2L_h + 2)||h_1^{(2)} - h_2^{(2)}||_\infty,$$

*where the last inequality holds by Equation (M.1) and Equation (M.2). Hence we complete the proof.* ∎

# N    DISCUSSION OF ASSUMPTIONS AND CONDITIONS IN SECTION 4

**A. Assumption 4.1 (Identifiability and Realizability).** This assumption, often referred to as a global identification assumption, is commonly made in nearly all nonparametric quantile IV literature. For instance, see Assumption 3.2(ii) in Chen & Pouzo (2012), Assumption 1 in Gagliardini & Scaillet (2012), and Assumption 1 in Horowitz & Lee (2007).

Regarding sufficient conditions for this assumption to hold, Chernozhukov & Hansen (2005) provides such conditions in the case where the context $X$ is discrete. For more general setups, Chen et al. (2014) discusses conditions for local identification in detail The only study we are aware of that carefully outlines conditions for global identification is Wong (2022). However, Wong's approach relies on a set of strong regularity conditions to establish these results.

**C. Condition 4.4:** We only verify that Condition 4.4 holds true for linear function classes because they already encompass a wide variety of functions, including polynomial splines, B-splines, wavelets, and Fourier series basis functions. These function classes can be represented as linear combinations of basis functions, making them instances of the linear function class. By examining our proof in Appendix E, we rely only on the property that linear function classes have a covering number and bracketing number bounded by $A_n \log(\frac{C_3}{t})$ where $A_n$ is the dimension of the linear function classes. Therefore, we can actually obtain different convergence rates $\eta_n$ under the supremum norm for the chosen function classes. The bracketing and covering numbers of various function classes can be found in different books. For example, van der vaart & Wellner (2013) provides an extensive treatment of covering numbers, and bracketing numbers in their book Weak Convergence and Empirical Processes. They discuss these concepts for a variety of function classes, including: Sobolev spaces, Besov spaces, Holder classes and reproducing kernel Hilbert spaces.

**B. Assumption 4.7 (Regularity of Conditional Density).** This paper assumes that the outcome variable $Y$ is continuous. The primary reason is that we are dealing with a **nonlinear** functional $\mathcal{T}h$. To link the regret to $\mathcal{T}h$, we conduct a local expansion of $\mathcal{T}h$ around the neighborhood of the structural quantile function $h_\alpha^*$. For this expansion, it is essential that the density of $Y$ is continuous and bounded, ensuring that the pathwise derivative $\mathcal{T}h$ at $h = h_\alpha^*$ exists. We remark that this assumption is needed in most of the nonparametric quantile IV literature. For instance, see Condition 6.1(i), (iv) in Chen & Pouzo (2012), Theorem 3.1 in Abadie et al. (2002), Condition A.1 in Chernozhukov & Hansen (2005), Assumption A.1 in Gagliardini & Scaillet (2012). While our paper does

not address discrete outcomes due to this constraint, we believe a similar algorithm could be applied in this case by first convolving the discrete outcome with Gaussian noise, which would smooth the distribution of $Y$. This smoothing allows for the use of techniques designed for continuous outcomes.

**D. Assumption 4.9 (Local Curvature for the Estimator of the Solution Set Algorithm).** In the initial version of our submission, this assumption was divided into two components. Upon reexamining the proof, we made slight modifications that allowed us to eliminate the need for Assumption 4.9 (i). As a result, only Assumption 4.9 (ii) is required, which asserts that the weaker pseudo metric $||h - h_\alpha^*||$ is Lipschitz continuous with respect to the population criterion function $||\mathcal{T}h||_2$ within a small neighborhood around $h_\alpha^*$. This condition is relatively mild and is commonly employed in the nonlinear functional analysis literature (Chen & Pouzo, 2012; Miao et al., 2023). Condition 6.3 (i) in Chen & Pouzo (2012) provides a sufficient condition for the assumption to hold in a broad class of functional spaces $\mathcal{H}$, including the linear function spaces we consider in Appendix E.

**E. Assumption 4.10 (Change of Measure).** The Change of Measure Assumption 4.10 is actually a weaker substitute of the standard concentrability assumption commonly used in the RL literature. In the tabular setting, Assumption 4.10 is satisfied when the right-hand side of the assumption is uniformly bounded, and the Moore-Penrose inverse of the probability mass matrix $P(Z \mid A, X)^+$ exists. This condition holds if the instrumental variable (IV) satisfies the standard completeness assumption: $\text{rank}[P(Z \mid A, X)] \geq |\mathcal{A}| \times |\mathcal{X}|$. We note that Theorem 4.11 and Theorem 4.13 remain valid if Assumption 4.10 is replaced with the usual single-policy concentrability assumption: there exists of a constant $\widetilde{c} > 0$ such that $\sup_{x \in \mathcal{X}, a \in \mathcal{A}} \frac{\widetilde{p}(x)\pi^*(a \mid x)}{p(x,a)} \leq \widetilde{c}$. Note $\frac{\widetilde{p}(x)\pi^*(a \mid x)}{p(x,a)}$ represents the density ratio of the distribution under the interventional process with the oracle policy $\pi^*$ over that of the offline data collection process.

