# OpenReview forum: "Quantile-Optimal Policy Learning under Unmeasured Confounding"
_ICLR.cc/2025/Conference — ICLR 2025 Conference Withdrawn Submission_

### Official Review · Reviewer_UmP4 · 2024-10-29

**Soundness:** 3
**Presentation:** 2
**Contribution:** 2
**Rating:** 5
**Confidence:** 3

**Summary:**

This work examines quantile-optimal policy learning, focusing on the whole distribution of rewards rather than just the mean. The authors consider a complex setting, addressing challenges such as the nonlinearity of the quantile objective, issues with unobserved confounding, and limited coverage in the offline dataset. They propose using causal inference tools, including instrumental variables (IVs), and adopt a minimax estimation approach with nonparametric models. Computationally, they provide two algorithmic solutions: the solution set algorithm and a regularized version, with the latter offering computational advantages. Theoretically, they analyze the regret rates of the policies derived from both algorithms. Lastly, they conduct a straightforward simulation to assess the performance of the regularized algorithm.

**Strengths:**

1. This work considers reward distribution and unobserved confounding, a challenging yet highly practical setting, especially as unobserved confounding is prevalent in real applications.
2. The algorithms leverage the Pessimism Principle within minimax estimation to address issues arising from insufficient sample size. Furthermore, the authors introduce a regularized version to handle computational difficulty.
3. Theoretical results are substantial. The authors discuss both IV and NC scenarios in detail and provide in-depth regret analyses for the policies obtained from the two algorithms. The theoretical derivation is comprehensive.

**Weaknesses:**

1. The motivation for studying quantiles in policy learning is not clear. The authors use an example of income in a job training program to illustrate the relevance of analyzing reward distribution, which has merit but remains insufficient. In particular, the related works section lacks any mention of quantile policy learning literature. Although work in this area is limited, introducing relevant studies on quantile treatment regimes and quantile treatment effects would help clarify the significance of studying quantiles in policy learning.
2. The experiments lack persuasiveness.
- The authors should compare quantile policy learning with standard policy learning. For instance, using job training program example to highlight the benefits of studying the median rather than the mean would strongly emphasize the advantages of the proposed method, yet this is regrettably absent.
- The authors only evaluate the performance of the regularized algorithm through simulations without comparing it to the solution set algorithm, making it difficult to illustrate the computational advantage of the regularized version. While the solution set algorithm may be computationally intractable, its performance could be evaluated in a simplified experiment.
- The simulated DGP is overly simple, with one-dimensional contexts and IVs, and only a single quantile (0.2) selected, which is insufficiently compelling.
3. Readability is lacking. The theoretical analysis in Section 4 is overly detailed, occupying excessive space. The authors could condense Section 4, such as simplifying the paragraph following Condition 4.4, and instead elaborate on experimental results in Section 5 to improve readability. Additionally, symbols and indices are overly complex, and further simplification is advisable.
4. Numerous typos and errors exist in the writing. Here are a few examples:
- Line 061: "we can'' should be "which can''
- Line 066: "our methods works" should be "our method works''
- Line 087: "he'' should be "the''
- Line 112: The notation $O\in\mathcal O$ appears inconsistently, as in Line 361, $\mathcal O (\eta_n)$
- Line 139: "$\pi$'' should be "$\hat{\pi}$"
- Section 3.4 uses ":='', whereas Section 3.5 uses "=''
- Line 294: "the challenge two'' should be "challenge (2)''
- Line 312: Consider changing "function class $\mathbb{F}$'' to "$\mathcal{F}$'' for notational consistency
- Different notations for order: $\mathcal O$ and $\tilde{\mathcal O}$
- Certain theorems lack periods, such as Corollary 4.6
- Line 446: "$l_2$ norm'' should be "$\ell_2$ norm''
- Line 506: The values of $p$ are 20, 50, 70, while in Appendix I (Line 1699), they are 30, 50, 80.

The authors should take a thorough proofreading of the manuscript.

5. Theoretical results in Section 4 rely on the strong Condition 4.4. While the authors mention in Appendix E that Condition 4.4 is satisfied when $\eta_n = \mathcal O(n^{-1/2})$ and both $\mathcal{H}$ and $\Theta$ are linear spaces. However, they do not discuss Condition 4.4 under more complex cases, nor do they consider whether Condition 4.4 can hold without the assumption of $\mathcal{H}$ and $\Theta$ as linear spaces.

**Questions:**

Could the authors provide a more detailed motivation for studying quantiles, like in which situations quantile policy learning offers a clear advantage over standard policy learning (mean)? Additionally, the authors could supplement the experiment part to highlight the advantages of QPL over standard policy learning.

---

> ### Author Response · Authors · 2024-11-25
> **Response to Reviewer UmP4**
>
> > **Weakness.** The motivation for studying quantiles in policy learning is not clear. The authors use an example of income in a job training program to illustrate the relevance of analyzing reward distribution, which has merit but remains insufficient. In particular, the related works section lacks any mention of quantile policy learning literature. Although work in this area is limited, introducing relevant studies on quantile treatment regimes and quantile treatment effects would help clarify the significance of studying quantiles in policy learning.
>
> **Response.** For additional motivating examples, please refer to our first response to Reviewer K74W. We also provide a motivating example involving hospital readmission times for negative controls in Appendix Section J.1 (p. 33). We appreciate the reviewer’s suggestion to enhance the literature review. In the revised manuscript, we have expanded the related works section to include the literature on quantile policy learning.
>
> ---
>
> > **Weakness.** The experiments lack persuasiveness.
>
> **Response.** For experiments under more complex settings and applications on real-world datasets, see Global Response III point 1).
>
> ---
>
> > **Weakness.** Readability is lacking. The theoretical analysis in Section 4 is overly detailed, occupying excessive space. The authors could condense Section 4, such as simplifying the paragraph following Condition 4.4, and instead elaborate on experimental results in Section 5 to improve readability. Additionally, symbols and indices are overly complex, and further simplification is advisable.
>
> **Response.** We sincerely apologize for any inconvenience caused while reading our paper. For details on how we plan to improve its readability, please refer to Global Response III, point 2).
>
> ---
>
> > **Weakness.** Numerous typos and errors exist in the writing...
>
> **Response.** We thank the reviewer for highlighting the typos and errors. We have thoroughly proofread the revised manuscript to address these issues.
>
> ---
>
> > **Weakness.** Theoretical results in Section 4 rely on the strong Condition 4.4. While the authors mention in Appendix E that Condition 4.4 is satisfied when $\eta_n = \mathcal{O}(n^{-1/2})$ and both $\mathcal{H}$ and $\Theta$ are linear spaces. However, they do not discuss Condition 4.4 under more complex cases, nor do they consider whether Condition 4.4 can hold without the assumption of $\mathcal{H}$ and $\Theta$ as linear spaces.
>
> **Response.** For a detailed discussion of Condition 4.4, please refer to Global Response II, point C.
>
> ---
>
> > **Question.** Could the authors provide a more detailed motivation for studying quantiles, like in which situations quantile policy learning offers a clear advantage over standard policy learning (mean)? Additionally, the authors could supplement the experiment part to highlight the advantages of QPL over standard policy learning.
>
> **Response.** As stated in the previous point, for additional examples, please refer to our first response to Reviewer K74W. We also provide a motivating example involving hospital readmission times for negative controls in Appendix Section J.1 (p. 33). For experiments under more complex settings and applications on real-world datasets, see Global Response point 2).

---

> > ### Comment · Reviewer_UmP4 · 2024-12-02
> > **Official Comment**
> >
> > Thank you for submitting your rebuttal. I have carefully reviewed your response, and I find the clarification regarding the assumptions in the theoretical section both necessary and convincing.
> >
> > However, the overall readability of the manuscript remains somewhat challenging, which could affect its accessibility to a broader audience. Additionally, the experimental section has not been updated, and without new results, I am unable to revise my initial assessment of this part of the work. Therefore, I will maintain my score.

---

### Official Review · Reviewer_Uxa7 · 2024-11-02

**Soundness:** 2
**Presentation:** 1
**Contribution:** 2
**Rating:** 5
**Confidence:** 3

**Summary:**

This paper investigates offline learning with partial observability under a quantile-based objective. The proposed algorithm uses minimax optimization and incorporates the principle of pessimism. Under various assumptions (including identifiability, compatibility of function approximation, coverage of the dataset, etc.), convergence rate of the proposed algorithm is provided.

**Strengths:**

It is interesting to see how the tricks from the theory of offline RL also apply in this setting.

**Weaknesses:**

**Readability of this paper.** The paper is difficult to read.

1. The notations are horrible. Letter W is both NCO variable and a function of (D,h). $\mathcal{O}$ stands for both observation space and big-O notation.

2. The bounds hide most dependency on the parameters, which makes it difficult how assumptions influence the results. For example, the dependency hidden in $o_p(\cdot)$ is never properly specified.

3. The assumptions are stated messily and lack sufficient explanation. For example: (1) It is not clear to me why the identifiability condition of Assumption 4.1 is reasonable, as E[W(D,h)|X,Z] takes expectation over A and can be under-determined. (2) I understand \eta_n corresponds to the complexity of the function class Q, but Condition 4.4 is stated in a unclear way. (3) Assumption 4.10 seems to assume certain coverage condition of the dataset, and should be made clearer (e.g. provide a simpler and sufficient condition for 4.10, possibly in terms of the concentrability). (4) Assumption 4.9 is also difficult to interpret.

4. It is not clear why the parameter eps_{\lambda_n} is introduced.

**Contribution.** It is hard to accurately evaluate the contribution of this paper given its readability  However, it seems to me that most tricks of this paper have already appeared in previous work. It could be beneficial to provide a more detailed comparison to previous works, e.g., Lu et al. 2022, and maybe also the paper of https://proceedings.mlr.press/v162/guo22a/guo22a.pdf.

**Questions:**

See the discussion of weakness.

---

> ### Author Response · Authors · 2024-11-25
> **Response to Reviewer Uxa7**
>
> > **Weakness.** The notations are horrible...
>
> **Response.** We thank you for your careful review and sincerely apologize for any inconvenience caused while reading our paper. In the revised manuscript, we have replaced $W$ with $V$ and reformulated the problem setup to eliminate the need for notation representing the general observational space.
>
> ---
>
> > **Weakness.** The bounds hide most dependency on the parameters...
>
> **Response.** We introduce $o_p(\cdot)$ solely to demonstrate the consistency of certain estimators of $\mathbf{h}^*_\alpha$, thereby justifying the local expansion around the neighborhood of $\mathbf{h}^*_\alpha.$ The $o_p(1)$ term appearing in the main theorem is included for simplicity of presentation, and the hidden variable it represents is inconsequential. All dependencies on other parameters or constants are detailed in the proofs provided in the appendix. We also note that the dependency on the constants specified in Assumptions 4.9 and 4.10 is explicitly reflected in the convergence rate of regret presented in Theorem 4.11.
>
> ---
>
> > **Weakness.** The assumptions are stated messily and lack sufficient explanation...
>
> **Response.** As stated in Global Response I, we have added Section N in the Appendix to provide a detailed discussion of the assumptions and conditions presented in Section 4. Specifically:
> - For Assumption 4.1, please refer to Global Response I, point A.
> - We acknowledge that Condition 4.4 may initially seem challenging to interpret, as it is specifically formulated to solve a quadratic equation of the test function $\theta$ (see Appendix Section G.1) in the proof for deriving a statistical fast rate. Similar formulations can be found in classical textbooks and papers that focus on achieving statistical fast rates, such as Theorem 14.20 in [Wainwright (2019)](#ref-wainwright2019) and Lemma 12 in [Foster and Syrgkanis (2023)](#ref-foster2023). Unfortunately, these sources also lack an intuitive explanation. For additional discussion on Condition 4.4, please refer to Global Response II, point C.
> - Regarding Assumption 4.10, we clarify its connection to the standard concentrability assumption in Global Response II, point E.
> - For Assumption 4.9, please see Global Response II, point D.
> - For Assumption 4.9, please see Global Response II, point D.
> - In short, we need $\epsilon_{\lambda_n}$ for the consistency result in Theorem 4.12. to hold. Intuitively, when $\epsilon_{\lambda_n}$ is small, the rate of convergence of $\hat{h}^{\pi^*}_{R}$ is slow. Consequently, the local expansion error $c_0$ could be large, which in terms leads to a slower regret rate.  For the technical details, please refer to the proof of the Theorem C.3 in Appendix G.3.
>
> > **Contribution.** It is hard...
>
> **Response.** For details on how we have improved readability, please refer to Global Response III, point 2. We assume the reviewer is referring to [Lu et al. (2022)](#ref-lu2022). The primary distinction between our paper and the two listed studies lies in what we consider the most substantial contribution of our work: we optimize the average structural quantile of the reward distribution, whereas they focus on the average reward. Unlike the average reward, which is linear in the structural function, the average structural quantile is a **nonlinear function** of the unknown structural quantile function. Many of the assumptions and technical components in our paper are explicitly designed to address this challenge, including Assumption 4.1, 4.7, 4.9 and Theorem 4.8. These assumptions are standard in quantile IV literature.
>
> We have explicitly highlighted the challenges and distinctions of our work in the introduction section, motivated why policymakers might prefer quantile objectives in Section 2, and emphasized the associated technical problems throughout Section 4. Additionally, we have clarified how our work compares with the two stydues in the "Quantile Objective and Regret" paragraph of Section 2:
>
> *Existing works focus on learning the policy that maximizes the average reward, $\mathbb{E}_{\hat{\pi}}[Y]$ [Lu et al. (2022)](#ref-lu2022), [Guo et al. (2022)](#ref-guo2022). The main goal of this paper is to maximize the average structural quantile of the reward.*
>
> **References.**
> - [Wainwright (2019)](#ref-wainwright2019): Martin J. Wainwright. *High-dimensional statistics: A non-asymptotic viewpoint.* Volume 48. Cambridge University Press, 2019.
> - [Foster and Syrgkanis (2023)](#ref-foster2023): Dylan J. Foster and Vasilis Syrgkanis. *Orthogonal statistical learning.* The Annals of Statistics, 51(3):879–908, 2023.
> - [Guo et al. (2022)](#ref-guo2022): Hongyi Guo, Qi Cai, Yufeng Zhang, Zhuoran Yang, and Zhaoran Wang. *Provably efficient offline reinforcement learning for partially observable Markov decision processes.* In *International Conference on Machine Learning,* pp. 8016–8038. PMLR, 2022.
> - [Lu et al. (2022)](#ref-lu2022): Miao Lu, Yifei Min, Zhaoran Wang, and Zhuoran Yang.

---

> > ### Comment · Reviewer_Uxa7 · 2024-12-02
> >
> > Thank you for the response. While I appreciate the authors' efforts to address various concerns, the revised paper does not seem satisfactory regarding readability. I am also not fully convinced that the quantile objective substantially differs from the expected objective and poses a significant challenge. Therefore, I will keep my original score.

---

### Official Review · Reviewer_K74W · 2024-11-07

**Soundness:** 4
**Presentation:** 2
**Contribution:** 2
**Rating:** 5
**Confidence:** 4

**Summary:**

The paper studies offline reinforcement learning with three main differentiators:

1. the objective (cumulative rewards) is measured in $\alpha$-quantile, rather than expectation.

2. observations have latent confounders.

3. how to define sufficient coverage in this setting.

The proposed method runs in two steps: (1) perform causal inference to get a nonparameteric model and (2) apply pessimism principle. Then for the practical use, a policy optimization method is proposed. It is claimed that this is the first sample-efficient policy learning algorithm in this setting.

**Strengths:**

- This new setting sounds interesting, and authors did a good job of motivating the setting relatively well (though also see weakness on this point).

- The writing quality is in general quite good, except the technical part (also see weakness on this point).

**Weaknesses:**

- The authors motivated the unmeasured confounders with a healthcare example, and quantile optimization with job training programs. However, these are distinct cases, and I wonder whether there is one *unified* motivating example that requires both.

-  Section 4, while highly technical, is the most important part of the paper as the main contribution of the paper claims novel methodologies. Such section must be written with extra care on the clarity even though it conveys complex ideas. As it stands now, it is almost impossible to decipher this section. Here are a few main points that call confusions:

1. How strong is Assumption 4.1? What would this imply if you bring it to the tabular setting for instance?

2. Theorem 4.5 -- again, if you could explain this in the tabular setting, that might help a lot.

3. Presentation became overly complicated by combining the quantile-objective formulation and nonparametric approaches, which I began to doubt why such complication was necessary in this work, and what does it have to do with novelty of the proposed method.

4. I think not many readers, even the experts in RL, can go through the subsections here.




- I think the experimental section could have been reinforced in the main text. The experimental setup is not clear: what are the "p% of random noises"? Does it mean a random exploration?

- It is also not very clear what the plots are saying: small regret if sample size grows large? Why is that surprising?

**Questions:**

- It is confusing whether we have two datasets, one with intervention and the other without intervention. If Z is an instrumental variable, does it appear in the dataset?

---

> ### Author Response · Authors · 2024-11-25
> **Reponse to Reviewer K74W**
>
> > **Weakness.** The authors motivated the unmeasured confounders with a healthcare example, and quantile optimization with job training programs. However, these are distinct cases, and I wonder whether there is one unified motivating example that requires both.
>
> **Response.** We actually motivate both the unmeasured confounding and quantile optimization using the job training programs. In Section 2, line 130, we state that: "$U_i$ could represent unmeasured factors such as the worker's motivation, which influence both their participation decision $A_i$ and their utility outcomes $Y_i.$"
>
> Another example is in finance. Consider the design of a finance portfolio optimization model, aimed at minimizing portfolio risk. In this setting, $Y_i$ represents the negative loss generated by the $i$-th portfolio. $X_i$ refers to the pre-investment characteristics of the portfolio, such as market conditions and investor risk tolerance. $A_i$ denotes the portfolio strategy implemented, such as the weighting of different asset classes. $Z_i$ serves as an instrumental variable that influences the portfolio strategy but does not directly affect the portfolio's returns, such as regulatory requirements or external economic indicators. In practice, financial institutions often seek to minimize a specific quantile of the loss distribution, a measure known as Value at Risk (VaR) [Duffie and Pan (1997)](#ref-duffie1997) in portfolio theory. The reason is that the expected loss can be heavily skewed by highly volatile portfolios, making the mean loss a less reliable measure of risk.  Examples of unmeasured confounders include investor sentiment, which may simultaneously influence asset allocation decisions (treatment) and market performance (outcome), or unobserved market shocks, such as geopolitical events or unexpected economic announcements.
>
> We also include a motivating example of hospital readmission times for the negative controls in Appendix Section J.1 (p. 33).
>
> ---
> > **Weakness.** Section 4, while highly technical... Here are a few main points that cause confusion: How strong is Assumption 4.1? What would this imply if you bring it to the tabular setting, for instance?
>
> **Response.** Please refer to Global Response I, point A, for a detailed discussion of Assumption 4.1. Specifically, for the tabular case where the context $X$ and action $A$ are discrete, Theorem 2 of [Chernozhukov and Hansen (2005)](#ref-chernozhukov2005) provides sufficient conditions for this global identification result to hold. We also remark that our paper, like almost all studies in nonparametric quantile IV literature, assumes that the reward $Y$ is continuous. Please see Global Response I, point B, for a discussion of continuous rewards.
>
> ---
>
> > **Weakness.** Theorem 4.5 -- again, if you could explain this in the tabular setting, that might help a lot.
>
> **Response.** Theorem 4.5 holds as long as Assumptions 4.1–4.3 hold, regardless of whether other variables are discrete or not.
>
> ---
>
> > **Weakness.** Presentation became overly complicated... I think not many readers, even the experts in RL, can go through the subsections here.
>
> **Response.** We sincerely apologize for any inconvenience caused while reading our paper. For details on how we plan to improve its readability, please refer to Global Response III, point 2.
>
> ---
>
> > **Weakness.** I think the experimental section could have been reinforced in the main text.
>
> **Response.** Thank you for your valuable suggestion to strengthen the experimental section. We will incorporate this improvement in the revised manuscript. Additionally, please refer to Global Response III, point 1, for details on the new semi-synthetic data study we plan to analyze.
>
> ---
> **References.**
> - [Chernozhukov and Hansen (2005)](#ref-chernozhukov2005): Victor Chernozhukov and Christian Hansen. *An IV model of quantile treatment effects.* Econometrica, 73(1):245–261, 2005.
> - [Duffie and Pan (1997)](#ref-duffie1997): Darrell Duffie and Jun Pan. *An overview of value at risk.* Journal of Derivatives, 4(3):7–49, 1997.

---

> > ### Author Response · Authors · 2024-11-25
> > **Reponse to Reviewer K74W**
> >
> > > **Weakness.** The experimental setup is not clear: what are the "p% of random noises"? Does it mean random exploration?
> >
> > **Response.** Yes, it is similar to random exploration. Specifically, among the $n$ samples collected from the ODCP, we consider offline datasets where p% of the actions are distributed as random noise, independent of $(X, Z, U),$ and affecting only $Y.$ We deliberately contaminate the offline dataset to evaluate the robustness of our algorithm. One of the most powerful features of pessimism [Jin et al. (2021)](#ref-jin2021) is its ability to handle suboptimal actions that may mislead the learned policy due to undercoverage in the offline dataset. Through this simulation experiment, we aim to demonstrate this advantage. We have added this clarification to the revised manuscript.
> >
> > ---
> >
> > > **Weakness.** It is also not very clear what the plots are saying: small regret if the sample size grows large? Why is that surprising?
> >
> > **Response.** Yes, you are absolutely correct that the decrease in regret as the sample size increases is not surprising. As mentioned in our previous response, the main takeaway from the plot is to demonstrate that the proposed algorithm can still learn the optimal policy even when a large proportion of the actions are contaminated. The proportion of random actions, p%, can be as high as 70%, as shown in the rightmost plot.
> >
> > ---
> >
> > > **Question.** It is confusing whether we have two datasets, one with intervention and the other without intervention. If $Z$ is an instrumental variable, does it appear in the dataset?
> >
> > **Response.** We only have one dataset, which is generated according to the offline data collection process (ODCP). As stated on lines 123–132 of page 1, the offline dataset consists of {$x_i, a_i, z_i, y_i$}$_{i=1}^{n}.$ Hence, the dataset includes $Z.$ After learning the policy $\hat{\pi}$ from this offline dataset, the policy is applied as a decision-making rule on new data generated during the interventional process. This approach aligns with the standard framework of offline reinforcement learning, where a policy learned from offline data is tested or deployed in a different environment or distribution.
> >
> > **Reference.**
> > - [Jin et al. (2021)](#ref-jin2021): Ying Jin, Zhuoran Yang, and Zhaoran Wang. *Is pessimism provably efficient for offline RL?* In *International Conference on Machine Learning,* pp. 5084–5096. PMLR, 2021.

---

### Official Review · Reviewer_6Knv · 2024-11-10

**Soundness:** 3
**Presentation:** 2
**Contribution:** 2
**Rating:** 3
**Confidence:** 3

**Summary:**

This paper studies the problem of identifying the policy whose reward distribution with the largest $\alpha$-quantile. The authors are particularly interested in the setting when there is unmeasured confounding i.e. not all variables that affect the choice of the actions in the offline datasets are recorded in the dataset. The general setting of unmeasured confounding is difficult and often impossible and the authors focus on two cases --  (a) instrumental variables, and (b) negative controls. The proposed approach effectively combines conditional moment restrictions (popular in econometrics) with pessimism based policy optimization (popular in offline reinforcement learning). The authors provide theoretical guarantees and statistical rate of the proposed method under reasonable assumptions on the data generating process.

**Strengths:**

1. I think the authors study an important problem as the issue of unmeasured confounding / partial observability is quite common in offline reinforcement learning. The proposed method also combines causal inference techniques with pessimism based policy optimization methods from offline reinforcement learning.

2. The theoretical results are interesting and provide statistical rates under the two settings -- (a) instrumental variables, and (b) negative controls. However, the results are proven under strong assumptions (see comments in the next section).

**Weaknesses:**

1. Since the paper deals with causal inference with unmeasured confounding, it should include experiments on real-world datasets. Handling real-world datasets introduces many challenges e.g. discrete outcomes, missing covariates etc. The current experimental setup considers a simulation based setting that considers linear functionals, one-dimensional context, and is too simple to demonstrate the effectiveness of the proposed method.

2. The authors make several strong assumptions in order to prove theoretical results. First, it is assumed that the parameter $\eta_n = O(n^{-1/2})$ but this is only true for simple classes of functions (as shown in appendix E) and is not true in general. Second, assumption 4.7 assumes that the conditional density is continuous and it eliminates all causal inference problems with discrete outcomes. Third, I believe assumption 4.9 is quite strong and there is no justification as to why it should be true.

3. Overall, I found the paper hard to follow. Many steps require further explanation and the assumptions require further justification.

**Questions:**

1. Can you please justify why assumption 4.9 should be true?

2. Solving offline reinforcement learning requires some type of coverage condition on the data e.g. single-policy concentrability. In the paper, I haven't seen any explicit mention of such coverage conditions. Can you please clarify what you assume regarding the overlap between the optimal policy and the data-generating policy?

---

> ### Author Response · Authors · 2024-11-25
> **Response to  Reviewer 6Knv**
>
> > **Weakness.** Since the paper deals with causal inference with unmeasured confounding, it should include experiments on real-world datasets. Handling real-world datasets introduces many challenges (e.g., discrete outcomes, missing covariates, etc.). The current experimental setup considers a simulation-based setting that uses linear functionals, one-dimensional context, and is too simple to demonstrate the effectiveness of the proposed method.
>
> **Response.** For a discussion on discrete outcomes, please refer to Global Response I point B. For experiments on real-world datasets, see Global Response III point 1).
>
>
>
> > **Weakness.** The authors make several strong assumptions in order to prove theoretical results. First, it is assumed that the parameter $\eta_n = O(n^{-1/2})$, but this is only true for simple classes of functions (as shown in Appendix E) and is not true in general. Second, Assumption 4.7 assumes that the conditional density is continuous, which eliminates all causal inference problems with discrete outcomes. Third, I believe Assumption 4.9 is quite strong, and there is no justification as to why it should be true.
>
> **Response.** For the first concern, please refer to Global Response II point C. For Assumption 4.7, see Global Response I point B. For Assumption 4.9, see Global Response II point D.
>
>
>
> > **Weakness.** Overall, I found the paper hard to follow. Many steps require further explanation, and the assumptions require further justification.
>
> **Response.** Please refer to Global Response III point 2).
>
>
>
> > **Question.** Can you please justify why Assumption 4.9 should be true?
>
> **Response.** Please refer to Global Response point II D.
>
>
>
> > **Question.** Solving offline reinforcement learning requires some type of coverage condition on the data (e.g., single-policy concentrability). In the paper, I haven't seen any explicit mention of such coverage conditions. Can you please clarify what you assume regarding the overlap between the optimal policy and the data-generating policy?
>
> **Response.** Please refer to Global Response point II E.

---

### Author Response · Authors · 2024-11-25
**Global Response I**

We sincerely thank all four reviewers for their time and effort in carefully reading our paper and providing valuable feedback. In this global response, we aim to address common concerns raised by the reviewers and clarify key aspects of our work. Subsequently, we address each reviewer's specific concerns in detail and outline the corresponding revisions, all of which have been incorporated into the updated manuscript.

Several reviewers noted that the assumptions presented in Section 4 require further justification. In response, we added a separate section in the appendix to provide a detailed discussion of each assumption. We also clarify each of the assumptions here.

**A. Assumption 4.1 (Identifiability and Realizability).** This assumption, often referred to as a global identification assumption, is commonly made in nearly all nonparametric quantile IV literature. For instance, see Assumption 3.2(ii) in [Chen and Pouzo (2012)](#ref-chen2012), Assumption 1 in [Gagliardini and Scaillet (2012)](#ref-gagliardini2012), and Assumption 1 in [Horowitz and Lee (2007)](#ref-horowitz2007).

Regarding sufficient conditions for this assumption to hold, [Chernozhukov and Hansen (2005)](#ref-chernozhukov2005) provides such conditions in the case where the context $X$ is discrete. For more general setups, [Chen et al. (2014)](#ref-chen2014) discusses conditions for local identification in detail. The only study we are aware of that carefully outlines conditions for global identification is [Wong (2022)](#ref-wong2022). However, Wong's approach relies on a set of strong regularity conditions to establish these results.

**B. Assumption 4.7 (Regularity of Conditional Density).** This paper assumes that the reward variable $Y$ is continuous. The primary reason is that we are dealing with a **nonlinear** functional $\mathcal{T} \mathbf{h}$. To link the regret to $\mathcal{T} \mathbf{h}$, we conduct a local expansion of $\mathcal{T} \mathbf{h}$ around the neighborhood of the structural quantile function $\mathbf{h}^*_\alpha$. For this expansion, it is essential that the density of $Y$ is continuous and bounded, ensuring that the pathwise derivative $\mathcal{T}\mathbf{h}$ at $\mathbf{h} = \mathbf{h}^*_\alpha$ exists. We remark that this assumption is needed in most of the nonparametric quantile IV literature. For instance, see Condition 6.1(i), (iv) in [Chen and Pouzo (2012)](#ref-chen2012), Theorem 3.1 in [Abadie et al. (2002)](#ref-abadie2002), Condition A.1 in [Chernozhukov and Hansen (2005)](#ref-chernozhukov2005), Assumption A.1 in [Gagliardini and Scaillet (2012)](#ref-gagliardini2012). While our paper does not address discrete outcomes due to this constraint, we believe a similar algorithm could be applied in this case by first convolving the discrete outcome with Gaussian noise, which would smooth the distribution of $Y$. This smoothing allows for the use of techniques designed for continuous outcomes.

**References**
- [Chen and Pouzo (2012)](#ref-chen2012): Xiaohong Chen and Demian Pouzo. *Estimation of nonparametric conditional moment models with possibly nonsmooth generalized residuals.* Econometrica, 80(1):277–321, 2012.
- [Abadie et al. (2002)](#ref-abadie2002): Alberto Abadie, Joshua Angrist, and Guido Imbens. *Instrumental variables estimates of the effect of subsidized training on the quantiles of trainee earnings.* Econometrica, 70(1):91–117, 2002.
- [Chernozhukov and Hansen (2005)](#ref-chernozhukov2005): Victor Chernozhukov and Christian Hansen. *An IV model of quantile treatment effects.* Econometrica, 73(1):245–261, 2005.
- [Chen et al. (2014)](#ref-chen2014): Xiaohong Chen, Victor Chernozhukov, Sokbae Lee, and Whitney K Newey. *Local identification of nonparametric and semiparametric models.* Econometrica, 82(2):785–809, 2014.
- [Gagliardini and Scaillet (2012)](#ref-gagliardini2012): Patrick Gagliardini and Olivier Scaillet. *Nonparametric instrumental variable estimation of structural quantile effects.* Econometrica, 80(4):1533–1562, 2012.
- [Horowitz and Lee (2007)](#ref-horowitz2007): Joel L Horowitz and Sokbae Lee. *Nonparametric instrumental variables estimation of a quantile regression model.* Econometrica, 75(4):1191–1208, 2007.
- [Van der Vaart and Wellner (2013)](#ref-van2013): A. van der Vaart and J. Wellner. *Weak Convergence and Empirical Processes: With Applications to Statistics.* Springer Series in Statistics. Springer New York, 2013.
- [Wong (2022)](#ref-wong2022): Wing Hung Wong. *An equation for the identification of average causal effect in nonlinear models.* Statistica Sinica, 32:539–545, 2022.
- [Miao et al. (2023)](#ref-miao2023): Rui Miao, Zhengling Qi, Cong Shi, and Lin Lin. *Personalized pricing with invalid instrumental variables: Identification, estimation, and policy learning.* arXiv preprint arXiv:2302.12670, 2023.

---

### Author Response · Authors · 2024-11-25
**Global Response II**

**C. Condition 4.4:** We only verify that Condition 4.4 holds true for linear function classes because they already encompass a wide variety of functions, including polynomial splines, B-splines, wavelets, and Fourier series basis functions. These function classes can be represented as linear combinations of basis functions, making them instances of the linear function class. By examining our proof in Appendix E, we rely only on the property that linear function classes have a covering number and bracketing number bounded by $A_n \log \left(\frac{C_3}{t}\right)$ where $A_n$ is the dimension of the linear function classes. Therefore, we can actually obtain different convergence rates \(\eta_n\) under the supremum norm for the chosen function classes. The bracketing and covering numbers of various function classes can be found in different books. For example, [Van der Vaart and Wellner (2013)](#ref-van2013) provides an extensive discussion of covering numbers and bracketing numbers. They discuss these concepts for a variety of function classes, including Sobolev spaces, Besov spaces, Hölder classes, and reproducing kernel Hilbert spaces.

**D. Assumption 4.9 (Local Curvature for the Estimator of the Solution Set Algorithm).** In the initial version of our submission, this assumption was divided into two components. Upon reexamining the proof, we made slight modifications that allowed us to eliminate the need for Assumption 4.9(i). As a result, only Assumption 4.9(ii) is required, which asserts that the weaker pseudo-metric $\mathbf{h} - \mathbf{h}^*_\alpha$ is Lipschitz continuous with respect to the population criterion function, the $l_2$ norm of $\mathcal{T} \mathbf{h}$  within a small neighborhood around $\mathbf{h}^*_\alpha.$ This condition is relatively mild and is commonly employed in the nonlinear functional analysis literature [Chen and Pouzo (2012)](#ref-chen2012), [Miao et al. (2023)](#ref-miao2023). Condition 6.3(i) in [Chen and Pouzo (2012)](#ref-chen2012) provides a sufficient condition for the assumption to hold in a broad class of functional spaces $\mathcal{H}$, including the linear function spaces we consider in Appendix E.

**E. Assumption 4.10 (Change of Measure).** The Change of Measure Assumption 4.10 is actually a weaker substitute for the standard concentrability assumption commonly used in the RL literature. In the tabular setting, Assumption 4.10 is satisfied when the right-hand side of the assumption is uniformly bounded, and the Moore-Penrose inverse of the probability mass matrix $P(Z \mid A, X)^+$ exists. This condition holds if the instrumental variable (IV) satisfies the standard completeness assumption: $\text{rank}[P(Z \mid A, X)] \geq |\mathcal{A}| \times |\mathcal{X}|.$ We note that Theorem 4.11 and Theorem 4.13 remain valid if Assumption 4.10 is replaced with the usual single-policy concentrability assumption: there exists a constant $\tilde{c} > 0$ such that $\sup_{x \in \mathcal{X}, a \in \mathcal{A}} \frac{\tilde{p}(x)\pi^*(a \mid x)}{p(x, a)} \leq \tilde{c}$. Note $\frac{\tilde{p}(x)\pi^*(a \mid x)}{p(x, a)}$ represents the density ratio of the distribution under the interventional process with the oracle policy $\pi^*$ over that of the offline data collection process.



**References**
- [Chen and Pouzo (2012)](#ref-chen2012): Xiaohong Chen and Demian Pouzo. *Estimation of nonparametric conditional moment models with possibly nonsmooth generalized residuals.* Econometrica, 80(1):277–321, 2012.
- [Van der Vaart and Wellner (2013)](#ref-van2013): A. van der Vaart and J. Wellner. *Weak Convergence and Empirical Processes: With Applications to Statistics.* Springer Series in Statistics. Springer New York, 2013.
- [Miao et al. (2023)](#ref-miao2023): Rui Miao, Zhengling Qi, Cong Shi, and Lin Lin. *Personalized pricing with invalid instrumental variables: Identification, estimation, and policy learning.* arXiv preprint arXiv:2302.12670, 2023.

---

### Author Response · Authors · 2024-11-25
**Global Response III**

**1)** Several authors have argued that our experimental setting is naive and lacks a real-world data application. The primary reason we do not include a real data study is that our method relies on causal intervention based on the given context $X$ during the interventional process. Specifically, conducting a real data analysis would require access to the counterfactual potential outcomes $Y_i(a)$ for each of the $i$-th trial for some random action $a \in \mathcal{A},$ which depends on the realization of the randomized policy $\hat{\pi}(X_i)$. However, a fundamental challenge in causal inference is that we cannot observe all potential outcomes; for each trial, only one outcome is observable. As a result, no real dataset exists that allows us to evaluate the performance of our method. Second, we do not include comprehensive experiments as our work is primarily theoretical, focusing on a provably sample-efficient quantile-optimal policy learning algorithm. We introduce the first unified framework that integrates quantile causal inference with offline reinforcement learning.

However, we agree with the reviewers that experiments would strengthen the value of our results. In the camera-ready version, we would apply our method to a semi-synthetic dataset derived from the Job Search Intervention Study [Jo and Stuart (2009)](#ref-jo2009use)
 which investigates the quantile effect of a job training program on earnings. This dataset includes an eight-dimensional context for each participant. We will compare our algorithm against the standard policy learning algorithm that aims to maximize the average effect.

**2)** Several reviewers expressed concerns about the readability of the theoretical analysis in Section 4, noting that it is overly detailed and questioning whether such complexity is necessary. We apologize for the dense content filled with technical details. The analysis of the nonparametric quantile treatment effect is inherently technical, as it involves linking the regret to the nonlinear criterion function, $l_2$ norm of $\mathcal{T}\mathbf{h}$ through a local expansion around the unknown structural quantile function $\mathbf{h}^*_\alpha.$ We aimed to rigorously establish the statistical fast rate of regret within the ten-page limit of ICLR.

To improve readability, we addressed all notational issues that may cause confusion in the manuscript. Additionally, we moved many technical components to the appendix, including the detailed discussion of Condition 4.4, the treatment of pathwise derivatives, and the theoretical results. In their place, we will shift the explanation of the experimental setting and the real data study to the main sections in the camera-ready version, making the paper more accessible to readers.

**References**
- [Jo and Stuart (2009)](#ref-jo2009use): Booil Jo and Elizabeth A. Stuart. *On the use of propensity scores in principal causal effect estimation.* Statistics in Medicine, 28(23):2857–2875, 2009.

---

### Note · Authors · 2024-12-05

**Comment:**

We have decided to withdraw our submission. We sincerely thank all four reviewers for their valuable feedback, which will undoubtedly help us improve our work further.

**Withdrawal Confirmation:**

I have read and agree with the venue's withdrawal policy on behalf of myself and my co-authors.